# Universal Gradient Descent Ascent Method for Nonconvex-Nonconcave Minimax Optimization

**Taoli Zheng**
CUHK
tlzheng@se.cuhk.edu.hk

**Linglingzhi Zhu**
CUHK
llzzhu@se.cuhk.edu.hk

**Anthony Man-Cho So**
CUHK
manchoso@se.cuhk.edu.hk

**José Blanchet**
Stanford University
jose.blanchet@stanford.edu

**Jiajin Li**$^*$
Stanford University
jiajinli@stanford.edu

## Abstract

Nonconvex-nonconcave minimax optimization has received intense attention over the last decade due to its broad applications in machine learning. Most existing algorithms rely on one-sided information, such as the convexity (resp. concavity) of the primal (resp. dual) functions, or other specific structures, such as the Polyak-Łojasiewicz (PŁ) and Kurdyka-Łojasiewicz (KŁ) conditions. However, verifying these regularity conditions is challenging in practice. To meet this challenge, we propose a novel universally applicable single-loop algorithm, the doubly smoothed gradient descent ascent method (DS-GDA), which naturally balances the primal and dual updates. That is, DS-GDA with the same hyperparameters is able to uniformly solve nonconvex-concave, convex-nonconcave, and nonconvex-nonconcave problems with one-sided KŁ properties, achieving convergence with $\mathcal{O}(\epsilon^{-4})$ complexity. Sharper (even optimal) iteration complexity can be obtained when the KŁ exponent is known. Specifically, under the one-sided KŁ condition with exponent $\theta \in (0,1)$, DS-GDA converges with an iteration complexity of $\mathcal{O}(\epsilon^{-2\max\{2\theta,1\}})$. They all match the corresponding best results in the literature. Moreover, we show that DS-GDA is practically applicable to general nonconvex-nonconcave problems even without any regularity conditions, such as the PŁ condition, KŁ condition, or weak Minty variational inequalities condition. For various challenging nonconvex-nonconcave examples in the literature, including "Forsaken", "Bilinearly-coupled minimax", "Sixth-order polynomial", and "PolarGame", the proposed DS-GDA can all get rid of limit cycles. To the best of our knowledge, this is the first first-order algorithm to achieve convergence on all of these formidable problems.

## 1 Introduction

In this paper, we are interested in studying nonconvex-nonconcave minimax problems of the form

$$\min_{x \in \mathcal{X}} \max_{y \in \mathcal{Y}} f(x,y), \tag{P}$$

where $f : \mathbb{R}^n \times \mathbb{R}^d \to \mathbb{R}$ is nonconvex in $x$ and nonconcave in $y$, and $\mathcal{X} \subseteq \mathbb{R}^n$, $\mathcal{Y} \subseteq \mathbb{R}^d$ are convex compact sets. Such problems have found significant applications in machine learning and operation research, including generative adversarial networks [27, 2], adversarial training [44, 53], multi-agent

---

$^*$Corresponding author

37th Conference on Neural Information Processing Systems (NeurIPS 2023).

reinforcement learning [16, 48], and (distributionally) robust optimization [6, 19, 38, 25, 7], to name a few.

For smooth functions, one natural idea is to use gradient descent ascent (GDA) [41], which applies gradient descent on the primal function and gradient ascent on the dual function. However, GDA is originally designed for the strongly-convex-strongly-concave problem, where either primal or dual players can dominate the game. When applying it to the nonconvex-concave case, a so-called two-timescale method can make it converge. In this scenario, the primal player is the dominant player in the game. We can regard this one-step GDA scheme as an inexact subgradient descent on the inner max function $\max_{y \in \mathcal{Y}} f(x, y)$, thus it is necessary for the dual update to be relatively faster than the primal update at each iteration. However, this two-timescale GDA yields a high iteration complexity of $\mathcal{O}(\epsilon^{-6})$. To achieve a lower iteration complexity, smoothed GDA (S-GDA) [62, 59] employs the Moreau-Yosida smoothing technique to stabilize the primal sequence. The resulting stabilized sequence can help S-GDA achieve a lower iteration complexity of $\mathcal{O}(\epsilon^{-4})$. Alternating Gradient Projection (AGP) [57] can also achieve this lower iteration complexity by adding regularizers to both primal and dual functions. It is worth noting that the convergence of all these algorithms is heavily contingent on the convexity/concavity of primal/dual functions, leading to asymmetric updates. In the convex-nonconcave scenario, the roles are reversed, and the dual player takes dominance in the game. To apply the aforementioned algorithms, their updates should be modified accordingly. Specifically, for SGDA and AGP, the smoothing and regularized sides also need to be changed to guarantee the optimal convergence rate. However, when dealing with nonconvex-nonconcave problems, no player inherently dominates the other and all existing first-order algorithms cannot be guaranteed to converge to stationary points (see Definition 1), and they can even suffer from *limit cycles*. That is, the generated trajectories of all these algorithms will converge to cycling orbits that do not contain any stationary point of $f$. Such *spurious* convergence phenomena arise from the inherent minimax structure of (P) and have no counterpart in pure minimization problems. Conceptually, nonconvex-nonconcave minimax optimization problems can be understood as a seesaw game, which means no player inherently dominates the other. More explicitly, the key difficulty lies in adjusting the primal and dual updates to achieve a good balance. Most existing works address this challenge by adding additional regularity conditions to ensure the automatic domination of one player over the other. Specifically, research works either impose the global Polyak-Łojasiewicz (PŁ) condition on the dual function $f(x, \cdot)$ [59, 47, 22, 58] or assume the satisfaction of $\alpha$-dominance condition [29, 30]. Both approaches fall into this line of research, enabling the adoption of algorithms and analysis for nonconvex-concave minimax problems. Although the recent work [40] extends the PŁ condition to the one-sided Kurdyka-Łojasiewicz (KŁ) property, it is still hard to provide the explicit KŁ exponent, and prior knowledge regarding which side satisfies the KŁ condition is required to determine the appropriate side to employ extrapolation. If we choose the wrong side, it will result in a slow convergence or even divergence (see Figure 5). On another front, variational inequality (VI) provides a unified framework for the study of equilibrium/minimax problems [46, 36, 26, 60, 45]. However, VI-related conditions are usually hard to check in practice. Hence, the convergence of the existing algorithms all highly rely on prior knowledge of the primal/dual function, which makes it paramount to design a universal algorithm for convex-nonconcave, nonconvex-concave, and nonconvex-nonconcave minimax problems.

We propose a new algorithm called the Doubly Smoothed Gradient Descent Ascent (DS-GDA) algorithm. DS-GDA builds upon S-GDA by applying the Moreau-Yosida smoothing technique to both the primal and dual variables, which allows a better trade-off between primal and dual updates. All hyperparameters, including the stepsize for gradient descent and ascent steps, and extrapolation parameters are carefully and explicitly controlled to ensure the sufficient descent property of a novel Lyapunov function that we introduce in our paper. The carefully selected variables automatically decide the dominant player, thereby achieving the balance of primal and dual updates by the interaction of parameters. Furthermore, the doubly smoothing technique enables the use of a set of symmetric parameters to achieve universality. This stands in sharp contrast to S-GDA, where only primal/dual function is smoothed. Specifically, the regularized function and the primal-dual updates in DS-GDA are inherently symmetric, which provides the possibility of applying the DS-GDA without prior information on the primal and dual functions.

To validate our idea and demonstrate the universality of DS-GDA, we provide a visual representation of the feasible symmetric parameter selections by relating the regularizer to Lipschitz constants and step sizes. This graphical illustration, depicted in Figure 1a, reveals that numerous parameter settings

can be chosen to guarantee convergence, which showcases the flexibility of DS-GDA. An experiment for a nonconvex-nonconcave problem has also been done to test the efficiency of using symmetric parameters.

We also evaluate the performance of DS-GDA on a range of representative and challenging nonconvex-nonconcave problems from the literature, which violate all known regularity conditions. These include the "Forsaken" example [32, Example 5.2], the "Bilinearly-coupled minimax" example [29], the "Sixth-order polynomial" example [18, 56, 13], and the "PolarGame" example [50]. In all cases, DS-GDA successfully escapes limit cycles and converges to the desired stationary point, while other methods either suffer from the recurrence behavior or diverge. Moreover, our algorithm exhibits robustness to parameter selection (see Section 4.2), offering practical flexibility.

To corroborate its superior performance and have a better understanding of its convergence behaviors, we demonstrate that DS-GDA converges to a stationary point for nonconvex-concave, convex-nonconcave, and nonconvex-nonconcave problems that satisfy a one-sided KŁ property. By employing a single set of parameters, DS-GDA converges with an iteration complexity of $\mathcal{O}(\epsilon^{-4})$ across all these scenarios. Remarkably, DS-GDA achieves this without prior verification of these conditions. However, if we have access to a one-sided KŁ condition or knowledge of the convexity (concavity) of the primal (dual) function, the range of allowable parameters can be expanded. What is more, DS-GDA attains a lower or even optimal iteration complexity of $\mathcal{O}(\epsilon^{-2\max\{2\theta,1\}})$ when the one-sided KŁ property with exponent $\theta \in (0,1)$ is satisfied. Notably, these convergence results match the best results for single-loop algorithms when the dual function is concave [62, 40] or satisfies KŁ condition [40]. To the best of our knowledge, our work demonstrates, for the first time, the possibility of having a simple and unified single-loop algorithm for solving nonconvex-nonconcave, nonconvex-concave, and convex-nonconcave minimax problems. However, it remains an open question whether convergence results can be derived without any regularity conditions. This question is intriguing and warrants further theoretical investigation of nonconvex-nonconcave minimax problems in the future.

Our main contributions are summarized as follows:

(i) We present DS-GDA, the first universal algorithm for convex-nonconcave, nonconvex-concave, and nonconvex-nonconcave problems with one-sided KŁ property. A single set of parameters can be applied across all these scenarios, guaranteeing an iteration complexity of $\mathcal{O}(\epsilon^{-4})$. With the KŁ exponent $\theta \in (0,1)$ of the primal or dual function, we improve the complexity to $\mathcal{O}(\epsilon^{-2\max\{2\theta,1\}})$. Our current convergence analysis achieves the best-known results in the literature.

(ii) We demonstrate that DS-GDA converges on various challenging nonconvex-nonconcave problems, even when no regularity conditions are satisfied. This makes DS-GDA the first algorithm capable of escaping limit cycles in all these hard examples.

## 2  Doubly Smoothed GDA

In this section, we propose our algorithm (i.e., DS-GDA) for solving (P). To start with, we introduce the blanket assumption, which is needed throughout the paper.

**Assumption 1 (Lipschitz gradient)** *The function $f$ is continuously differentiable and there exist positive constant $L_x, L_y > 0$ such that for all $x, x' \in \mathcal{X}$ and $y, y' \in \mathcal{Y}$*

$$\|\nabla_x f(x,y) - \nabla_x f(x',y')\| \leq L_x(\|x-x'\| + \|y-y'\|),$$
$$\|\nabla_y f(x,y) - \nabla_y f(x',y')\| \leq L_y(\|x-x'\| + \|y-y'\|).$$

*For simplicity, we assume $L_y = \lambda L_x = \lambda L$ with $\lambda > 0$.*

For general smooth nonconvex-concave problems, a simple and natural algorithm is GDA, which suffers from oscillation even for the bilinear problem $\min_{x\in[-1,1]} \max_{y\in[-1,1]} xy$. To address this issue, a smoothed GDA algorithm that uses Moreau-Yosida smoothing techniques is proposed in [62]. Specifically, they introduced an auxiliary variable $z$ and defined a regularized function as follows:

$$F(x,y,z) := f(x,y) + \frac{r}{2}\|x-z\|^2.$$

The additional quadratic term smooths the primal update. Consequently, the algorithm can achieve a better trade-off between primal and dual updates. We adapt the smoothing technique to the nonconvex-nonconcave setting, where the balance of primal and dual updates is not a trivial task. To tackle this

problem, we also smooth the dual update by subtracting a quadratic term of dual variable and propose a new regularized function $F : \mathbb{R}^n \times \mathbb{R}^d \times \mathbb{R}^n \times \mathbb{R}^d \to \mathbb{R}$ as

$$F(x, y, z, v) := f(x, y) + \frac{r_1}{2}\|x - z\|^2 - \frac{r_2}{2}\|y - v\|^2$$

with different smoothed parameters $r_1 > L_x$, $r_2 > L_y$ for $x$ and $y$, respectively. Then, our DS-GDA is formally presented in Algorithm 1.

---

**Algorithm 1:** Doubly Smoothed GDA (DS-GDA)

---

**Data:** Initial $x^0, y^0, z^0, v^0$, stepsizes $\alpha, c > 0$, and extrapolation parameters $0 < \beta, \mu < 1$
1 **for** $t = 0, \dots, T$ **do**
2 $\quad x^{t+1} = \text{proj}_{\mathcal{X}}(x^t - c\nabla_x F(x^t, y^t, z^t, v^t))$;
3 $\quad y^{t+1} = \text{proj}_{\mathcal{Y}}(y^t + \alpha\nabla_y F(x^{t+1}, y^t, z^t, v^t))$;
4 $\quad z^{t+1} = z^t + \beta(x^{t+1} - z^t)$;
5 $\quad v^{t+1} = v^t + \mu(y^{t+1} - v^t)$.
6 **end**

---

The choice of $r_1$ and $r_2$ is crucial for the convergence of the algorithm in both theoretical and practical senses. In particular, when $r_1 = r_2$, it reduces to the *proximal-point mapping* proposed in [43] and inexact proximal point method (PPM) is only known to be convergent under certain VI conditions. Even with the exact computation of proximal mapping, PPM will diverge in the absence of regularity conditions [29]. By contrast, with an unbalanced $r_1$ and $r_2$, our algorithm can always converge. The key insight here is to carefully adjust $r_1$ and $r_2$ to balance the primal-dual updates, ensuring the sufficient descent property of a novel Lyapunov function introduced in our paper. In fact, as we will show later in Section 3, $r_1$ and $r_2$ are typically not equal theoretically and practically. The two auxiliary variables $z$ and $v$, which are updated by averaging steps, are also indispensable in our convergence proof. Intuitively, the exponential averaging applied to proximal variables $z$ and $v$ ensures they do not deviate too much from $x$ and $y$, contributing to sequence stability.

We would like to highlight that the way we use the Moreau-Yosida smoothing technique is a notable departure from the usual approach. Smoothing techniques are commonly invoked when solving nonconvex-concave problems to achieve better iteration complexity [62, 40, 59]. However, we target at smoothing both the primal and dual variables with different magnitudes to ensure global convergence.

## 3 Convergence Analysis

The convergence result of the proposed DS-GDA (i.e., Algorithm 1) will be discussed in this section. To illustrate the main result, we first provide the stationarity measure in Definition 1.

**Definition 1 (Stationarity measure)** *The point $(\hat{x}, \hat{y}) \in \mathcal{X} \times \mathcal{Y}$ is said to be an*
(i) *$\epsilon$-game stationary point (GS) if*

$$\text{dist}(\mathbf{0}, \nabla_x f(\hat{x}, \hat{y}) + \partial \mathbf{1}_{\mathcal{X}}(\hat{x})) \leq \epsilon \quad and \quad \text{dist}(\mathbf{0}, -\nabla_y f(\hat{x}, \hat{y}) + \partial \mathbf{1}_{\mathcal{Y}}(\hat{y})) \leq \epsilon;$$

(ii) *$\epsilon$-optimization stationary point (OS) if*

$$\left\| \text{prox}_{\max_{y \in \mathcal{Y}} f(\cdot, \hat{y}) + \mathbf{1}_{\mathcal{X}}}(\hat{x}) - \hat{x} \right\| \leq \epsilon.$$

**Remark 1** *The definition of game stationary point is a natural extension of the first-order stationary point in minimization problems. It is a necessary condition for local minimax point [34] and has been widely used in nonconvex-nonconcave optimization [21, 37]. We have investigated their relationships in Appendix K.*

## 3.1 Complexity under Nonconvex-(Non)concave Setting

Inspired by [62, 40], we consider a novel Lyapunov function $\Phi : \mathbb{R}^n \times \mathbb{R}^d \times \mathbb{R}^n \times \mathbb{R}^d \to \mathbb{R}$ defined as follows:

$$\Phi(x,y,z,v) := \underbrace{F(x,y,z,v) - d(y,z,v)}_{\text{Primal descent}} + \underbrace{p(z,v) - d(y,z,v)}_{\text{Dual ascent}} + \underbrace{q(z) - p(z,v)}_{\text{Proximal ascent}} + \underbrace{q(z) - \underline{F}}_{\text{Proximal descent}} + \underline{F}$$

$$= F(x,y,z,v) - 2d(y,z,v) + 2q(z),$$

where $d(y,z,v) := \min_{x \in \mathcal{X}} F(x,y,z,v)$, $p(z,v) := \max_{y \in \mathcal{Y}} d(y,z,v)$, $q(z) := \max_{v \in \mathbb{R}^d} p(z,v)$, and $\underline{F} := \min_{z \in \mathbb{R}^n} q(z)$. Obviously, this Lyapunov function is lower bounded, that is, $\Phi \geq \underline{F}$. To gain a better understanding of the rationale behind the construction of $\Phi$, it is observed that the Lyapunov function has a strong connection to the updates of the iterations. The primal update corresponds to "primal descent" and gradient ascent in dual variable corresponds to the "dual ascent". The averaging updates of proximal variables could be understood as an approximate gradient descent of $p(z,v)$ and an approximate gradient ascent of $g(v)$, resulting in the "proximal descent" and "proximal ascent" terms in the Lyapunov function. Compared with that in [62, 40], we have an additional "proximal ascent" term. It is introduced by the regularized term for dual variable in $F$ and the update of proximal variable $v$. Essentially, the "nonconcavity" of $f(x, \cdot)$ brings the additional term. With this Lyapunov function, we can establish the following basic descent property as our first important result.

**Proposition 1 (Basic descent estimate)** *Suppose that Assumption 1 holds and $r_1 \geq 2L$, $r_2 \geq 2\lambda L$ with the parameters*

$$0 < c \leq \min\left\{\frac{4}{3(L+r_1)}, \frac{1}{6\lambda L}\right\}, 0 < \alpha \leq \min\left\{\frac{2}{3\lambda L \sigma^2}, \frac{1}{6L_d}, \frac{1}{5\lambda\sqrt{\lambda+5}L}\right\},$$

$$0 < \beta \leq \min\left\{\frac{24r_1}{360r_1 + 5r_1^2\lambda + (2\lambda L + 5r_1)^2}, \frac{\alpha\lambda^2 L^2}{384r_1(\lambda+5)(\lambda+1)^2}\right\},$$

$$0 < \mu \leq \min\left\{\frac{2(\lambda+5)}{2(\lambda+5)+\lambda^2 L^2}, \frac{\alpha\lambda^2 L^2}{64r_2(\lambda+5)}\right\}.$$

*Then for any $t \geq 0$,*

$$\Phi^t - \Phi^{t+1} \geq \frac{r_1}{32}\|x^{t+1} - x^t\|^2 + \frac{r_2}{15}\|y^t - y_+^t(z^t,v^t)\|^2 + \frac{r_1}{5\beta}\|z^t - z^{t+1}\|^2 + \frac{r_2}{4\mu}\|v_+^t(z^{t+1}) - v^t\|^2$$
$$- 4r_1\beta\|x(z^{t+1}, v(z^{t+1})) - x(z^{t+1}, v_+^t(z^{t+1}))\|^2. \tag{1}$$

*where $\sigma := \frac{2cr_1+1}{c(r_1-L)}$ and $L_d := \left(\frac{\lambda L}{r_1 - L} + 2\right)\lambda L + r_2$. Moreover, we have $y_+(z,v) := \text{proj}_{\mathcal{Y}}(y + \alpha\nabla_y F(x(y,z,v),y,z,v))$ and $v_+(z) := v + \mu(y(x(z,v),z,v) - v)$ with the following definitions:* (i) $x(y,z,v) := \text{argmin}_{x \in \mathcal{X}} F(x,y,z,v)$, (ii) $y(x,z,v) := \text{argmax}_{y \in \mathcal{Y}} F(x,y,z,v)$, (iii) $x(z,v) := \text{argmin}_{x \in \mathcal{X}} \max_{y \in \mathcal{Y}} F(x,y,z,v)$, (iv) $v(z) := \text{argmax}_{v \in \mathbb{R}^d} P(z,v)$.

The lower bound of $\Phi$ by $\underline{F}$ is established by its construction, so the crux of proving subsequence convergence is to establish the decreasing property of the Lyapunov function. Although Proposition 1 quantifies the variation of the Lyapunov function values between two consecutive iterates, there is a negative error term $\|x(z^{t+1}, v(z^{t+1})) - x(z^{t+1}, v_+^t(z^{t+1}))\|$ that makes the decreasing property of $\Phi$ unclear.

Next, we characterize the negative error term in terms of other positive terms and then exhibit the sufficient descent property by bounding the coefficients. Conceptually, the error term is related to $\|v_+^t(z^{t+1}) - v(z^{t+1})\|$ by the Lipschitz property of the solution mapping $x(z, \cdot)$. However, $\|v_+^t(z^{t+1}) - v(z^{t+1})\|$ may not be a suitable surrogate since it includes the information about the optimal solution $v(z^{t+1})$. Fortunately, with the help of the global KŁ property or concavity for the dual function (see Assumption 2 and 3), we can bound the negative error term by $\|v_+^t(z^{t+1}) - v(z^{t+1})\|$ (called the proximal error bound). The explicit form of this bound is provided in the following Propositions 2.

**Assumption 2 (KŁ property with exponent $\theta$ of the dual function)** *For any fixed point $x \in \mathcal{X}$, the problem $\max_{y \in \mathcal{Y}} f(x,y)$ has a nonempty solution set and a finite optimal value. Moreover, there*

*exist $\tau > 0$, $\theta \in (0, 1)$ such that for any $x \in \mathcal{X}, y \in \mathcal{Y}$*

$$\left( \max_{y' \in \mathcal{Y}} f(x, y') - f(x, y) \right)^{\theta} \leq \frac{1}{\tau} \operatorname{dist}(\mathbf{0}, -\nabla_y f(x, y) + \partial \mathbf{1}_{\mathcal{Y}}(y)).$$

**Assumption 3 (Concavity of the dual function)** *For any fixed point $x \in \mathcal{X}$, $f(x, \cdot)$ is concave.*

**Proposition 2 (Proximal error bound)** *Under the setting of Proposition 1 with Assumption 2 or 3, for any $z \in \mathbb{R}^n, v \in \mathbb{R}^d$ one has*
(i) KŁ exponent $\theta \in (0, 1)$:

$$\|x(z^{t+1}, v_+^t(z^{t+1})) - x(z^{t+1}, v(z^{t+1}))\|^2 \leq \omega_0 \|v_+^t(z^{t+1}) - v^t\|^{\frac{1}{\theta}};$$

(ii) Concave:

$$\|x(z^{t+1}, v_+^t(z^{t+1})) - x(z^{t+1}, v(z^{t+1}))\|^2 \leq \omega_1 \|v_+^t(z^{t+1}) - v^t\|,$$

*where $\omega_0 := \frac{2}{(r_1 - L)\tau} \left( \frac{r_2(1-\mu)}{\mu} + \frac{r_2^2}{r_2 - \lambda L} \right)^{\frac{1}{\theta}}$ and $\omega_1 := \frac{4r_2 \operatorname{diam}(\mathcal{Y})}{r_1 - L} \left( \frac{1-\mu}{\mu} + \frac{r_2}{r_2 - \lambda L} \right)$. Here, $\operatorname{diam}(\mathcal{Y})$ denotes the diameter of the set $\mathcal{Y}$.*

Armed with Proposition 1 and Proposition 2, we establish the main theorem concerning the iteration complexity of DS-GDA with respect to the above-mentioned standard stationarity measure for (P).

**Theorem 1 (Iteration complexity for nonconvex-(non)concave problems)** *Under the setting of Theorem 1 and Proposition 2, for any $T > 0$, there exists a $t \in \{1, 2, \ldots, T\}$ such that*
(i) KŁ exponent $\theta \in (\frac{1}{2}, 1)$: $(x^{t+1}, y^{t+1})$ is an $\mathcal{O}(T^{-\frac{1}{4\theta}})$-GS and $z^{t+1}$ is an $\mathcal{O}(T^{-\frac{1}{4\theta}})$-OS if $\beta \leq \mathcal{O}(T^{-\frac{2\theta-1}{2\theta}})$;
(ii) KŁ exponent $\theta \in (0, \frac{1}{2}]$: $(x^{t+1}, y^{t+1})$ is an $\mathcal{O}(T^{-\frac{1}{2}})$-GS and $z^{t+1}$ is an $\mathcal{O}(T^{-\frac{1}{2}})$-OS if $\beta \leq \frac{r_2}{32 r_1 \mu \omega_0 (2 \operatorname{diam}(\mathcal{Y}))^{\frac{1}{\theta}-2}}$;
(iii) Concave: $(x^{t+1}, y^{t+1})$ is an $\mathcal{O}(T^{-\frac{1}{4}})$-GS and $z^{t+1}$ is an $\mathcal{O}(T^{-\frac{1}{4}})$-OS if $\beta \leq \mathcal{O}(T^{-\frac{1}{2}})$.

Moreover, when the problem (P) equipped with the widely fulfilled semi-algebraic structure [9, 4, 15] and the dual function satisfies the KŁ property with $\theta \in (0, \frac{1}{2}]$, we additionally have the following sequential convergence result of the DS-GDA.

**Theorem 2 (Last-iterate convergence of DS-GDA)** *Consider the setting of Theorem 1 and suppose that Assumption 2 holds with $\theta \in (0, \frac{1}{2}]$. Suppose that $f(\cdot, y)$ is semi-algebraic and $\mathcal{X}, \mathcal{Y}$ are semi-algebraic sets. Then, the sequence $\{(x^t, y^t, z^t, v^t)\}$ converges to $(x^*, y^*, z^*, v^*)$, where $(x^*, y^*)$ is a GS and $z^*$ is an OS.*

### 3.2 Universal Results

For convex-nonconcave or nonconvex-nonconcave minimax problems in which the primal function satisfies the KŁ property, analogous results to Theorem 1 can be established. This can be accomplished by introducing an alternative Lyapunov function $\Psi : \mathbb{R}^n \times \mathbb{R}^d \times \mathbb{R}^n \times \mathbb{R}^d \to \mathbb{R}$ as follows:

$$\Psi(x, y, z, v) := \underbrace{h(x, z, v) - F(x, y, z, v)}_{\text{Dual ascent}} + \underbrace{h(x, z, v) - p(z, v)}_{\text{Primal descent}} + \underbrace{p(z, v) - g(v)}_{\text{Proximal descent}} + \underbrace{\overline{F} - g(v)}_{\text{Proximal ascent}},$$

where $h(x, z, v) := \max_{y \in \mathcal{Y}} F(x, y, z, v)$, $g(v) := \min_{z \in \mathbb{R}^n} p(z, v)$, and $\overline{F} := \max_{v \in \mathbb{R}^d} g(v)$.

It is worth noting that our Lyapunov function exhibits symmetry with respect to nonconvex-(non)concave problems, since the adjustment only entails interchanging the position between $(d(y, z, v), q(z))$ and $(h(x, z, v), g(v))$. Therefore, similar convergence results as Theorem 1 could be derived without any effort. A more detailed proof can be found in Appendix H.

Based on these results, we are ready to show that our DS-GDA is a universal algorithm. By incorporating the choices of parameters, we can identify a consistent set of parameters that ensures the convergence of DS-GDA in nonconvex-nonconcave, nonconvex-concave, and convex-nonconcave problems. The universal convergence rate is stated as follows:

**Theorem 3 (Universal convergence of DS-GDA)** *Without loss of generality, we consider the case where $\lambda = 1$, implying $L_x = L_y = L$. For convex-nonconcave, nonconvex-concave, and nonconvex-nonconcave minimax problems that satisfy the one-sided KŁ property, if we further set $r_1 = r_2 = r \geq 20L$, DS-GDA converges provided certain parameter conditions are met:*

$$\frac{-(r-L)\sqrt{L^2 - 14Lr + r^2} + L^2 - 8Lr + r^2}{12Lr^2} \leq c = \alpha \leq \frac{1}{6L_d},$$

$$0 < \beta = \mu \leq \min\left\{\frac{24r}{360r + 5r^2 + (2L+5r)^2}, \frac{cL^2}{9216r}, \frac{12}{12+L^2}\right\} \leq \mathcal{O}(T^{-\frac{1}{2}}).$$

*When KŁ exponent $\theta \in (0, \frac{1}{2}]$, we further require*

$$\beta = \mu \leq \frac{1}{4\sqrt{2\max\left\{\omega_0(2\operatorname{diam}(\mathcal{Y}))^{\frac{1}{\theta}-2}, \omega_2(2\operatorname{diam}(\mathcal{X}))^{\frac{1}{\theta}-2}\right\}}},$$

*where $\omega_0$ and $\omega_2$ are the coefficients in Propositions 2 and 6. Then, $(x^{t+1}, y^{t+1})$ is an $\mathcal{O}(T^{-\frac{1}{4}})$-GS.*

**Remark 2** *It is worth noting that our results are more general compared to AGP in [57], where a unified analysis for convex-nonconcave and nonconvex-concave problems is provided. Here, our algorithm can also be applied to the nonconvex-nonconcave problem with one-sided KŁ property. Moreover, our algorithm is universal, meaning that one single set of parameters can ensure convergence across all these scenarios. By contrast, different choices of parameters are required to guarantee optimal convergence in AGP.*

## 4 Empirical Validation of DS-GDA

### 4.1 Universality of DS-GDA

To validate the universality of DS-GDA, we commence by providing a graphical description of feasible regions for the choice of parameters. Subsequently, employing a set of symmetric parameters, we will show that the KŁ-nonconvex problems can converge to the desired stationary point, which supports our universality results.

Without loss of generality, we consider the case where $\lambda = 1$. In the pursuit of symmetric parameter selection, we initially fix $\beta = \mu = 1/5000$, thus reducing the problem to determining only two remaining parameters: $c$ and $r$. We then explore their relationships by setting $r_1 = r_2 = t_2 L$ and $c = \alpha = 1/(t_1 r)$. Restricting $0 \leq t_1, t_2 \leq 100$, the feasible choices of $t_1$ and $t_2$ are selected to guarantee the first four coefficients in basic descent estimate (1) are positive. As visually depicted in Figure 1a, it becomes apparent that a large number of choices for $r$ and $c$ are available to ensure convergence.

Next, we test our algorithm on a nonconvex-nonconcave problem that satisfies the one-sided KŁ property. With a set of symmetric parameters, we find that DS-GDA can easily converge to the desired stationary point. We first introduce the KŁ-nonconcave problems as follows:

**KŁ-Nonconcave Example** The following example satisfies two-sided KŁ property with exponent $\frac{1}{2}$, which is studied in [58]:

$$\min_{x \in \mathcal{X}} \max_{y \in \mathcal{Y}} x^2 + 3\sin(x)^2 \sin(y)^2 - 4y^2 - 10\sin(y)^2, \tag{2}$$

where $\mathcal{X} = \mathcal{Y} = \{z : -1 \leq z \leq 1\}$. The only saddle point is $u^* = [x^*; y^*] = [0; 0]$.

From Figure 1b, we can observe that symmetric parameter selection is effective for addressing KŁ-nonconcave problems. To be specific, by setting $r_1 = r_2 = 0.125$, $c = \alpha = 0.04$, $\beta = \mu = 0.8$, DS-GDA directly converges to the saddle point, which validates the universal results in Theorem 3.

### 4.2 Robustness of DS-GDA

In this section, we compare the convergence performance of DS-GDA with the closely related algorithm S-GDA on some illustrative examples. Additionally, we report the range of parameters for these two algorithms to demonstrate the robustness of DS-GDA. To begin, we present some simple polynomial examples.

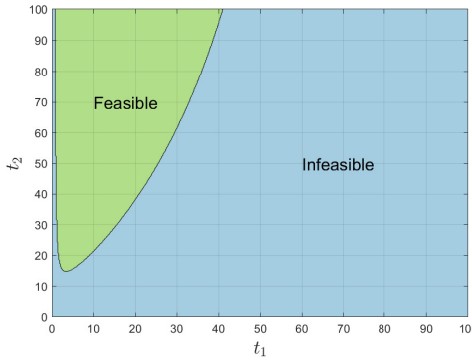
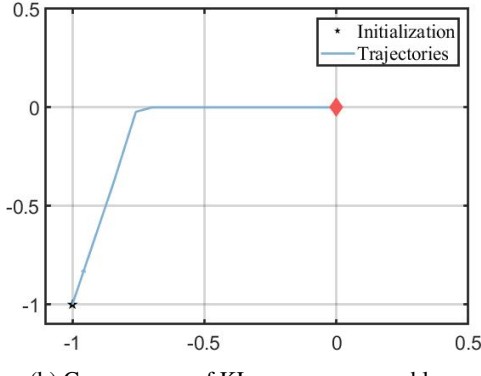

(a) Feasible region of different $t_1$ and $t_2$    (b) Convergence of KŁ-nonconcave problem

Figure 1: In Figure (a), the blue region indicates where convergence cannot be guaranteed. The green region indicates a series of parameters that can be chosen to guarantee convergence. Figure (b) demonstrates the effectiveness of symmetric parameter selection for (non)convex-(non)concave problems.

**Convex-Nonconcave Example**    The following example is convex-nonconcave, which is studied in [13, 1]:

$$\min_{x \in \mathcal{X}} \max_{y \in \mathcal{Y}} 2x^2 - y^2 + 4xy + 4y^3/3 - y^4/4,$$

where $\mathcal{X} = \mathcal{Y} = \{z : -1 \le z \le 1\}$ and $u^* = [x^*; y^*] = [0; 0]$ is the only stationary point.

**KŁ-Nonconcave Example**    The KŁ-nonconcave example considered here is mentioned in Section 4.1 (see equation (2)). The $u^* = [x^*; y^*] = [0; 0]$ is a saddle point and the only stationary point.

**Nonconvex-Nonconcave Example**    The nonconvex-nonconcave example considered here is the "Bilinearly-coupled Minimax" example (3) discussed in [29]:

$$\min_{x \in \mathcal{X}} \max_{y \in \mathcal{Y}} f(x) + Axy - f(y), \tag{3}$$

where $f(z) = (z+1)(z-1)(z+3)(z-3)$, $A = 11$, and $\mathcal{X} = \mathcal{Y} = \{z : -4 \le z \le 4\}$. It does not satisfy any existing regularity condition, and the point $u^* = [x^*; y^*] = [0; 0]$ is the only stationary point.

The extrapolation parameters $z$ and $v$ are initialized as $x$ and $y$, respectively. According to Lemma 8, our algorithm terminates when $\|z - v\|$ and $\|y - v\|$ are less than $10^{-6}$ or when the number of iterations exceeds $10^7$. To ensure a fair comparison, we use the same initializations for DS-GDA and S-GDA in each test. The algorithm parameters are tuned so that both DS-GDA and S-GDA are optimal. In other words, they can converge to a stationary point in a minimum number of iterations. We compare the convergence of two algorithms by plotting the iteration gap $\|u^k - u^*\|$ against the number of iterations for each example, where $u^*$ denotes the stationary point. Our results show that DS-GDA and S-GDA have similar convergence performance when the primal function is convex or satisfies the KŁ property, as depicted in Figure 2a and 2b. However, for the nonconvex-nonconcave example where no regularity condition is satisfied, DS-GDA achieves much faster convergence than S-GDA, as shown in Figure 2c.

To demonstrate the robustness of the proposed DS-GDA, we present feasible regions of all common hyperparameters in Figure 3. We tune the lower and upper bounds of each parameter while keeping other parameters fixed at their optimal values. As illustrated in Figures 3a and 3b, for examples that satisfy the one-sided KŁ property or with a convex primal function, the implementable ranges of DS-GDA and S-GDA for common parameters are roughly similar. In addition, the two auxiliary parameters for DS-GDA, i.e., $r_2$ and $\mu$, have relatively wide ranges of values, indicating that they allow for flexibility in their selection. However, for nonconvex-nonconcave problems, DS-GDA exhibits a wider range of viable parameter values compared to S-GDA. This observation highlights the robustness of DS-GDA when it comes to parameter selection (refer to Figure 3c).

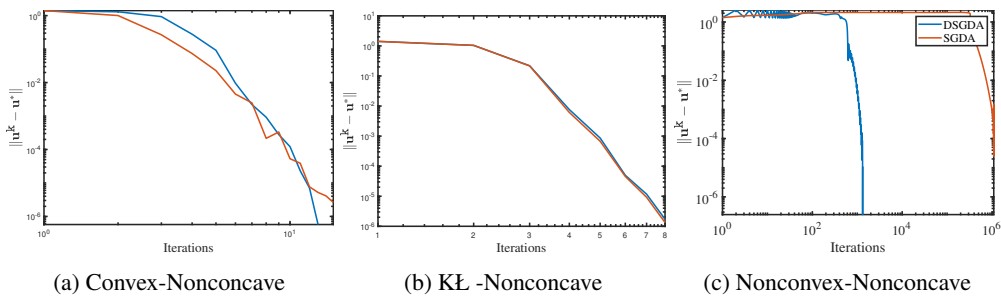

(a) Convex-Nonconcave     (b) KŁ -Nonconcave     (c) Nonconvex-Nonconcave

Figure 2: Convergence performance of different problems.

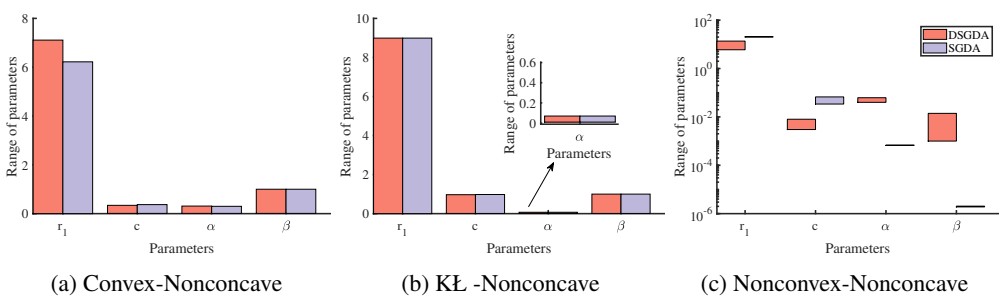

(a) Convex-Nonconcave     (b) KŁ -Nonconcave     (c) Nonconvex-Nonconcave

Figure 3: Range of parameters for different problems.

## 4.3 Effectiveness of Getting Rid of Limit Cycle

In this section, we demonstrate the effectiveness of the proposed DS-GDA on some illustrative examples that are widely used in literature. Notably, they do not satisfy any of the regularity conditions in previous literature (i.e., KŁ condition, weak MVI, and $\alpha$-dominant condition). We refer the readers to Appendix L for details on how to check the failure of these conditions for these examples. In addition to the violation of regularity conditions, it has been verified that none of the existing first-order algorithms can achieve convergence for all four examples. The detailed description of the four examples can be found in Appendix B.

To showcase the convergence behavior and effectiveness of the proposed DS-GDA, we compare it with three other state-of-the-art methods, that is, damped extragradient method (Damped EGM) [30], S-GDA [40, 62], and generalized curvature extragradient method (CurvatureEG+) [50]. Damped EGM is guaranteed to converge under $\alpha$-dominant condition, and S-GDA converges when the dual function is concave. CurvatureEG+ could converge under weak MVI condition, which is the weakest variational inequality-based condition as far as we know in the literature.

For the experiments, we use the same initializations of primal and dual variables for all four methods to ensure a fair comparison. For DS-GDA, the exponentially weighted variables are initialized as the same values of the primal and dual variables, respectively. We stop DS-GDA when the differences between primal-dual variables and the corresponding exponentially weighted variables are all less than $10^{-6}$ or when the number of iterations exceeds $10^5$. For other baseline methods, we terminate them when either the number of iterates reaches $10^5$ or the difference between two consecutive iterates is less than $10^{-6}$.

Figure 4 shows the trajectories of different methods with various initializations for the aforementioned four examples. We observe from the first column that our DS-GDA successfully gets rid of limit cycles in all examples. While S-GDA exhibits similar performance as DS-GDA, it is still not as potent in terms of its overall effectiveness. Specifically, in the case of the "Bilinearly-coupled minimax" example, S-GDA gets trapped in a limit cycle, while our DS-GDA successfully avoids it and achieves convergence. The figure in the second column provides more details on this comparison. With the violation of the of $\alpha$-interaction dominance condition, damped EGM either suffers from the spurious cycling convergence phenomenon or diverges to a point on the boundary (see the third row in the fourth column). It converges only when the initialization is very close to the stationary point (see the fourth row in the last column). Similar results are observed for CurvatureEG+ (see Figures in the third column for details). Thus, the proposed DS-GDA outperforms other methods. It is the only

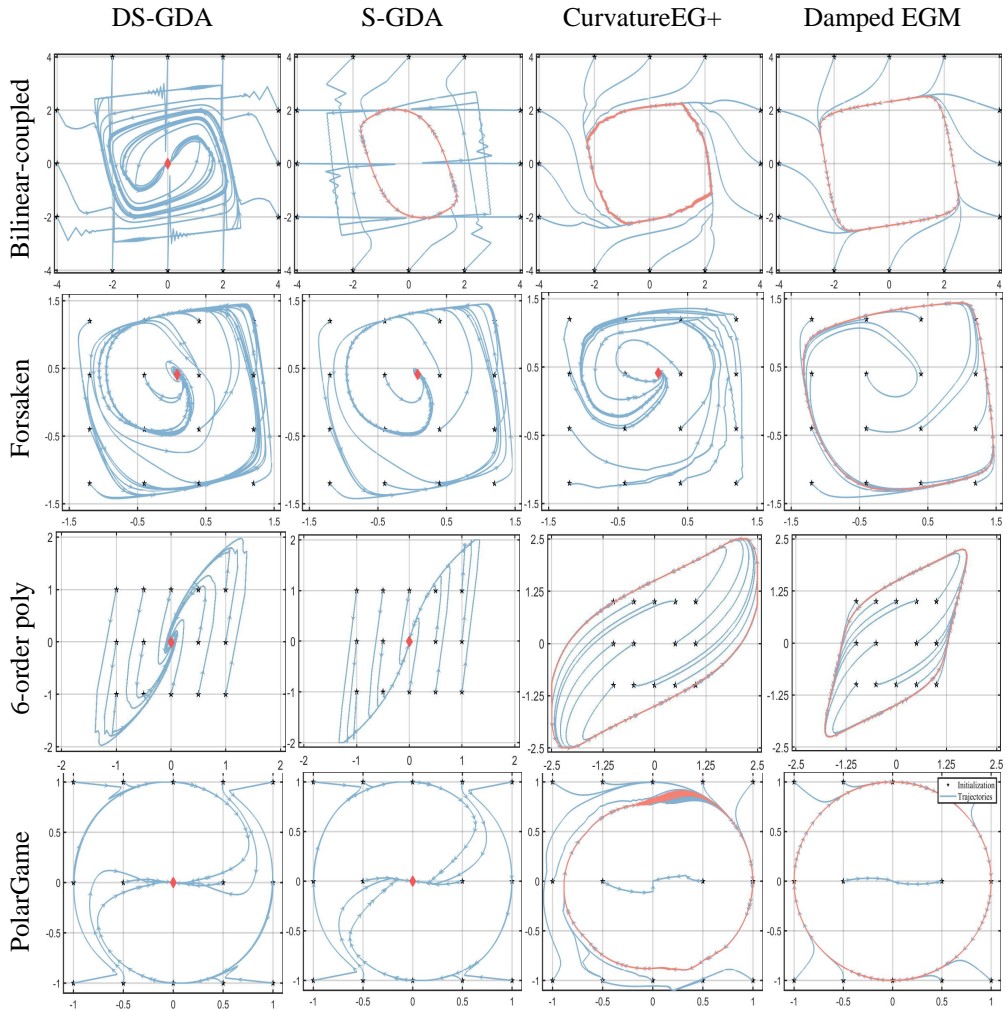

Figure 4: Trajectories of different methods with various initialization for the aforementioned four examples. The initialization of every trajectory is marked as a black star, the blue (resp. red) line represents the path (resp. limit cycle) of the methods, and the red rhombus is the stationary point.

algorithm that can get rid of the limit cycle and enjoy global convergence for all these challenging examples.

## 5   Conclusion

In this paper, we propose a single-loop algorithm called the doubly smoothed gradient descent ascent (DS-GDA) algorithm, which offers a natural balance between primal-dual updates for constrained nonconvex-nonconcave minimax problems. This is the first simple and universal algorithm for nonconvex-concave, convex-nonconcave, and nonconvex-nonconcave problems with one-sided KŁ property. By employing a single set of parameters, DS-GDA achieves convergence with an iteration complexity of $\mathcal{O}(\epsilon^{-4})$ across all these scenarios. Sharper iteration complexity can be obtained when the one-sided KŁ property is satisfied with an exponent $\theta \in (0, 1)$, matching the state-of-the-art results. We further conduct experiments to validate the universality and efficiency of DS-GDA for avoiding the limit cycle phenomenon, which commonly occurs in challenging nonconvex-nonconcave examples. There is still a gap between theory and practice, and it would be intriguing to explore the possibility of achieving global convergence for DS-GDA without any regularity conditions. This opens up new avenues for research on nonconvex-nonconcave minimax problems.

## Acknowledgments and Disclosure of Funding

Anthony Man-Cho So is supported in part by the Hong Kong Research Grants Council (RGC) General Research Fund (GRF) projects CUHK 14203920 and 14216122. Jose Blanchet and Jiajin Li are supported by the Air Force Office of Scientific Research under award number FA9550-20-1-0397 and NSF 1915967, 2118199, 2229012, 2312204.

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

## A Organization of the Appendix

We organize the appendix as follows:

- The related work and the four challenging nonconvex-nonconcave examples are listed in Section B;
- Notations and some useful lemmas such as Lipschitz error bounds are provided in Section C;
- The characterization of changes in the Lyapunov function between successive iterations is established in Section D;
- The proof of Proposition 1 is given in Section E;
- The proof of proximal and dual error bounds are given in Section F;
- The proof of Theorem 1 is given in Section G;
- The convergence proof for nonconvex(convex)-nonconcave problem is provided in Section H.
- The proof of Theorem 2 is provided in Section I;
- The proof of Theorem 3 is provided in Section J;
- The quantitative relationship between different notions of the stationary point is provided in Section K;
- The properties of examples mentioned in Section 4.3 are checked in Section L. We also show that the wrong selection of smoothing sides will result in slow convergence for S-GDA.

## B Related Works

There are three representative types of regularity conditions in the literature to restrict the problem class that algorithms are developed to get rid of the limit cycle.

**Polyak-Łojasiewicz (PŁ) condition** The PŁ condition (4) was originally proposed by [51] and is a crucial tool for establishing linear convergence of first-order algorithms for pure minimization problems [35]. Suppose that the problem $\max_{x \in \mathbb{R}^d} h(x)$ has a nonempty solution set and a finite optimal value. The PŁ condition states that there exist a constant $\mu > 0$ such that for any $x \in \mathbb{R}^d$,

$$\frac{1}{2}\|\nabla h(x)\|^2 \geq \mu \left( h(x) - \min_{x \in \mathbb{R}^d} h(x) \right). \tag{4}$$

There is a host of works trying to invoke the PŁ condition on the dual function $f(x, \cdot)$ [59, 47, 22, 58]. Unfortunately, we would like to point out that this condition is too restrictive and inherently avoid the main difficulty in addressing general nonconvex-nonconcave minimax problems. With PŁ condition imposed on the dual function, the inner maximization value function $\phi(\cdot) = \max_{y \in \mathcal{Y}} f(\cdot, y)$ is $L$-smooth [47, Lemma A.5]. Thus, the dual update can naturally be controlled by the primal since we can regard minimax problems as pure smooth (weakly convex) minimization problems over $x$. However, for general cases, the inner value function $\phi$ may not even be Lipschitz. Recently, [47] proposed a so-called multi-step GDA method with an iteration complexity of $\mathcal{O}(\log(\epsilon^{-1})\epsilon^{-2})$. [22] further developed the single-loop two-timescale GDA method to better take the computational tractability into account and the complexity is improved to $\mathcal{O}(\epsilon^{-2})$. Following the smoothing (extrapolation) technique developed in [62], [59] extended the proposed smoothed GDA to the stochastic setting and obtained an iteration complexity of $\mathcal{O}(\epsilon^{-4})$.

**Varitional Inequality (VI)** Variational inequalities can be regarded as generalizations of minimax optimization problems [20]. In convex-concave minimax optimization, finding a saddle point is equivalent to solving the Stampacchia Variational Inequality (SVI):

$$\langle G(u^\star), u - u^\star \rangle \geq 0, \quad \forall u \in \mathcal{U}. \tag{5}$$

Here $u := [x; y]$, $u^\star$ is the optimal solution, and the operator $G$ is a gradient operator: $G(u) := [\nabla_x f(x, y); -\nabla_y f(x, y)]$ with $\mathcal{U} = \mathcal{X} \times \mathcal{Y}$. The solution of (5) is referred to as a strong solution of

the VI corresponding to $G$ and $\mathcal{U}$ [31]. For the nonconvex-nonconcave minimax problem, without the monotonicity of $G$, the solution of SVI may not even exist. One alternative condition is to assume the existence of solutions $u^\star$ for the Minty Variational Inequality (MVI):

$$\langle G(u), u - u^\star \rangle \geq 0, \quad \forall u \in \mathcal{U}. \tag{6}$$

The solution of (6) is called a weak solution of the VI [24]. In the setting where $G$ is continuous and monotone, the solution sets of (5) and (6) are equivalent. However, these two solution sets are different in general and a weak solution may not exist when a strong solution exists. Many works have established the convergence results under the MVI condition or its variants [21, 28, 45, 43, 42, 17, 55, 8, 23]. Although MVI leads to convergence, it is hard to check in practice and is inapplicable to many functions (see examples in Section 4.3). A natural question is: *Can we further relax the MVI condition to ensure convergence*? One possible way is to relax the nonnegative lower bound to a negative one [33, 37, 11, 12, 21], i.e., the so-called weak MVI condition:

$$\langle G(u), u - u^\star \rangle \geq -\frac{\rho}{2}\|G(u)\|^2, \quad \forall u \in \mathcal{U}.$$

Here, we restricted $\rho \in [0, \frac{1}{4L})$ and [21] proposed a Generalized extragradient method (Generalized EGM) with $\mathcal{O}(\epsilon^{-2})$ iteration complexity. To include a wider function class, [50] enlarged the range of $\rho$ to $[0, \frac{1}{L})$ and $\rho$ can be larger if more curvature information of $f$ is involved. However, for general smooth nonconvex-nonconcave problems, various VI conditions are hard to check and $\rho$ would easily violate the constraints. In this case, the proposed CurvatureEG+ still suffers from the limit cycle issue.

$\alpha$**-interaction dominant condition**     Another line of work is to impose the $\alpha$-interaction dominant conditions (7a), (7b) on $f$, i.e.,

$$\nabla_{xx}^2 f(x,y) + \nabla_{xy}^2 f(x,y)(\eta \boldsymbol{I} - \nabla_{yy}^2 f(x,y))^{-1}\nabla_{yx}^2 f(x,y) \succeq \alpha \boldsymbol{I}, \tag{7a}$$

$$-\nabla_{yy}^2 f(x,y) + \nabla_{yx}^2 f(x,y)(\eta \boldsymbol{I} + \nabla_{xx}^2 f(x,y))^{-1}\nabla_{xy}^2 f(x,y) \succeq \alpha \boldsymbol{I}. \tag{7b}$$

Intuitively, this condition is to characterize how the interaction part of $f$ affects the landscape of saddle envelope $f_\eta(x,y) = \min_{z\in\mathcal{X}} \max_{v\in\mathcal{Y}} f(z,v) + \frac{\eta}{2}\|x-z\|^2 - \frac{\eta}{2}\|y-v\|^2$ [5]. We say $\alpha$ is in the interaction dominant regime if $\alpha$ in (7a), (7b) is a sufficiently large positive number and in the interaction weak regime when $\alpha$ is a small but nonzero positive number. Convergence results for the damped proximal point method (Damped PPM) can only be obtained for these two regimes [29]. Otherwise, the method may fall into the limit cycle or even diverge. Unfortunately, conditions (7a) and (7b) only hold with $\alpha = -L < 0$ for general $L$-smooth nonconvex-nonconcave function, which will dramatically restrict the problem class. Moreover, second-order information of $f$ is required. For instance, if we choose Exponential Linear Units (ELU) with $a = 1$ [14] as the activation function in neural networks, $f$ is $L$-smooth but not twice differentiable. [29] studied the convergence of Damped PPM and showed that in the interaction dominant regime, their method converges with only one-sided dominance. In the interaction weak regime, their method also enjoys a local convergence rate of $\mathcal{O}(\log(\epsilon^{-1}))$. Taking computational efficiency into consideration, [30] developed the Damped EGM, which has an iteration complexity of $\mathcal{O}(\log(\epsilon^{-1}))$ under two-sided dominance conditions.

There are four representative challenging nonconvex-nonconcave examples in the literature, which have been mentioned in Section 4.3.

**"Bilinearly-coupled minimax" example**     The first one is the "Bilinearly-coupled Minimax" example (3) with $A = 10$. It is a representative example to showcase the limit cycle phenomenon as it breaks the $\alpha$-dominant condition. When the bilinear intersection term between primal and dual variables, i.e., $x$ and $y$, is moderate, it becomes uncertain which variable, either the primal or dual, holds dominance over the other. As a result, this particular example poses a great challenge in terms of ensuring convergence.

**"Forsaken" example**     The second one is the "Forsaken" example considered in [32, Example 5.2], i.e.,

$$\min_{x\in\mathcal{X}} \max_{y\in\mathcal{Y}} x(y - 0.45) + \phi(x) - \phi(y), \tag{8}$$

where $\phi(z) = \frac{1}{4}z^2 - \frac{1}{2}z^4 + \frac{1}{6}z^6$ and $\mathcal{X} = \mathcal{Y} = \{z : -1.5 \leq z \leq 1.5\}$. This example serves as a representative case, highlighting the limitations of min-max optimization algorithms. It demonstrates

| Optimization problems | Function values | Optimal solutions |
|---|---|---|
| $\displaystyle\min_{x\in\mathcal{X}} F(x,y,z,v)$ | $d(y,z,v)$ | $x(y,z,v)$ |
| $\displaystyle\max_{y\in\mathcal{Y}} F(x,y,z,v)$ | $h(x,z,v)$ | $y(x,z,v)$ |
| $\displaystyle\min_{x\in\mathcal{X}}\max_{y\in\mathcal{Y}} F(x,y,z,v)$ | $p(z,v)$ | |
| $\displaystyle\min_{x\in\mathcal{X}} h(x,z,v)$ | $p(z,v)$ | $x(z,v)=x(y(z,v),z,v)$ |
| $\displaystyle\max_{y\in\mathcal{Y}} d(y,z,v)$ | $p(z,v)$ | $y(z,v)=y(x(z,v),z,v)$ |
| $\displaystyle\min_{z\in\mathbb{R}^n} p(z,v)$ | $g(v)$ | $z(v)$ |
| $\displaystyle\max_{v\in\mathbb{R}^d} p(z,v)$ | $q(z)$ | $v(z)$ |

Table 1: Notation

situations where standard algorithms fail to converge to the desired critical points but instead converge to spurious, non-critical points. Specifically, for problem (8), two spurious limit cycles exist across the entire domain. Worse, the one closer to the optimal solution $[x^\star; y^\star] \simeq [0.08; 0.4]$ is unstable, which potentially pushes the trajectories to fall into the recurrent orbit.

**"Sixth-order polynomial" example**    The third one is a sixth-order polynomial scaled by an exponential function. It is studied in [18, 56, 13], i.e.,

$$\min_{x\in\mathcal{X}}\max_{y\in\mathcal{Y}}(4x^2 - (y - 3x + 0.05x^3)^2 - 0.1y^4)\exp(-0.01(x^2 + y^2)),$$

where $\mathcal{X} = \mathcal{Y} = \{z : -2 \leq z \leq 2\}$. It is shown in [56] that existing first-order methods will suffer from limit cycles around $[x^*; y^*] = [0; 0]$. As far as we know, all existing convergence results rely on second-order information. Therefore, it is intriguing to investigate the potential for ensuring convergence using first-order methods.

**"PolarGame" example**    The last example is constructed in [50, Example 3]:

$$G(u) = [\nabla_x f(x,y); -\nabla_y f(x,y)] = [\Phi(x,y) - y, \Phi(y,x) + x],$$

where $u = [x; y]$ satisfy $u \in \mathcal{X} \times \mathcal{Y}$ with $\mathcal{X} = \mathcal{Y} = \{z : -1 \leq z \leq 1\}$ and $\Phi(x,y) = x(-1 + x^2 + y^2)(-9 + 16x^2 + 16y^2)$. It has been verified that there exist limit cycles at $\|u\| = 1$ [2] and $\|u\| = \frac{3}{4}$, which definitely make the iterates hard to converge to $[x^*; y^*] = [0; 0]$. To further demonstrate the effectiveness of our DS-GDA, we intentionally initialize the algorithm on the limit cycle $\|u\| = 1$.

## C  Notation and Useful Lemmas

We first list some useful notations in Table 1. In the following parts, some technical lemmas are presented. Recall that $r_1 > L_x$ and $r_2 > L_y$.

**Lemma 1** *For any $x, x' \in \mathcal{X}$, $y, y' \in \mathcal{Y}$, $z \in \mathbb{R}^n$ and $v \in \mathbb{R}^d$, it follows that*

$$\frac{r_1 - L_x}{2}\|x - x'\|^2 \leq F(x',y,z,v) - F(x,y,z,v) - \langle\nabla_x F(x,y,z,v), x' - x\rangle \leq \frac{L_x + r_1}{2}\|x - x'\|^2,$$

$$-\frac{L_y + r_2}{2}\|y - y'\|^2 \leq F(x,y',z,v) - F(x,y,z,v) - \langle\nabla_y F(x,y,z,v), y' - y\rangle \leq \frac{L_y - r_2}{2}\|y - y'\|^2.$$

**Proof**    Since $f$ is $L$-smooth (from the Assumption 1), we have

$$-\frac{L_x}{2}\|x - x'\|^2 \leq f(x',y) - f(x,y) - \langle\nabla_x f(x,y), x' - x\rangle \leq \frac{L_x}{2}\|x - x'\|^2,$$

$$-\frac{L_y}{2}\|y - y'\|^2 \leq f(x,y') - f(x,y) - \langle\nabla_y f(x,y), y' - y\rangle \leq \frac{L_y}{2}\|y - y'\|^2. \tag{9}$$

---

[2]$\|\cdot\|$ represents the $\ell_2$-norm.

On the other hand, we know that

$$F(x', y, z, v) - F(x, y, z, v) - \langle \nabla_x F(x, y, z, v), x' - x \rangle$$

$$= f(x', y) - f(x, y) - \langle \nabla_x f(x, y) + r_1(x - z), x' - x \rangle + \frac{r_1}{2} \|x' - z\|^2 - \frac{r_1}{2} \|x - z\|^2 \quad (10)$$

$$= f(x', y) - f(x, y) - \langle \nabla_x f(x, y), x' - x \rangle + \frac{r_1}{2} \|x' - x\|^2$$

and similarly

$$F(x, y', z, v) - F(x, y, z, v) - \langle \nabla_y F(x, y, z, v), y' - y \rangle$$

$$= f(x, y') - f(x, y) - \langle \nabla_y f(x, y) - r_2(y - v), y' - y \rangle - \frac{r_2}{2} \|y' - v\|^2 + \frac{r_2}{2} \|y - v\|^2 \quad (11)$$

$$= f(x, y') - f(x, y) - \langle \nabla_y f(x, y), y' - y \rangle - \frac{r_2}{2} \|y' - y\|^2.$$

Combing (9), (10) and (11), we directly obtain the desired results. $\qquad\square$

**Lemma 2 (Lipschitz type error bound conditions)** *Suppose that* $r_2 > (\frac{L_y}{r_1 - L_x} + 2)L_y$, *then for any* $x, x' \in \mathcal{X}$, $y, y' \in \mathcal{Y}$, $z, z' \in \mathbb{R}^n$ *and* $v, v' \in \mathbb{R}^d$. *Then the following inequalities hold:*

    (i) $\|x(y', z, v) - x(y, z, v)\| \le \sigma_1 \|y' - y\|$,

    (ii) $\|x(y, z', v) - x(y, z, v)\| \le \sigma_2 \|z - z'\|$,

    (iii) $\|x(z', v) - x(z, v)\| \le \sigma_2 \|z - z'\|$,

    (iv) $\|y(z, v) - y(z', v)\| \le \sigma_3 \|z - z'\|$,

    (v) $\|y(x, z, v) - y(x', z, v)\| \le \sigma_4 \|x - x'\|$,

    (vi) $\|y(x, z, v) - y(x, z, v')\| \le \sigma_5 \|v - v'\|$,

    (vii) $\|y(z, v) - y(z, v')\| \le \sigma_5 \|v - v'\|$,

*where* $\sigma_1 = \frac{L_y + r_1 - L_x}{r_1 - L_x}$, $\sigma_2 = \frac{r_1}{r_1 - L_x}$, $\sigma_3 = \frac{r_1 \sigma_1}{r_2 - L_y} + \frac{\sigma_2}{\sigma_1}$, $\sigma_4 = \frac{L_x + r_2 - L_y}{r_2 - L_y}$, *and* $\sigma_5 = \frac{r_2}{r_2 - L_y}$.

**Proof** (i) From Lemma 1, we know that

$$F(x(y, z, v), y', z, v) - F(x(y', z, v), y', z, v) \ge \frac{r_1 - L_x}{2} \|x(y, z, v) - x(y', z, v)\|^2,$$

$$F(x(y, z, v), y', z, v) - F(x(y, z, v), y, z, v) \le \langle \nabla_y F(x(y, z, v), y, z, v), y' - y \rangle + \frac{L_y - r_2}{2} \|y - y'\|^2,$$

$$F(x(y', z, v), y, z, v) - F(x(y', z, v), y', z, v) \le \langle \nabla_y F(x(y', z, v), y, z, v), y - y' \rangle + \frac{L_y + r_2}{2} \|y - y'\|^2,$$

$$F(x(y, z, v), y, z, v) - F(x(y', z, v), y, z, v) \le \frac{L_x - r_1}{2} \|x(y, z, v) - x(y', z, v)\|^2.$$

Combining above inequalities, one has that

$$(r_1 - L_x)\|x(y, z, v) - x(y', z, v)\|^2$$

$$\le \langle \nabla_y F(x(y, z, v), y, z, v) - \nabla_y F(x(y', z, v), y, z, v), y' - y \rangle + L_y \|y - y'\|^2$$

$$\le L_y \|x(y', z, v) - x(y, z, v)\| \|y' - y\| + L_y \|y - y'\|^2,$$

where the second inequality is from Cauchy-Schwarz inequality and $L$-smooth property. Let $\xi := \|x(y', z, v) - x(y, z, v)\| / \|y - y'\|$. Then it follows that

$$\xi^2 \le \frac{L_y}{r_1 - L_x} + \frac{L_y}{r_1 - L_x} \xi.$$

Consequently, utilizing AM-GM inequality we derive $\xi \le \frac{\sqrt{L_y^2 + 2r_1 L_y - 2L_x L_y}}{r_1 - L_x} \le \frac{L_y + r_1 - L_x}{r_1 - L_x} := \sigma_1$.

(ii-iii) Again from Lemma 1, we know that

$$F(x(y, z, v), y, z', v) - F(x(y, z', v), y, z', v) \ge \frac{r_1 - L_x}{2} \|x(y, z, v) - x(y, z', v)\|^2,$$

$$\quad (12)$$

$$F(x(y, z, v), y, z, v) - F(x(y, z', v), y, z, v) \le \frac{L_x - r_1}{2} \|x(y, z, v) - x(y, z', v)\|^2,$$

From the definition of $F$, we know that

$$
\begin{aligned}
F(x(y,z,v),y,z',v) - F(x(y,z,v),y,z,v) &= \frac{r_1}{2}\langle z' + z - 2x(y,z,v), z' - z\rangle, \\
F(x(y,z',v),y,z,v) - F(x(y,z',v),y,z',v) &= \frac{r_1}{2}\langle z + z' - 2x(y,z',v), z - z'\rangle.
\end{aligned}
\tag{13}
$$

Incorporating (12), (13) and using the Cauchy-Schwarz inequality, we have

$$
\begin{aligned}
(r_1 - L_x)\|x(y,z,v) - x(y,z',v)\|^2 &\le r_1\langle x(y,z',v) - x(y,z,v), z' - z\rangle \\
&\le r_1\|x(y,z',v) - x(y,z,v)\|\|z' - z\|,
\end{aligned}
$$

which completes the proof of (ii). Moreover, since $\max_{y\in\mathcal{Y}} F(\cdot, y, \cdot, \cdot)$ is $(r_1 - L_x)$-strongly convex in $x$, the similar argument leads to (iii).

(iv) We will now proceed to prove inequality (iv). From Lemma 1, we know that

$$
\begin{aligned}
d(y(z,v),z,v) - d(y(z',v),z,v) &\ge \frac{r_2 - L_y}{2}\|y(z,v) - y(z',v)\|^2, \\
d(y(z,v),z',v) - d(y(z',v),z',v) &\le \frac{L_y - r_2}{2}\|y(z,v) - y(z',v)\|^2.
\end{aligned}
$$

On the other hand, we have

$$
\begin{aligned}
&d(y(z,v),z,v) - d(y(z,v),z',v) \\
&\le F(x(y(z,v),z',v),y(z,v),z,v) - F(x(y(z,v),z',v),y(z,v),z',v) \\
&= \frac{r_1}{2}\langle z + z' - 2x(y(z,v),z',v), z - z'\rangle
\end{aligned}
$$

and

$$
\begin{aligned}
&d(y(z',v),z',v) - d(y(z',v),z,v) \\
&\le F(x(y(z',v),z,v),y(z',v),z',v) - F(x(y(z',v),z,v),y(z',v),z,v) \\
&= \frac{r_1}{2}\langle z' + z - 2x(y(z',v),z,v), z' - z\rangle.
\end{aligned}
$$

Armed with these inequalities, we conclude that

$$
\begin{aligned}
(r_2 - L_y)\|y(z,v) - y(z',v)\|^2 &\le r_1\langle x(y(z',v),z,v) - x(y(z,v),z',v), z - z'\rangle \\
&\le r_1\|z - z'\|\left(\sigma_1\|y(z,v) - y(z',v)\| + \sigma_2\|z - z'\|\right) \\
&\le r_1\sigma_1\|z - z'\|\|y(z,v) - y(z',v)\| + r_1\sigma_2\|z - z'\|^2,
\end{aligned}
$$

where the last inequality is from error bounds (i) and (ii). Let $\xi := \frac{\|y(z,v) - y(z',v)\|}{\|z - z'\|}$, we have $\xi^2 \le \frac{r_1\sigma_1}{r_2 - L_y}\xi + \frac{r_1\sigma_2}{r_2 - L_y}$. Using AM-GM inequality, we obtain

$$
\frac{\|y(z,v) - y(z',v)\|}{\|z - z'\|} \le \frac{r_1\sigma_1}{r_2 - L_y} + \frac{\sigma_2}{\sigma_1} := \sigma_3.
$$

Next, we consider (v). Still from Lemma 1, we have the following inequalities:

$$
F(x,y(x,z,v),z,v) - F(x,y(x',z,v),z,v) \ge \frac{r_2 - L_y}{2}\|y(x,z,v) - y(x',z,v)\|^2,
$$

$$
F(x',y(x,z,v),z,v) - F(x',y(x',z,v),z,v) \le \frac{L_y - r_2}{2}\|y(x,z,v) - y(x',z,v)\|^2,
$$

$$
F(x,y(x,z,v),z,v) - F(x',y(x,z,v),z,v) \le \langle\nabla_x F(x',y(x,z,v),z,v), x - x'\rangle + \frac{L_x + r_1}{2}\|x - x'\|^2,
$$

$$
F(x',y(x',z,v),z,v) - F(x,y(x',z,v),z,v) \le \langle\nabla_x F(x',y(x',z,v),z,v), x' - x\rangle + \frac{L_x - r_1}{2}\|x - x'\|^2.
$$

Summing them up, we derive that

$$
(r_2 - L_y)\|y(x,z,v) - y(x',z,v)\|^2 \le L_x\|x - x'\|^2 + L_x\|x - x'\|\|y(x,z,v) - y(x',z,v)\|.
$$

Let $\xi := \frac{\|y(x,z,v)-y(x',z,v)\|}{\|x-x'\|}$. Then, we have $\xi^2 \leq \frac{L_x}{r_2-L_y} + \frac{L_x}{r_2-L_y}\xi$ and consequently $\xi \leq \frac{L_x+r_2-L_y}{r_2-L_y} = \sigma_4$.

(vi-viii) From the definition of $F$, we get

$$F(x,y(x,z,v),z,v) - F(x,y(x,z,v),z,v') = \frac{r_2}{2}\langle v'+v-2y(x,z,v), v'-v\rangle,$$

$$F(x,y(x,z,v'),z,v') - F(x,y(x,z,v'),z,v) = \frac{r_2}{2}\langle v+v'-2y(x,z,v'), v-v'\rangle.$$

Moreover, by the strong concavity of $F(x,\cdot,z,v)$, we have

$$F(x,y(x,z,v),z,v) - F(x,y(x,z,v'),z,v) \geq \frac{r_2-L_y}{2}\|y(x,z,v)-y(x,z,v')\|^2,$$

$$F(x,y(x,z,v),z,v') - F(x,y(x,z,v'),z,v') \leq \frac{L_y-r_2}{2}\|y(x,z,v)-y(x,z,v')\|^2.$$

Armed with these inequalities and Cauchy-Schwarz inequality, we conclude that

$$(r_2-L_y)\|y(x,z,v)-y(x,z,v')\|^2 \leq r_2\langle y(x,z,v)'-y(x,z,v), v'-v\rangle$$
$$\leq r_2\|y(x,z,v')-y(x,z,v)\|\|v-v'\|,$$

which gives the inequality (vi). Since $d(\cdot,z,v) = \min_{x\in\mathcal{X}} F(x,\cdot,z,v)$ is $(r_2-L_y)$-strongly concave, similarly we can derive the Lipschitz property of $y(z,v)$, as shown in (vii). $\qquad\square$

**Lemma 3** (*$L$-smooth property of dual function*) *For any fixed $z \in \mathbb{R}^n$, $v \in \mathbb{R}^d$, the dual function $d(\cdot,z,v)$ is continuously differentiable with the gradient $\nabla_y d(y,z,v) = \nabla_y F(x(y,z,v),y,z,v)$ and*

$$\|\nabla_y d(y,z,v) - \nabla_y d(y',z,v)\| \leq L_d\|y-y'\|,$$

*where $L_d := L_y\sigma_1 + L_y + r_2$.*

**Proof** Using Danskin's theorem, we know that $d(\cdot,z,v)$ is differentiable with $\nabla_y d(y,z,v) = \nabla_y F(x(y,z,v),y,z,v)$. Also, we know from the $L$-smooth property of $f$ that

$$\|\nabla_y d(y,z,v) - \nabla_y d(y',z,v)\| = \|\nabla_y F(x(y,z,v),y,z,v) - \nabla_y F(x(y',z,v),y',z,v)\|$$
$$\leq \|\nabla_y F(x(y,z,v),y,z,v) - \nabla_y F(x(y',z,v),y,z,v)\|+$$
$$\|\nabla_y F(x(y',z,v),y,z,v) - \nabla_y F(x(y',z,v),y',z,v)\|$$
$$\leq L_y\|x(y',z,v) - x(y,z,v)\| + (L_y+r_2)\|y-y'\|$$
$$\leq (L_y\sigma_1 + L_y + r_2)\|y-y'\| = L_d\|y-y'\|,$$

where the last inequality is due to the error bound in Lemma 2 (i). $\qquad\square$

Recall that $y_+(z,v) = \operatorname{proj}_{\mathcal{Y}}(y+\alpha\nabla_y F(x(y,z,v),y,z,v))$. Incorporating the iterates of DS-GDA, we have the following error bounds:

**Lemma 4** *For any $t \geq 0$, the following inequalities hold:*

(i) $\|x^{t+1} - x(y^t,z^t,v^t)\| \leq \sigma_6\|x^{t+1}-x^t\|,$

(ii) $\|y^{t+1} - y(x^{t+1},z^t,v^t)\| \leq \sigma_7\|y^{t+1}-y^t\|,$

(iii) $\|y(z^t,v^t)-y^t\| \leq \sigma_8\|y^t-y_+^t(z^t,v^t)\|,$

(iv) $\|y^{t+1} - y_+^t(z^t,v^t)\| \leq L_y\alpha\sigma_6\|x^t-x^{t+1}\|,$

*where $\sigma_6 = \frac{2cr_1+1}{cr_1-cL_x}$, $\sigma_7 = \frac{2\alpha r_2+1}{\alpha r_2-\alpha L_y}$, and $\sigma_8 = \frac{1+\alpha L_d}{\alpha(r_2-L_y)}$.*

**Proof** (i-iii) First, we consider (i), which is also called "primal error bound". From Lemma 1, we know that the mapping $\nabla_x F(\cdot,y,z,v)$ is $(r_1-L_x)$-strongly monotone and Lipschitz continuous with constant $(r_1+L_x)$ on set $\mathcal{X}$ for all $y \in \mathcal{Y}, z \in \mathbb{R}^n, v \in \mathbb{R}^d$. Adopting the proof in [49, Theorem 3.1], we can easily derive that

$$\|x^t - x(y^t,z^t,v^t)\| \leq \frac{cL_x+cr_1+1}{cr_1-cL_x}\|x^{t+1}-x^t\|,$$

which implies that

$$\|x^{t+1} - x(y^t, z^t, v^t)\| \leq \|x^{t+1} - x^t\| + \|x^t - x(y^t, z^t, v^t)\| \leq \frac{2cr_1 + 1}{cr_1 - cL_x}\|x^{t+1} - x^t\|.$$

Similarly, since $-\nabla_y F(x, \cdot, z, v)$ is $(r_2 - L_y)$-strongly monotone and Lipschitz continuous with constant $(r_2 + L_y)$ on $\mathcal{Y}$ for all $x \in \mathcal{X}, z \in \mathbb{R}^n, v \in \mathbb{R}^d$, we can derive the "primal error bound" for $y_t$ in the inequality (ii). As for (iii), notice that $y(z, v) = \text{argmax}_{y \in \mathcal{Y}} d(y, z, v)$, the "primal error bound" is defined with operator $\nabla_y d(\cdot, z, v)$. Since $d(\cdot, z, v) = \min_{x \in \mathcal{X}} F(x, \cdot, z, v)$ is $(r_2 - L_y)$-strongly concave, the operator $-\nabla_y d(\cdot, z, v)$ is $(r_2 - L_y)$-strongly monotone. Moreover, from the Lemma 3, we find $\nabla_y d(\cdot, z, v)$ is $L_d$-Lipschitz continuous on $\mathcal{Y}$ for all $z \in \mathbb{R}^n, v \in \mathbb{R}^d$. Thus, (iii) can be derived correspondingly.

(iv) Utilizing the inequality (i), we can further bound the desired term

$$\begin{aligned}
&\|y^{t+1} - y_+^t(z^t, v^t)\| \\
&= \|\text{proj}_{\mathcal{Y}}(y^t + \alpha \nabla_y F(x^{t+1}, y^t, z^t, v^t)) - \text{proj}_{\mathcal{Y}}(y^t + \alpha \nabla_y F(x(y^t, z^t, v^t), y^t, z^t, v^t))\| \\
&\leq \alpha \|\nabla_y F(x^{t+1}, y^t, z^t, v^t) - \nabla_y F(x(y^t, z^t, v^t), y^t, z^t, v^t)\| \\
&\leq \alpha L_y \|x^{t+1} - x(y^t, z^t, v^t)\| \leq L_y \alpha \sigma_6 \|x^t - x^{t+1}\|,
\end{aligned}$$

where the first inequality follows from the non-expansivity of the projection operator. □

## D  Basic Descent Lemmas

**Lemma 5 (Primal descent)** *For any $t \geq 0$, the following inequality holds:*

$$\begin{aligned}
F(x^t, y^t, z^t, v^t) \geq{}& F(x^{t+1}, y^{t+1}, z^{t+1}, v^{t+1}) + \left(\frac{1}{c} - \frac{L_x + r_1}{2}\right)\|x^{t+1} - x^t\|^2 + \\
& \langle \nabla_y F(x^{t+1}, y^t, z^t, v^t), y^t - y^{t+1}\rangle + \frac{r_2 - L_y}{2}\|y^{t+1} - y^t\|^2 + \\
& \frac{2 - \beta}{2\beta}r_1\|z^{t+1} - z^t\|^2 + \frac{\mu - 2}{2\mu}r_2\|v^{t+1} - v^t\|^2.
\end{aligned}$$

**Proof**  We firstly split the target into four parts as follows:

$$\begin{aligned}
&F(x^t, y^t, z^t, v^t) - F(x^{t+1}, y^{t+1}, z^{t+1}, v^{t+1}) \\
&= \underbrace{F(x^t, y^t, z^t, v^t) - F(x^{t+1}, y^t, z^t, v^t)}_{①} + \underbrace{F(x^{t+1}, y^t, z^t, v^t) - F(x^{t+1}, y^{t+1}, z^t, v^t)}_{②} + \\
&\quad \underbrace{F(x^{t+1}, y^{t+1}, z^t, v^t) - F(x^{t+1}, y^{t+1}, z^{t+1}, v^t)}_{③} + \underbrace{F(x^{t+1}, y^{t+1}, z^{t+1}, v^t) - F(x^{t+1}, y^{t+1}, z^{t+1}, v^{t+1})}_{④}.
\end{aligned}$$

As for ①, we have

$$\begin{aligned}
F(x^t, y^t, z^t, v^t) - F(x^{t+1}, y^t, z^t, v^t) &\geq \langle \nabla_x F(x^t, y^t, z^t, v^t), x^t - x^{t+1}\rangle - \frac{L_x + r_1}{2}\|x^{t+1} - x^t\|^2 \\
&\geq \left(\frac{1}{c} - \frac{L_x + r_1}{2}\right)\|x^{t+1} - x^t\|^2,
\end{aligned}$$

where the first inequality is from Lemma 1 and the second is due to the projection update of $x^{t+1}$, i.e., $\langle x^t - c\nabla_x F(x^t, y^t, z^t, v^t) - x^{t+1}, x^t - x^{t+1}\rangle \leq 0$. Next, from Lemma 1, one has for the inequality ② that

$$F(x^{t+1}, y^t, z^t, v^t) - F(x^{t+1}, y^{t+1}, z^t, v^t) \geq \langle \nabla_y F(x^{t+1}, y^t, z^t, v^t), y^t - y^{t+1}\rangle + \frac{r_2 - L_y}{2}\|y^{t+1} - y^t\|^2.$$

For ③, on top of the update of $z^{t+1}$, i.e, $z^{t+1} = z^t + \beta(x^{t+1} - z^t)$, we obtain

$$F(x^{t+1}, y^{t+1}, z^t, v^t) - F(x^{t+1}, y^{t+1}, z^{t+1}, v^t) = \frac{2 - \beta}{2\beta}r_1\|z^{t+1} - z^t\|^2.$$

Similarly, as for ④, following the update of $v^{t+1}$, i.e, $v^{t+1} = v^t + \mu(y^{t+1} - v^t)$, we can verify that

$$F(x^{t+1}, y^{t+1}, z^{t+1}, v^t) - F(x^{t+1}, y^{t+1}, z^{t+1}, v^{t+1}) = \frac{\mu - 2}{2\mu} r_2 \|v^{t+1} - v^t\|^2.$$

Combining all the above bounds leads to the conclusion. □

**Lemma 6 (Dual ascent)** *For any $t \geq 0$, the following inequality holds:*

$$d(y^{t+1}, z^{t+1}, v^{t+1}) \geq d(y^t, z^t, v^t) + \frac{(2-\mu)r_2}{2\mu} \|v^{t+1} - v^t\|^2 +$$
$$\frac{r_1}{2} \langle z^{t+1} + z^t - 2x(y^{t+1}, z^{t+1}, v^t), z^{t+1} - z^t \rangle +$$
$$\langle \nabla_y F(x(y^t, z^t, v^t), y^t, z^t, v^t), y^{t+1} - y^t \rangle - \frac{L_d}{2} \|y^{t+1} - y^t\|^2.$$

**Proof** The difference of the update for the dual function is controlled by the following three parts:

$$d(y^{t+1}, z^{t+1}, v^{t+1}) - d(y^t, z^t, v^t)$$
$$= \underbrace{d(y^{t+1}, z^{t+1}, v^{t+1}) - d(y^{t+1}, z^{t+1}, v^t)}_{①} + \underbrace{d(y^{t+1}, z^{t+1}, v^t) - d(y^{t+1}, z^t, v^t)}_{②} +$$
$$\underbrace{d(y^{t+1}, z^t, v^t) - d(y^t, z^t, v^t)}_{③}.$$

For the first part, following from the update of $v^{t+1}$, we have

$$① = \frac{r_2}{2} \left( \|y^{t+1} - v^t\|^2 - \|y^{t+1} - v^{t+1}\|^2 \right) = \frac{(2-\mu)r_2}{2\mu} \|v^{t+1} - v^t\|^2.$$

For the second part,

$$② = F(x(y^{t+1}, z^{t+1}, v^t), y^{t+1}, z^{t+1}, v^t) - F(x(y^{t+1}, z^t, v^t), y^{t+1}, z^t, v^t)$$
$$\geq F(x(y^{t+1}, z^{t+1}, v^t), y^{t+1}, z^{t+1}, v^t) - F(x(y^{t+1}, z^{t+1}, v^t), y^{t+1}, z^t, v^t)$$
$$= \frac{r_1}{2} \langle z^{t+1} + z^t - 2x(y^{t+1}, z^{t+1}, v^t), z^{t+1} - z^t \rangle.$$

Finally, consider the third part, from the Lemma 3, we get

$$③ = \langle \nabla_y F(x(y^t, z^t, v^t), y^t, z^t, v^t), y^{t+1} - y^t \rangle - \frac{L_d}{2} \|y^{t+1} - y^t\|^2.$$

Combining the above inequalities finishes the proof. □

**Lemma 7 (Proximal descent)** *For all $t \geq 0$, the following inequality holds:*

$$q(z^t) \geq q(z^{t+1}) + \frac{r_1}{2} \langle z^t + z^{t+1} - 2x(z^t, v(z^{t+1})), z^t - z^{t+1} \rangle.$$

**Proof** From Sion's minimax theorem [54], we have

$$q(z) = \max_v p(z, v) = \max_v \min_{x \in \mathcal{X}} \max_{y \in \mathcal{Y}} F(x, y, z, v)$$
$$= \max_v h(x(z, v), z, v) = \max_v F(x(y(z, v), z, v), y(z, v), z, v).$$

Thus, it follows that

$$q(z^t) - q(z^{t+1}) = h(x(z^t, v(z^t)), z^t, v(z^t)) - h(x(z^{t+1}, v(z^{t+1})), z^{t+1}, v(z^{t+1}))$$
$$\geq h(x(z^t, v(z^{t+1})), z^t, v(z^{t+1})) - h(x(z^{t+1}, v(z^{t+1})), z^{t+1}, v(z^{t+1}))$$
$$\geq h(x(z^t, v(z^{t+1})), z^t, v(z^{t+1})) - h(x(z^t, v(z^{t+1})), z^{t+1}, v(z^{t+1}))$$
$$\geq F(x(z^t, v(z^{t+1})), y(x(z^t, v(z^{t+1})), z^{t+1}, v(z^{t+1})), z^t, v(z^{t+1})) -$$
$$F(x(z^t, v(z^{t+1})), y(x(z^t, v(z^{t+1})), z^{t+1}, v(z^{t+1})), z^{t+1}, v(z^{t+1}))$$
$$= \frac{r_1}{2} \langle z^t + z^{t+1} - 2x(z^t, v(z^{t+1})), z^t - z^{t+1} \rangle.$$

The proof is complete. □

# E  Proof of Proposition 1

From the results in Lemmas 5, 6, and 7, we know that

$$\Phi(x^t, y^t, z^t, v^t) \geq \Phi(x^{t+1}, y^{t+1}, z^{t+1}, v^{t+1}) + \left(\frac{1}{c} - \frac{L_x + r_1}{2}\right)\|x^{t+1} - x^t\|^2 +$$

$$\left(\frac{r_2 - L_y}{2} - L_d\right)\|y^{t+1} - y^t\|^2 + \frac{(2-\beta)r_1}{2\beta}\|z^{t+1} - z^t\|^2 +$$

$$\frac{(2-\mu)r_2}{2\mu}\|v^{t+1} - v^t\|^2 + \underbrace{\langle \nabla_y F(x^{t+1}, y^t, z^t, v^t), y^{t+1} - y^t\rangle}_{\text{①}} +$$

$$\underbrace{2\langle \nabla_y F(x(y^t, z^t, v^t), y^t, z^t, v^t) - \nabla_y F(x^{t+1}, y^t, z^t, v^t), y^{t+1} - y^t\rangle}_{\text{②}} +$$

$$\underbrace{2r_1\langle x(z^t, v(z^{t+1})) - x(y^{t+1}, z^{t+1}, v^t), z^{t+1} - z^t\rangle}_{\text{③}}.$$

For the part ①, using the projection update of $y^{t+1}$, we have

$$\langle \nabla_y F(x^{t+1}, y^t, z^t, v^t), y^{t+1} - y^t\rangle \geq \frac{1}{\alpha}\|y^t - y^{t+1}\|^2.$$

The part ② is due to the Lipschitz gradient property (see Assumption 1) and error bound (i) in Lemma 4:

$$2\langle \nabla_y F(x(y^t, z^t, v^t), y^t, z^t, v^t) - \nabla_y F(x^{t+1}, y^t, z^t, v^t), y^{t+1} - y^t\rangle$$

$$\geq -2L_y\|x(y^t, z^t, v^t) - x^{t+1}\|\|y^{t+1} - y^t\|$$

$$\geq -L_y\sigma_6^2\|y^{t+1} - y^t\|^2 - L_y\sigma_6^{-2}\|x(y^t, z^t, v^t) - x^{t+1}\|^2$$

$$\geq -L_y\sigma_6^2\|y^{t+1} - y^t\|^2 - L_y\|x^{t+1} - x^t\|^2.$$

As for the part ③, for any $\kappa > 0$ it follows that

$$2r_1\langle x(z^t, v(z^{t+1})) - x(y^{t+1}, z^{t+1}, v^t), z^{t+1} - z^t\rangle$$

$$= 2r_1\langle x(z^t, v(z^{t+1})) - x(z^{t+1}, v(z^{t+1})), z^{t+1} - z^t\rangle + 2r_1\langle x(z^{t+1}, v(z^{t+1})) - x(y^{t+1}, z^{t+1}, v^t), z^{t+1} - z^t\rangle$$

$$\geq -2r_1\sigma_2\|z^{t+1} - z^t\|^2 - \frac{r_1}{\kappa}\|z^{t+1} - z^t\|^2 - r_1\kappa\|x(z^{t+1}, v(z^{t+1})) - x(y^{t+1}, z^{t+1}, v^t)\|^2,$$

where the inequality is from the Cauchy-Schwarz inequality, AM-GM inequality, and error bound (iii) in Lemma 2. Hence,

$$\Phi(x^t, y^t, z^t, v^t) - \Phi(x^{t+1}, y^{t+1}, z^{t+1}, v^{t+1})$$

$$\geq \left(\frac{1}{c} - \frac{L_x + r_1}{2} - L_y\right)\|x^{t+1} - x^t\|^2 + \left(\frac{1}{\alpha} + \frac{r_2 - L_y}{2} - L_d - L_y\sigma_6^2\right)\|y^{t+1} - y^t\|^2 +$$

$$r_1\left(\frac{2-\beta}{2\beta} - 2\sigma_2 - \frac{1}{\kappa}\right)\|z^{t+1} - z^t\|^2 + \frac{(2-\mu)r_2}{2\mu}\|v^{t+1} - v^t\|^2 -$$

$$r_1\kappa \underbrace{\|x(z^{t+1}, v(z^{t+1})) - x(y^{t+1}, z^{t+1}, v^t)\|^2}_{\text{④}}.$$

Next, we focus on the negative term. From the fact $x(z, v) = x(y(z, v), z, v)$ and $x(y, z, v) = x(y, z, v')$, the inequality ④ is bounded as follows:

$$\|x(z^{t+1}, v(z^{t+1})) - x(y^{t+1}, z^{t+1}, v^t)\|^2$$

$$\leq 2\|x(z^{t+1}, v(z^{t+1})) - x(z^{t+1}, v_+^t(z^{t+1}))\|^2 + 2\|x(z^{t+1}, v_+^t(z^{t+1})) - x(y^{t+1}, z^{t+1}, v^t)\|^2$$

$$\leq 2\|x(z^{t+1}, v(z^{t+1})) - x(z^{t+1}, v_+^t(z^{t+1}))\|^2 + 2\sigma_1^2\|y^{t+1} - y(z^{t+1}, v_+^t(z^{t+1}))\|^2. \tag{14}$$

Here, the second inequality is due to the error bound (i) in Lemma 2. We can further simplify the second error term by (iii)-(iv) in Lemma 4 and (iv), (vii) in Lemma 2:

$$\|y^{t+1} - y(z^{t+1}, v_+^t(z^{t+1}))\|^2$$

$$\leq 3\|y^{t+1} - y(z^t, v^t)\|^2 + 3\|y(z^t, v^t) - y(z^{t+1}, v^t)\|^2 + 3\|y(z^{t+1}, v^t) - y(z^{t+1}, v_+^t(z^{t+1}))\|^2$$

$$\leq 6L_y^2\alpha^2\sigma_6^2\|x^t - x^{t+1}\|^2 + 6(\sigma_8 + 1)^2\|y^t - y_+^t(z^t, v^t)\|^2 + 3\sigma_3^2\|z^t - z^{t+1}\|^2 + 3\sigma_5^2\|v^t - v_+^t(z^{t+1})\|^2. \tag{15}$$

Plugging (15) into (14), we have

$$\|x(z^{t+1}, v(z^{t+1})) - x(y^{t+1}, z^{t+1}, v^t)\|^2$$
$$\leq 2\|x(z^{t+1}, v(z^{t+1})) - x(z^{t+1}, v^t_+(z^{t+1}))\|^2 + 12L_y^2\alpha^2\sigma_1^2\sigma_6^2\|x^t - x^{t+1}\|^2 +$$
$$12\sigma_1^2(\sigma_8 + 1)^2\|y^t - y^t_+(z^t, v^t)\|^2 + 6\sigma_1^2\sigma_3^2\|z^t - z^{t+1}\|^2 + 6\sigma_1^2\sigma_5^2\|v^t - v^t_+(z^{t+1})\|^2.$$

Summing the above inequalities and letting $s_1 := \frac{1}{c} - \frac{L_x + r_1}{2} - L_y$, $s_2 := \frac{1}{\alpha} + \frac{r_2 - L_y}{2} - L :=_d -L_y\sigma_6^2$, and $s_3 := r_1\left(\frac{2-\beta}{2\beta} - 2\sigma_2 - \frac{1}{\kappa}\right)$, we have

$$\Phi(x^t, y^t, z^t, v^t) - \Phi(x^{t+1}, y^{t+1}, z^{t+1}, v^{t+1})$$
$$\geq s_1\|x^{t+1} - x^t\|^2 + s_2\|y^{t+1} - y^t\|^2 + s_3\|z^{t+1} - z^t\|^2 + \frac{(2-\mu)r_2}{2\mu}\|v^{t+1} - v^t\|^2 -$$
$$12r_1\kappa L_y^2\alpha^2\sigma_1^2\sigma_6^2\|x^t - x^{t+1}\|^2 - 12r_1\kappa\sigma_1^2(\sigma_8 + 1)^2\|y^t - y^t_+(z^t, v^t)\|^2 - 6r_1\kappa\sigma_1^2\sigma_3^2\|z^t - z^{t+1}\|^2 -$$
$$6r_1\kappa\sigma_1^2\sigma_5^2\|v^t - v^t_+(z^{t+1})\|^2 - 2r_1\kappa\|x(z^{t+1}, v(z^{t+1})) - x(z^{t+1}, v^t_+(z^{t+1}))\|^2$$
$$= \left(s_1 - 12r_1\kappa L_y^2\alpha^2\sigma_1^2\sigma_6^2\right)\|x^t - x^{t+1}\|^2 + s_2\|y^{t+1} - y^t\|^2 + \left(s_3 - 6r_1\kappa\sigma_1^2\sigma_3^2\right)\|z^{t+1} - z^t\|^2 +$$
$$\frac{(2-\mu)r_2}{2\mu}\|v^{t+1} - v^t\|^2 - 6r_1\kappa\sigma_1^2\sigma_5^2\|v^t - v^t_+(z^{t+1})\|^2 - 12r_1\kappa\sigma_1^2(\sigma_8 + 1)^2\|y^t - y^t_+(z^t, v^t)\|^2 -$$
$$2r_1\kappa\|x(z^{t+1}, v(z^{t+1})) - x(z^{t+1}, v^t_+(z^{t+1}))\|^2.$$

(16)

Moreover, to make the terms in the upper bound consistent, by error bound (iv) in Lemma 4, we have

$$\|y^{t+1} - y^t\|^2 \geq \frac{1}{2}\|y^t - y^t_+(z^t, v^t)\|^2 - \|y^{t+1} - y^t_+(z^t, v^t)\|^2$$
$$\geq \frac{1}{2}\|y^t - y^t_+(z^t, v^t)\|^2 - L_y^2\alpha^2\sigma_6^2\|x^t - x^{t+1}\|^2.$$

(17)

Similarly, we provide a lower bound for $\|v^{t+1} - v^t\|^2$ as follows:

$$\|v^{t+1} - v^t\|^2 \geq \frac{1}{2}\|v^t - v^t_+(z^{t+1})\|^2 - \|v^{t+1} - v^t_+(z^{t+1})\|^2$$
$$\geq \frac{1}{2}\|v^t - v^t_+(z^{t+1})\|^2 - \mu^2\left(4L_y^2\alpha^2\sigma_6^2\|x^t - x^{t+1}\|^2 + 4(\sigma_8 + 1)^2\|y^t - y^t_+(z^t, v^t)\|^2 + 2\sigma_3^2\|z^t - z^{t+1}\|^2\right).$$

(18)

Recalling that $v^t_+(z^{t+1}) = v^t + \mu(y(z^{t+1}, v^t) - v^t)$, the last inequality can be obtained by using error bounds (iv) in Lemma 2 and (iii), (iv) in Lemma 4, i.e.,

$$\|v^{t+1} - v^t_+(z^{t+1})\|^2 = \mu^2\|y^{t+1} - y(z^{t+1}, v^t)\|^2$$
$$\leq \mu^2\left(2\|y^{t+1} - y(z^t, v^t)\|^2 + 2\|y(z^t, v^t) - y(z^{t+1}, v^t)\|^2\right)$$
$$\leq \mu^2\left(4L_y^2\alpha^2\sigma_6^2\|x^t - x^{t+1}\|^2 + 4(\sigma_8 + 1)^2\|y^t - y^t_+(z^t, v^t)\|^2 + 2\sigma_3^2\|z^t - z^{t+1}\|^2\right).$$

Substituting (17) and (18) to (16) yields

$$\Phi(x^t, y^t, z^t, v^t) - \Phi(x^{t+1}, y^{t+1}, z^{t+1}, v^{t+1})$$
$$\geq \left(s_1 - (12r_1\kappa\sigma_1^2 + s_2 + 2\mu(2-\mu)r_2)L_y^2\alpha^2\sigma_6^2\right)\|x^{t+1} - x^t\|^2 +$$
$$\left(\frac{s_2}{2} - (12r_1\kappa\sigma_1^2 + 2\mu(2-\mu)r_2)(1+\sigma_8)^2\right)\|y^t - y^t_+(z^t, v^t)\|^2 +$$
$$\left(s_3 - (\mu(2-\mu)r_2 + 6r_1\kappa\sigma_1^2)\sigma_3^2\right)\|z^t - z^{t+1}\|^2 + \left(\frac{(2-\mu)r_2}{4\mu} - 6r_1\kappa\sigma_1^2\sigma_5^2\right)\|v^t - v^t_+(z^{t+1})\|^2 -$$
$$2r_1\kappa\|x(z^{t+1}, v(z^{t+1})) - x(z^{t+1}, v^t_+(z^{t+1}))\|^2.$$

Recalling that $L_y = \lambda L_x$, we have the following results:

- As $r_1 \geq 2L_x$, we have $\sigma_1 = \frac{L_y}{r_1 - L_x} + 1 \leq \frac{L_y}{L_x} + 1 = \lambda + 1$ and $\sigma_2 = \frac{r_1}{r_1 - L_x} \leq 2$. As $r_2 \geq 2L_y$, similarly, we find $\sigma_3 = \frac{r_1\sigma_1}{r_2 - L_y} + \frac{\sigma_2}{\sigma_1} \leq \frac{r_1(\lambda+1)}{L_y} + 2$ and $\sigma_5 = \frac{r_2}{r_2 - L_y} \leq 2$. With these

bounds and set $\kappa := 2\beta$ with $0 < \beta \le \min\{\frac{24r_1}{360r_1 + 5r_1^2\lambda + (2L_y + 5r_1)^2}, \frac{\alpha L_y^2}{384r_1(\lambda+5)(\lambda+1)^2}\}$ and $0 < \mu \le \min\{\frac{2(\lambda+5)}{2(\lambda+5)+L_y^2}, \frac{\alpha L_y^2}{64r_2(\lambda+5)}\}$ we derive that

$$\frac{(2-\mu)r_2}{4\mu} - 6r_1\kappa\sigma_1^2\sigma_5^2 \ge \frac{r_2}{2\mu} - \frac{r_2}{4} - 48(\lambda+1)^2 r_1\beta$$

$$\ge \frac{r_2}{\mu}\left(\frac{1}{2} - \frac{\mu}{4} - \frac{L_y^2\alpha\mu}{8(\lambda+5)}\right) \ge \frac{r_2}{4\mu}$$

and

$$\mu(2-\mu)r_2 + 6r_1\kappa\sigma_1^2 \le 2r_2\mu + 12r_1\beta(\lambda+1)^2 \le \frac{\alpha L_y^2}{16(\lambda+5)}. \tag{19}$$

With the upper bound (19), we can further bound the coefficient of $\|z^{t+1} - z^t\|^2$ by

$$s_3 - \left(\mu(2-\mu)r_2 + 6r_1\kappa\sigma_1^2\right)\sigma_3^2$$

$$\ge r_1\left(\frac{1}{\beta} - \frac{1}{2} - 4 - \frac{1}{2\beta}\right) - \frac{L_y^2}{16(\lambda+5)}\left(\frac{r_1^2(\lambda+1)^2}{L_y^2} + 4 + \frac{4r_1(\lambda+1)}{L_y}\right)$$

$$\ge \frac{r_1}{\beta}\left(\frac{1}{2} - \left(\frac{9}{2} + \frac{r_1(\lambda+1)}{16} + \frac{L_y^2}{20r_1} + \frac{L_y}{4}\right)\beta\right) \ge \frac{r_1}{5\beta}.$$

- Let $\frac{1}{c} - \frac{L_x + r_1}{2} \ge \frac{1}{3c}$ and $\frac{1}{3c} - L_y \ge \frac{1}{6c}$. Then, we have $\frac{1}{c} \ge \max\left\{\frac{3}{4}(L_x + r_1), 6L_y\right\}$, which implies that $s_1 \ge \frac{1}{6c}$ and $\sigma_6 = \frac{2r_1 + \frac{1}{c}}{r_1 - L_x} \ge 2 + \frac{1}{cr_1} \ge \frac{11}{4}$.

- Let $\frac{1}{\alpha} - L_y\sigma_6^2 \ge \frac{1}{3\alpha}$ and $\frac{1}{3\alpha} - L_d \ge \frac{1}{6\alpha}$. Then, we have $\frac{1}{\alpha} \ge \max\left\{\frac{3}{2}L_y\sigma_6^2, 6L_d\right\}$ and $s_2 \ge \frac{1}{6\alpha} + \frac{r_2 - L_y}{2}$. $\sigma_8 = \frac{\frac{1}{\alpha} + L_d}{r_2 - L_y} \le \frac{2}{\alpha r_2} + \lambda + 4$. With these bounds, if we further assume that $\frac{1}{\alpha} \ge 5\sqrt{\lambda+5}L_y$, we obtain

$$\frac{s_2}{2} - \left(12r_1\kappa\sigma_1^2 + 2\mu(2-\mu)r_2\right)(1+\sigma_8)^2$$

$$\ge \frac{1}{12\alpha} + \frac{r_2 - L_y}{4} - \frac{\alpha L_y^2}{8(\lambda+5)}\left(\frac{4}{\alpha^2 r_2^2} + (\lambda+5)^2 + \frac{4(\lambda+5)}{\alpha r_2}\right)$$

$$\ge \frac{1}{12\alpha} + \frac{L_y}{4} - \frac{1}{8(\lambda+5)\alpha} - \frac{(\lambda+5)\alpha L_y^2}{8} - \frac{L_y}{4}$$

$$\ge \frac{1}{60\alpha} - \frac{(\lambda+5)\alpha L_y^2}{8} \ge \frac{1}{90\alpha} \ge \frac{r_2}{15}$$

and

$$s_1 - (12r_1\kappa\sigma_1^2 + s_2 + 2\mu(2-\mu)r_2)L_y^2\alpha^2\sigma_6^2 \ge \frac{1}{6c} - \frac{\alpha^3 L_y^4\sigma_6^2}{8(\lambda+5)} - \frac{2L_y}{3}$$

$$\ge \frac{1}{6c} - \frac{13L_y}{2500} - \frac{2L_y}{3}$$

$$\ge \frac{1}{6c} - \frac{7L_y}{10} \ge \frac{1}{24c} \ge \frac{r_1}{32},$$

where the first inequality is due to $\frac{2}{3\alpha} \ge L_y\sigma_6^2$ and

$$s_2 \le \frac{1}{\alpha} + \frac{r_2 - L_y}{2} - L_y\sigma_6^2 - L_d = \frac{1}{\alpha} - \frac{r_2 + L_y}{2} - L_y\sigma_6^2 - (\sigma_1 + 1)L_y \le \frac{1}{\alpha}.$$

Putting together all the pieces, we get

$$\Phi(x^t, y^t, z^t, v^t) - \Phi(x^{t+1}, y^{t+1}, z^{t+1}, v^{t+1})$$

$$\ge \frac{r_1}{32}\|x^{t+1} - x^t\|^2 + \frac{r_2}{15}\|y^t - y_+^t(z^t, v^t)\|^2 + \frac{r_1}{5\beta}\|z^t - z^{t+1}\|^2 + \frac{r_2}{4\mu}\|v_+^t(z^{t+1}) - v^t\|^2 - 4r_1\beta\|x(z^{t+1}, v(z^{t+1})) - x(z^{t+1}, v_+^t(z^{t+1}))\|^2.$$

The proof is complete.

# F  Proof of Proximal Error Bounds and Dual Error Bounds

## F.1  Proof of Proposition 2 under Assumption 2

Note that $h(\cdot, z, v) = \max_{y \in \mathcal{Y}} F(\cdot, y, z, v)$ is $(r_1 - L_x)$-strongly convex. Hence, we have

$$
\max_v h(x(z^{t+1}, v_+^t(z^{t+1})), z^{t+1}, v) - h(x(z^{t+1}, v(z^{t+1})), z^{t+1}, v(z^{t+1}))
$$
$$
\geq h(x(z^{t+1}, v_+^t(z^{t+1})), z^{t+1}, v(z^{t+1})) - h(x(z^{t+1}, v(z^{t+1})), z^{t+1}, v(z^{t+1})) \tag{20}
$$
$$
\geq \frac{r_1 - L_x}{2} \|x(z^{t+1}, v_+^t(z^{t+1})) - x(z^{t+1}, v(z^{t+1}))\|^2.
$$

On the other side, by leveraging the KŁ Assumption 2, $h(x, z, \cdot)$ satisfies the KŁ property with same exponent [61, Theorem 5.2]. Thus, we get

$$
\max_v h(x(z^{t+1}, v_+^t(z^{t+1})), z^{t+1}, v) - h(x(z^{t+1}, v(z^{t+1})), z^{t+1}, v(z^{t+1}))
$$
$$
\leq \max_v h(x(z^{t+1}, v_+^t(z^{t+1})), z^{t+1}, v) - h(x(z^{t+1}, v_+^t(z^{t+1})), z^{t+1}, v_+^t(z^{t+1})) \tag{21}
$$
$$
\leq \frac{1}{\tau} \|\nabla_v h(x(z^{t+1}, v_+^t(z^{t+1})), z^{t+1}, v_+^t(z^{t+1}))\|^{\frac{1}{\theta}},
$$

where the first inequality is because

$$
h(x(z^{t+1}, v(z^{t+1})), z^{t+1}, v(z^{t+1})) = \max_v h(x(z^{t+1}, v), z^{t+1}, v)
$$
$$
\geq h(x(z^{t+1}, v_+^t(z^{t+1})), z^{t+1}, v_+^t(z^{t+1})).
$$

Next, we further bound the right-hand part. Recalling the definition of $v_+^t(z^{t+1})$ and with the bound (vii) in Lemma 2, we obtain

$$
\|\nabla_v h(x(z^{t+1}, v_+^t(z^{t+1})), z^{t+1}, v_+^t(z^{t+1}))\|
$$
$$
= r_2 \|v_+^t(z^{t+1}) - y(x(z^{t+1}, v_+^t(z^{t+1})), z^{t+1}, v_+^t(z^{t+1}))\|
$$
$$
= r_2 \|(1 - \mu)v^t + \mu y(z^{t+1}, v^t) - y(z^{t+1}, v_+^t(z^{t+1}))\| \tag{22}
$$
$$
\leq r_2(1 - \mu)\|v^t - y(z^{t+1}, v^t)\| + r_2 \|y(z^{t+1}, v^t) - y(z^{t+1}, v_+^t(z^{t+1}))\|
$$
$$
\leq r_2 \left( \frac{(1 - \mu)}{\mu} + \sigma_5 \right) \|v_+^t(z^{t+1}) - v^t\|.
$$

Combining (20) (21), and (22), we have

$$
\|x(z^{t+1}, v_+^t(z^{t+1})) - x(z^{t+1}, v(z^{t+1}))\|^2 \leq \frac{2}{(r_1 - L_x)\tau} \left( \frac{r_2(1 - \mu)}{\mu} + r_2\sigma_5 \right)^{\frac{1}{\theta}} \|v_+^t(z^{t+1}) - v^t\|^{\frac{1}{\theta}}.
$$

The proof is complete.

## F.2  Proof of Proposition 2 under Assumption 3

By the strong convexity of $h(\cdot, z, v) = \max_{y \in \mathcal{Y}} F(\cdot, y, z, v)$, we have

$$
h(x(z^{t+1}, v_+^t(z^{t+1})), z^{t+1}, v(z^{t+1})) - h(x(z^{t+1}, v_+^t(z^{t+1})), z^{t+1}, v_+^t(z^{t+1}))
$$
$$
\geq h(x(z^{t+1}, v_+^t(z^{t+1})), z^{t+1}, v(z^{t+1})) - h(x(z^{t+1}, v(z^{t+1})), z^{t+1}, v(z^{t+1}))
$$
$$
\geq \frac{r_1 - L_x}{2} \|x(z^{t+1}, v_+^t(z^{t+1})) - x(z^{t+1}, v(z^{t+1}))\|^2.
$$

On the other side,

$$
h(x(z^{t+1}, v_+^t(z^{t+1})), z^{t+1}, v(z^{t+1})) - h(x(z^{t+1}, v_+^t(z^{t+1})), z^{t+1}, v_+^t(z^{t+1}))
$$
$$
\leq \langle \nabla_v h(x(z^{t+1}, v_+^t(z^{t+1})), z^{t+1}, v_+^t(z^{t+1})), v(z^{t+1}) - v_+^t(z^{t+1}) \rangle
$$
$$
\leq 2 \operatorname{diam}(\mathcal{Y}) r_2 \left( \frac{1 - \mu}{\mu} + \sigma_5 \right) \|v_+^t(z^{t+1}) - v^t\|.
$$

Here, the first inequality is because $h(x, z, \cdot) = \max_{y \in \mathcal{Y}} f(x, y) - \frac{r_2}{2} \|y - \cdot\|^2$ is concave [52, Theorem 2.26]. The last inequality is from (22) and the fact that $\|v^t\| \leq \operatorname{diam}(\mathcal{Y})$, which could be derived from mathematical induction. Moreover, $v(z) = \operatorname{argmax}_v \{ \max_{y \in \mathcal{Y}} f(x(y, z, v), y) + \frac{r_1}{2} \|x(y, z, v) - z\|^2 - \frac{r_2}{2} \|y - v\|^2 \}$ should satisfy $v(z) = \operatorname{prox}_{\frac{f(x(\cdot, z, v), \cdot) + \frac{r_1}{2} \|x(\cdot, z, v) - z\|^2 + \mathbf{1}_{\mathcal{Y}}(\cdot)}{r_2}} (z(v))$ [52, Theorem 2.26] , which indicates that $\|v(z)\| \leq \operatorname{diam}(\mathcal{Y})$. Thus,

$$\|x(z^{t+1}, v_+^t(z^{t+1})) - x(z^{t+1}, v(z^{t+1}))\|^2 \leq \frac{4r_2 \operatorname{diam}(\mathcal{Y})}{r_1 - L_x} \left( \frac{1 - \mu}{\mu} + \sigma_5 \right) \|v_+^t(z^{t+1}) - v^t\|.$$

### F.3 Dual Error Bounds

Before presenting the results, we first introduce some useful notation:

- $x_{r_1}(y, z) := \operatorname{argmin} f(x, y) + \frac{r_1}{2} \|x - z\|^2.$
- $x^*(z) := \operatorname{argmin}_{x \in \mathcal{X}} \max_{y \in \mathcal{Y}} f(x, y) + \frac{r_1}{2} \|x - z\|^2.$
- $y_+(z) := \operatorname{proj}_{\mathcal{Y}}(y + \alpha \nabla_y f(x_{r_1}(y, z), y)).$

**Proposition 3** *Suppose that Assumption 2 holds. Then, we have*

$$\|x^*(z) - x(z, v)\|^2 \leq \omega_2 \|v - y(z, v)\|^{\frac{1}{\theta}},$$

*where* $\omega_2 := \frac{2r_2^{\frac{1}{\theta}}}{\tau(r_1 - L_x)}.$

**Proof** By the strong convexity of $\phi(\cdot, z) = \max_{y \in \mathcal{Y}} f(\cdot, y) + \frac{r_1}{2} \| \cdot -z\|^2$, we have

$$\max_{y \in \mathcal{Y}} f(x(z, v), y) + \frac{r_1}{2} \|x(z, v) - z\|^2 - \max_{y \in \mathcal{Y}} f(x^*(z), y) + \frac{r_1}{2} \|x^*(z) - z\|^2$$

$$\geq \frac{r_1 - L_x}{2} \|x^*(z) - x(z, v)\|^2.$$

On the other hand, we have

$$\max_{y \in \mathcal{Y}} f(x(z, v), y) + \frac{r_1}{2} \|x(z, v) - z\|^2 - \max_{y \in \mathcal{Y}} f(x^*(z), y) + \frac{r_1}{2} \|x^*(z) - z\|^2$$

$$\leq \max_{y \in \mathcal{Y}} f(x(z, v), y) + \frac{r_1}{2} \|x(z, v) - z\|^2 - \left( f(x(z, v), y(z, v)) + \frac{r_1}{2} \|x(z, v) - z\|^2 \right)$$

$$= \max_{y \in \mathcal{Y}} f(x(z, v), y) - f(x(z, v), y(z, v)).$$

Recall that $x(z, v) = \operatorname{argmin}_{x \in \mathcal{X}} \max_{y \in \mathcal{Y}} F(x, y, z, v)$. Then, the first inequality is from the fact that

$$\max_{y \in \mathcal{Y}} f(x^*(z), y) + \frac{r_1}{2} \|x^*(z) - z\|^2 \geq \max_{y \in \mathcal{Y}} \min_{x \in \mathcal{X}} f(x, y) + \frac{r_1}{2} \|x - z\|^2$$

$$\geq \min_{x \in \mathcal{X}} f(x, y(z, v)) + \frac{r_1}{2} \|x - z\|^2$$

$$= f(x(z, v), y(z, v)) + \frac{r_1}{2} \|x(z, v) - z\|^2.$$

Under Assumption 2, we find that

$$\max_{y \in \mathcal{Y}} f(x(z, v), y) - f(x(z, v), y(z, v)) \leq \frac{1}{\tau} \operatorname{dist}(\mathbf{0}, -\nabla_y f(x(z, v), y(z, v)) + \partial \mathbf{1}_{\mathcal{Y}}(y(z, v)))^{\frac{1}{\theta}}$$

$$\leq \frac{1}{\tau} \|r_2(v - y(z, v))\|^{\frac{1}{\theta}},$$

where the last inequality is due to the first optimality condition of $y(z, v) = \operatorname{argmax}_{y \in \mathcal{Y}} f(x(y, z, v), y) - \frac{r_2}{2} \|y - v\|^2$. $\square$

**Proposition 4** *Suppose that Assumption 3 holds. Then, we have*

$$\|x^*(z) - x(z, v)\|^2 \leq \omega_3 \|v - y(z, v)\|,$$

*where* $\omega_3 := \frac{4r_1 \operatorname{diam}(\mathcal{Y})}{r_1 - L_x}.$

**Proof** We first define $\psi(x,y,z) = f(x,y) + \frac{r_1}{2}\|x - z\|^2$ and $Y(z) = \operatorname{argmax}_{y\in\mathcal{Y}} \min_{x\in\mathcal{X}} \psi(x,y,z) = \operatorname{argmax}_{y\in\mathcal{Y}} \psi(x_{r_1}(y,z),y,z)$. Let $y^*(z)$ be an arbitrary solution in $Y(z)$. By the strong convexity of $\psi(\cdot,y,z) = f(\cdot,y) + \frac{r_1}{2}\|\cdot -z\|^2$, we have

$$\psi(x(z,v),y^*(z),z) - \psi(x(z,v),y(z,v),z) \geq \psi(x(z,v),y^*(z),z) - \psi(x^*(z),y^*(z),z)$$
$$\geq \frac{r_1 - L_x}{2}\|x^*(z) - x(z,v)\|^2.$$

Here, the first inequality is beacuse $\psi(x^*(z),y^*(z),z) = \max_{y\in\mathcal{Y}} \psi(x_{r_1}(y,z),y,z) \geq \psi(x_{r_1}(y(z,v),z),y(z,v),z) = \psi(x(z,v),y(z,v),z)$. On the other hand, by the concavity of $\Psi(x,\cdot,z)$,

$$\psi(x(z,v),y^*(z),z) - \psi(x(z,v),y(z,v),z) \leq \langle \nabla_y\psi(x(z,v),y(z,v),z), y^*(z) - y(z,v)\rangle$$
$$\leq r_1\|y(z,v) - v\|\|y^*(z) - y(z,v)\|$$
$$\leq 2r_1\operatorname{diam}(\mathcal{Y})\|y(z,v) - v\|.$$

The second inequality is due to $\langle \nabla_y f(x(z,v),y(z,v)) - r_1(y(z,v) - v), y^*(z) - y(z,v)\rangle \leq 0$ and the last inequality is because both $y^*(z)$ and $y(z,v)$ lie in set $\mathcal{Y}$. $\qquad\square$

## G  Proof of Theorem 1

Before presenting the proof of Theorem 1, we first introduce the following lemma.

**Lemma 8** *Let $\epsilon \geq 0$. Suppose that*

$$\max\left\{\frac{\|x^t - x^{t+1}\|}{c}, \frac{\|y^t - y^t_+(z^t,v^t)\|}{\alpha}, \frac{\|z^t - z^{t+1}\|}{\beta}, \frac{\|v^t - v^{t+1}\|}{\mu}\right\} \leq \epsilon.$$

*Then, there exists a $\rho > 0$ such that $(x^{t+1}, y^{t+1})$ is a $\rho\epsilon$-GS.*

**Proof** In accordance with definition 1, we just need to evaluate the two terms $\operatorname{dist}(\mathbf{0}, \nabla_x f(x^{t+1},y^{t+1}) + \partial\mathbf{1}_{\mathcal{X}}(x^{t+1}))$ and $\operatorname{dist}(\mathbf{0}, -\nabla_y f(x^{t+1},y^{t+1}) + \partial\mathbf{1}_{\mathcal{Y}}(y^{t+1}))$. We first note that

$$x^{t+1} = \operatorname{proj}_{\mathcal{X}}(x^t - c\nabla_x F(x^t,y^t,z^t,v^t)) = \operatorname*{argmin}_{x\in\mathcal{X}}\left\{\frac{1}{2}\|x - x^t + c\nabla_x F(x^t,y^t,z^t,v^t)\|^2\right\}.$$

Its optimality condition yields

$$\mathbf{0} \in x^{t+1} - x^t + c\nabla_x f(x^t,y^t) + cr_1(x^t - z^t) + \partial\mathbf{1}_{\mathcal{X}}(x^{t+1}),$$

which implies that

$$\frac{1}{c}\left(x^t - x^{t+1}\right) - r_1(x^t - z^t) + \nabla_x f(x^{t+1},y^{t+1}) - \nabla_x f(x^t,y^t) \in \nabla_x f(x^{t+1},y^{t+1}) + \partial\mathbf{1}_{\mathcal{X}}(x^{t+1}).$$

Then, we are ready to simplify the stationarity measure as follows:

$$\operatorname{dist}(\mathbf{0}, \nabla_x f(x^{t+1},y^{t+1}) + \partial\mathbf{1}_{\mathcal{X}}(x^{t+1}))$$
$$\leq \left(\frac{1}{c} + L_x\right)\|x^t - x^{t+1}\| + r_1\|x^t - z^t\| + L_x\|y^t - y^{t+1}\|$$
$$\leq \left(\frac{1}{c} + L_x + r_1 + L_x L_y \alpha\sigma_6\right)\|x^t - x^{t+1}\| + \frac{r_2}{\beta}\|z^t - z^{t+1}\| + L_x\|y^t - y^t_+(z^t,v^t)\|$$
$$\leq \frac{r_1 - L_x + r_1^2 - L_x^2 + (2r_1 + 1)L_x L_y}{c(r_1 - L_x)}\|x^t - x^{t+1}\| + \frac{r_2}{\beta}\|z^t - z^{t+1}\| + \frac{L_x}{\alpha}\|y^t - y^t_+(z^t,v^t)\|.$$

The last inequality is because $\max\left\{\frac{\|x^t-x^{t+1}\|}{c}, \frac{\|y^t-y^t_+(z^t,v^t)\|}{\alpha}, \frac{\|z^t-z^{t+1}\|}{\beta}\right\} \leq \epsilon$. Similarly, we have

$$\frac{1}{\alpha}(y^t-y^{t+1}) - r_2(y^t-v^t) + \nabla_y f(x^{t+1},y^t) - \nabla_y f(x^{t+1},y^{t+1}) \in -\nabla_y f(x^{t+1},y^{t+1}) + \partial\mathbf{1}_{\mathcal{Y}}(y^{t+1}),$$

which can be directly derived from the update of $y^{t+1}$. Then, we can simplify the expression

$$\text{dist}(\mathbf{0}, -\nabla_y f(x^{t+1}, y^{t+1}) + \partial \mathbf{1}_{\mathcal{Y}}(y^{t+1}))$$

$$\leq \left(\frac{1}{\alpha} + L_y\right) \|y^t - y^{t+1}\| + r_2 \|y^t - v^t\|$$

$$\leq \left(\frac{1}{\alpha} + L_y + r_2\right) \|y^t - y_+^t(z^t, v^t)\| + \left(\frac{1}{\alpha} + L_y + r_2\right) L_y \alpha \sigma_6 \|x^t - x^{t+1}\| + \frac{r_2}{\mu} \|v^t - v^{t+1}\|$$

$$\leq \frac{1 + L_y + r_2}{\alpha} \|y^t - y_+^t(z^t, v^t)\| + \frac{L_y(1 + L_y + r_2)(2r_1 + 1)}{c(r_1 - L_x)} \|x^t - x^{t+1}\| + \frac{r_2}{\mu} \|v^t - v^{t+1}\|.$$

The last inequality is due to $\max\left\{\frac{\|x^t - x^{t+1}\|}{c}, \frac{\|y^t - y_+^t(z^t, v^t)\|}{\alpha}, \frac{\|v^t - v^{t+1}\|}{\mu}\right\} \leq \epsilon$. The proof is complete. $\qquad \square$

**Proof of Theorem 1** Firstly, it is easy to check that $\Phi(x, y, z, v)$ is lower bounded by $\bar{F}$. Let

$$\zeta := \max\left\{\frac{r_1}{32} \|x^{t+1} - x^t\|^2, \frac{r_2}{15} \|y^t - y_+^t(z^t, v^t)\|^2, \frac{r_1}{5\beta} \|z^t - z^{t+1}\|^2, \frac{r_2}{4\mu} \|v_+^t(z^{t+1}) - v^t\|^2\right\}.$$

Then, we consider the following two cases separately:

- There exists $t \in \{0, 1, \ldots, T-1\}$ such that

$$\frac{1}{2}\zeta \leq 4r_1\beta \|x(z^{t+1}, v(z^{t+1})) - x(z^{t+1}, v_+^t(z^{t+1}))\|^2.$$

- For any $t \in \{0, 1, \ldots, T-1\}$, we have

$$\frac{1}{2}\zeta \geq 4r_1\beta \|x(z^{t+1}, v(z^{t+1})) - x(z^{t+1}, v_+^t(z^{t+1}))\|^2.$$

We first consider the first case. If the KŁ exponent $\theta \in (\frac{1}{2}, 1)$, then from Proposition 2,

$$\|v_+^t(z^{t+1}) - v^t\|^2 \leq \frac{32r_1\mu\beta}{r_2} \|x(z^{t+1}, v(z^{t+1})) - x(z^{t+1}, v_+^t(z^{t+1}))\|^2$$

$$\leq \frac{32r_1\mu\beta\omega_0}{r_2} \|v_+^t(z^{t+1}) - v^t\|^{\frac{1}{\theta}}.$$

Then, we have $\|v_+^t(z^{t+1}) - v^t\| \leq \rho_1 \beta^{\frac{\theta}{2\theta-1}}$, where $\rho_1 := \left(\frac{32r_1\mu\omega_0}{r_2}\right)^{\frac{\theta}{2\theta-1}}$. Armed with this, we can bound other terms as follows:

$$\frac{\|x^{t+1} - x^t\|^2}{c^2} \leq \frac{256\beta}{c^2} \|x(z^{t+1}, v(z^{t+1})) - x(z^{t+1}, v_+^t(z^{t+1}))\|^2$$

$$\leq \frac{256\beta\omega_0}{c^2} \|v_+^t(z^{t+1}) - v^t\|^{\frac{1}{\theta}} = \rho_2 \beta^{\frac{2\theta}{2\theta-1}},$$

$$\frac{\|y^t - y_+^t(z^t, v^t)\|^2}{\alpha^2} \leq \frac{120r_1\beta}{r_2\alpha^2} \|x(z^{t+1}, v(z^{t+1})) - x(z^{t+1}, v_+^t(z^{t+1}))\|^2$$

$$\leq \frac{120r_1\beta\omega_0}{r_2\alpha^2} \|v_+^t(z^{t+1}) - v^t\|^{\frac{1}{\theta}} = \rho_3 \beta^{\frac{2\theta}{2\theta-1}},$$

$$\frac{\|z^{t+1} - z^t\|^2}{\beta^2} \leq 40 \|x(z^{t+1}, v(z^{t+1})) - x(z^{t+1}, v_+^t(z^{t+1}))\|^2 \leq 40\omega_0 \|z_+^t(v^{t+1}) - z^t\|^{\frac{1}{\theta}} = \rho_4 \beta^{\frac{1}{2\theta-1}},$$

$$\frac{\|v^{t+1} - v^t\|^2}{\mu^2} \leq \frac{2\|v_+^t(z^{t+1}) - v^t\|^2}{\mu^2} + 2\left(4L_y^2\alpha^2\sigma_6^2 \|x^t - x^{t+1}\|^2 + 4(\sigma_8 + 1)^2 \|y^t - y_+^t(z^t, v^t)\|^2 + \right.$$

$$\left. 2\sigma_3^2 \|z^t - z^{t+1}\|^2\right)$$

$$\leq \frac{2\rho_1^2}{\mu^2} \beta^{\frac{2\theta}{2\theta-1}} + 8L_y^2\alpha^2\sigma_6^2 c^2 \rho_2 \beta^{\frac{2\theta}{2\theta-1}} + 8(\sigma_8 + 1)^2\alpha^2\rho_3 \beta^{\frac{2\theta}{2\theta-1}} + 4\sigma_3^2\rho_3 \beta^{\frac{4\theta-1}{2\theta-1}}$$

$$\leq \rho_5 \beta^{\frac{2\theta}{2\theta-1}},$$

where $\rho_2 := \frac{256\omega_0}{c^2}\rho_1^{\frac{1}{\theta}}$, $\rho_3 := \frac{120r_1\omega_0}{r_2\alpha^2}\rho_1^{\frac{1}{\theta}}$, $\rho_4 := 40\omega_0\rho_1^{\frac{1}{\theta}}$ and $\rho_5 := \frac{2\rho_1^2}{\mu^2} + 8L_y^2\alpha^2\sigma_6^2c^2\rho_2 + 8(\sigma_8 + 1)^2\alpha^2\rho_3 + 4\sigma_3^2\rho_3$. According to Lemma 8, there exists a $\rho > 0$ such that $(x^{t+1}, y^{t+1})$ is a $\rho\epsilon$-GS, where $\epsilon = \max\{\sqrt{\rho_2}\beta^{\frac{\theta}{2\theta-1}}, \sqrt{\rho_3}\beta^{\frac{\theta}{2\theta-1}}, \sqrt{\rho_4}\beta^{\frac{\theta}{2\theta-1}}, \sqrt{\rho_5}\beta^{\frac{1}{4\theta-2}}\}$. For the general concave case, the above analysis holds with $\theta = 1$ and $(x^{t+1}, y^{t+1})$ is a $\rho\max\{\sqrt{\rho_2}\beta\sqrt{\rho_4}\beta, \sqrt{\rho_5}\beta, \sqrt{\rho_3}\beta^{\frac{1}{2}}\}$-GS by replacing $\omega_0$ with $\omega_1$.

One remaining thing is to prove that $z^{t+1}$ is an $\mathcal{O}(\beta^{\frac{1}{4\theta-2}})$-OS. By the dual error bounds in Proposition 3 and error bounds in Lemmas 2 and 4, we have

$$\|z^{t+1} - x^*(z^{t+1})\|$$
$$\leq \|z^{t+1} - z^t\| + \|z^t - x^{t+1}\| + \|x^{t+1} - x(y^t, z^t, v^t)\| + \|x(y^t, z^t, v^t) - x(z^t, v^t)\| +$$
$$\|x(z^t, v^t) - x(z^{t+1}, v^t)\| + \|x(z^{t+1}, v^t) - x^*(z^{t+1})\|$$
$$\leq (1 + \sigma_2)\|z^{t+1} - z^t\| + \frac{\|z^t - z^{t+1}\|}{\beta} + \sigma_6\|x^t - x^{t+1}\| + \sigma_1\sigma_8\|y^t - y_+^t(z^t, v^t)\| +$$
$$\omega_2\|v^t - y(z^{t+1}, v^t)\|^{\frac{1}{2\theta}}$$
$$\leq (1 + \sigma_2)\sqrt{\rho_4}\beta^{\frac{4\theta-1}{4\theta-2}} + \sqrt{\rho_4}\beta^{\frac{1}{4\theta-2}} + (\sigma_6\sqrt{\rho_2} + \sigma_1\sigma_8\sqrt{\rho_3})\beta^{\frac{\theta}{2\theta-1}} + \omega_2\rho_1\beta^{\frac{1}{4\theta-2}}$$
$$= \mathcal{O}(\beta^{\frac{1}{4\theta-2}}). \tag{23}$$

Now, we consider the second case. Since

$$\Phi(x^t, y^t, z^t, v^t) - \Phi(x^{t+1}, y^{t+1}, z^{t+1}, v^{t+1})$$
$$\geq \frac{r_1}{64}\|x^{t+1} - x^t\|^2 + \frac{r_2}{30}\|y^t - y_+^t(z^t, v^t)\|^2 + \frac{r_1}{10\beta}\|z^t - z^{t+1}\|^2 + \frac{r_2}{8\mu}\|v_+^t(z^{t+1}) - v^t\|^2$$

holds for $t \in \{0, 1, \dots, T-1\}$ and $\Phi(x, y, z, v) \geq \bar{F}$, we know that

$$\Phi(x^0, y^0, z^0, v^0) - \bar{F}$$
$$\geq \sum_{T=0}^{T-1} \frac{r_1}{64}\|x^{t+1} - x^t\|^2 + \frac{r_2}{30}\|y^t - y_+^t(z^t, v^t)\|^2 + \frac{r_1}{10\beta}\|z^t - z^{t+1}\|^2 + \frac{r_2}{8\mu}\|v_+^t(z^{t+1}) - v^t\|^2$$
$$\geq T\min\left\{\frac{r_1c^2}{64}, \frac{r_2\alpha^2}{30}, \frac{r_1}{10}, \frac{r_2\mu}{8}\right\}\left(\frac{\|x^{t+1} - x^t\|^2}{c^2} + \frac{\|y^t - y_+^t(z^t, v^t)\|^2}{\alpha^2}\right) +$$
$$T\min\left\{\frac{r_1c^2}{64}, \frac{r_2\alpha^2}{30}, \frac{r_1}{10}, \frac{r_2\mu}{8}\right\}\left(\frac{\|z^t - z^{t+1}\|^2}{\beta} + \frac{\|v_+^t(z^{t+1}) - v^t\|^2}{\mu^2}\right).$$

Since $\Phi(x, y, z, v) \geq \bar{F}$, there exists a $t \in \{0, 1, \dots, T-1\}$ such that

$$\max\left\{\frac{\|x^t - x^{t+1}\|^2}{c^2}, \frac{\|y^t - y_+^t(z^t, v^t)\|^2}{\alpha^2}, \frac{\|z^t - z^{t+1}\|^2}{\beta}, \frac{\|v^t - v_+^t(z^{t+1})\|^2}{\mu^2}\right\}$$
$$\leq \frac{\Phi(x^0, y^0, z^0, v^0) - \bar{F}}{T\min\left\{\frac{r_1c^2}{64}, \frac{r_2\alpha^2}{30}, \frac{r_1}{10}, \frac{r_2\mu}{8}\right\}} =: \frac{\eta}{T}.$$

Note that

$$\frac{\|v^{t+1} - v^t\|^2}{\mu^2} \leq \frac{2\|v_+^t(z^{t+1}) - v^t\|^2}{\mu^2} + 2\left(4L_y^2\alpha^2\sigma_6^2\|x^t - x^{t+1}\|^2 + 4(\sigma_8 + 1)^2\|y^t - y_+^t(z^t, v^t)\|^2 +\right.$$
$$\left. 2\sigma_3^2\|z^t - z^{t+1}\|^2\right) \leq \frac{\eta\left(2 + 8L_y^2\alpha^2\sigma_6^2c^2 + 8(\sigma_8 + 1)^2\alpha^2 + 4\sigma_3^2\beta\right)}{T} \leq \frac{\rho_6}{T},$$

where $\rho_6 = \eta\left(2 + 8L_y^2\alpha^2\sigma_6^2c^2 + 8(\sigma_8 + 1)^2\alpha^2 + 4\sigma_3^2\right)$. Thus, we know there exists a $\rho > 0$ such that $(x^{t+1}, y^{t+1})$ is a $\rho\epsilon$-GS, where $\epsilon = \sqrt{\frac{\rho_6}{T\beta}}$. Similar argument as (23) shows that $z^{t+1}$ is an $\mathcal{O}(\frac{1}{\sqrt{T\beta}})$-OS. Moreover, if we choose $\beta \leq \mathcal{O}(T^{-\frac{2\theta-1}{2\theta}})$ when the KŁ exponent $\theta \in (\frac{1}{2}, 1)$ and $\beta = \mathcal{O}(T^{-\frac{1}{2}})$ for general concave case, the two cases coincide. We conclude that under KŁ

assumption, $(x^{t+1}, y^{t+1})$ is an $\mathcal{O}(T^{-\frac{1}{4\theta}})$-GS and $z^{t+1}$ is an $\mathcal{O}(T^{-\frac{1}{4\theta}})$-OS. When the dual function is concave, $(x^{t+1}, y^{t+1})$ is an $\mathcal{O}(T^{-\frac{1}{4}})$-GS and $z^{t+1}$ is an $\mathcal{O}(T^{-\frac{1}{4}})$-OS.

Next, suppose that the KŁ exponent $\theta \in (0, \frac{1}{2}]$. From Proposition 2, we obtain

$$
\begin{aligned}
\|x(z^{t+1}, v(z^{t+1})) - x(z^{t+1}, v_+^t(z^{t+1}))\|^2 &\leq \omega_0 \|v_+^t(z^{t+1}) - v^t\|^{\frac{1}{\theta}} \\
&\leq \omega_0 \left(2 \operatorname{diam}(\mathcal{Y})\right)^{\frac{1}{\theta}-2} \|v_+^t(z^{t+1}) - v^t\|^2.
\end{aligned}
\tag{24}
$$

The last inequality is from $\|v_+^t(z^{t+1}) - v^t\| = \beta \|y(z^{t+1}, v^t) - v^t\| \leq 2 \operatorname{diam}(\mathcal{Y})$ since $\|v^t\| \leq \operatorname{diam}(\mathcal{Y})$. Armed with the homogeneous proximal error bound (24), we have

$$
\begin{aligned}
&\Phi(x^t, y^t, z^t, v^t) - \Phi(x^{t+1}, y^{t+1}, z^{t+1}, v^{t+1}) \\
&\geq \frac{r_1}{32}\|x^{t+1} - x^t\|^2 + \frac{r_2}{15}\|y^t - y_+^t(z^t, v^t)\|^2 + \frac{r_1}{5\beta}\|z^t - z^{t+1}\|^2 + \frac{r_2}{4\mu}\|v_+^t(z^{t+1}) - v^t\|^2 - \\
&\quad 4r_1\beta\omega_0 \left(2\operatorname{diam}(\mathcal{Y})\right)^{\frac{1}{\theta}-2} \|v_+^t(z^{t+1}) - v^t\|^2 \\
&\geq \frac{r_1}{32}\|x^{t+1} - x^t\|^2 + \frac{r_2}{15}\|y^t - y_+^t(z^t, v^t)\|^2 + \frac{r_1}{5\beta}\|z^t - z^{t+1}\|^2 + \\
&\quad \left(\frac{r_2}{4\mu} - 4r_1\beta\omega_0 \left(2\operatorname{diam}(\mathcal{Y})\right)^{\frac{1}{\theta}-2}\right) \|v_+^t(z^{t+1}) - v^t\|^2 \\
&\geq \frac{r_1}{32}\|x^{t+1} - x^t\|^2 + \frac{r_2}{15}\|y^t - y_+^t(z^t, v^t)\|^2 + \frac{r_1}{5\beta}\|z^t - z^{t+1}\|^2 + +\frac{r_2}{8\mu}\|v_+^t(z^{t+1}) - v^t\|^2,
\end{aligned}
$$

where the last inequality is because $\beta \leq \frac{r_2}{32r_1\mu\omega_0(2\operatorname{diam}(\mathcal{Y}))^{\frac{1}{\theta}-2}}$. The rest of the proof is similar to that in the second case when $\theta \in (\frac{1}{2}, 1)$, and we conclude that $(x^{t+1}, y^{t+1})$ is a $\mathcal{O}(T^{-\frac{1}{2}})$-GS and $z^{t+1}$ is an $\mathcal{O}(T^{-\frac{1}{2}})$-OS.

# H   Convergence Anaylsis for (Non)convex-Nonconcave Problems

Similar convergence results for (non)convex-nonconcave problems can be established with the symmetric property of our Lyapunov function. Specifically, we can derive the following Proposition 5 by doing an easy transformation of $(x, z, r_1, c, \beta, L_x)$ to $(y, v, r_2, \alpha, \mu, L_y)$:

**Proposition 5 (Basic descent estimate)** *Suppose that Assumption 1 holds and $r_1 \geq 2L$, $r_2 \geq 2\lambda L$ with the parameters*

$$
0 < \alpha \leq \min\left\{\frac{4}{3(\lambda L + r_2)}, \frac{1}{6L}\right\}, 0 < c \leq \min\left\{\frac{2}{3L\sigma'^2}, \frac{1}{6L_h}, \frac{1}{5\sqrt{\lambda + 5}L}\right\}
$$

$$
0 < \mu \leq \min\left\{\frac{24r_2}{360r_2 + 5r_2^2\lambda + (2L + 5r_2)^2}, \frac{cL^2}{384r_2(\lambda + 5)(\lambda + 1)^2}\right\},
$$

$$
0 < \beta \leq \min\left\{\frac{2(\lambda + 5)}{2(\lambda + 5) + L^2}, \frac{cL^2}{64r_1(\lambda + 5)}\right\}.
$$

*Then, for any $t \geq 0$,*

$$
\begin{aligned}
\Psi^t - \Psi^{t+1} &\geq \frac{r_2}{32}\|x^t - x_+^t(z^t, v^t)\|^2 + \frac{r_1}{15}\|y^t - y^{t+1}\|^2 + \frac{r_2}{5\mu}\|v^t - v^{t+1}\|^2 + \frac{r_1}{4\beta}\|z_+^t(v^{t+1}) - z^t\|^2 \\
&\quad - 4r_2\mu\|y(z(v^{t+1}), v^{t+1}) - y(z_+^t(v^{t+1}), v^{t+1})\|^2,
\end{aligned}
$$

*where $\sigma' := \frac{2\alpha r_2 + 1}{\alpha(r_2 - \lambda L)}$ and $L_h := \left(\frac{L}{r_2 - \lambda L} + 2\right)L + r_1$. Moreover, we set $x_+(z, v) := \operatorname{proj}_{\mathcal{X}}(z + c\nabla_x F(x, y(x, z, v), z, v))$ and $z_+(v) := z + \mu(x(y(z, v), z, v) - z)$ with the following definitions: (i) $y(z, v) := \operatorname{argmin}_{y \in \mathcal{Y}} \max_{x \in \mathcal{X}} F(x, y, z, v)$, (ii) $z(v) := \operatorname{argmin}_{z \in \mathbb{R}^n} P(z, v)$.*

To handle the negative term in the basic decrease estimate (Proposition 5), we would impose the following assumptions on primal function:

**Assumption 4 (KŁ property for primal function)** *For any fixed point $y \in \mathcal{Y}$, $z \in \mathbb{R}^n$, and $v \in \mathbb{R}^d$, the problem $\min_{x \in \mathcal{X}} F(x, y, z, v)$ has a nonempty solution set and a finite optimal value. Moreover, there exist $\tau_1 > 0$, $\theta_1 \in (0, 1)$ such that*

$$\left( f(x, y) - \min_{x' \in \mathcal{X}} f(x', y) \right)^{\theta_1} \leq \frac{1}{\tau_1} \operatorname{dist}(\mathbf{0}, \nabla_x f(x, y) + \partial \mathbf{1}_{\mathcal{X}}(x))$$

*holds for any $y \in \mathcal{Y}$ and $x \in \mathcal{X}$.*

**Assumption 5 (Convexity of the primal function)** *For any fixed point $y \in \mathcal{Y}$, $f(\cdot, y)$ is convex.*

**Proposition 6 (Proximal error bound)** *Under the setting of Proposition 5 with Assumption 4 or 5, for any $z \in \mathbb{R}^n, v \in \mathbb{R}^d$, we have*
(i) KŁ exponent $\theta_1 \in (0, 1)$:

$$\|y(z_+^t(v^{t+1}), v^{t+1}) - y(z(v^{t+1}), v^{t+1})\|^2 \leq \omega_2 \|z_+^t(v^{t+1}) - z^t\|^{\frac{1}{\theta_1}};$$

(ii) Convex:

$$\|y(z_+^t(v^{t+1}), v^{t+1}) - y(z(v^{t+1}), v^{t+1})\|^2 \leq \omega_3 \|z_+^t(v^{t+1}) - z^t\|,$$

*where $\omega_2 := \frac{2}{(r_2 - L_y)\tau_1} \left( \frac{r_1(1-\beta)}{\beta} + \frac{r_1^2}{r_1 - L_x} \right)^{\frac{1}{\theta_1}}$ and $\omega_3 := \frac{4r_1 \operatorname{diam}(\mathcal{X})}{r_2 - L_y} \left( \frac{1-\beta}{\beta} + \sigma_2 \right)$. Here, $\operatorname{diam}(\mathcal{X})$ denotes the diameter of the set $\mathcal{X}$.*

Equipped with these results, we could also derive similar complexity results for (non)convex-nonconcave minimax problems as Theorem 1.

# I  Proof of Theorem 2

In general, analyzing the local convergence rate of sequences could be challenging. Fortunately, when $\theta \in (0, \frac{1}{2}]$, the homogeneous sufficient descent property holds as follows:

$$\Phi(x^t, y^t, z^t, v^t) - \Phi(x^{t+1}, y^{t+1}, z^{t+1}, v^{t+1})$$
$$\geq \frac{r_1}{40}\|x^{t+1} - x^t\|^2 + \frac{r_2}{20}\|y^t - y_+^t(z^t, v^t)\|^2 + \frac{r_1}{4\beta}\|z^t - z^{t+1}\|^2 + \frac{6r_2}{25\mu}\|v_+^t(z^{t+1}) - v^t\|^2.$$

Consequently, we can use the unified analysis framework in [3]. Next, we prove the "Safeguard" property as required there.

**Proposition 7 ("Safeguard" property)** *There exists a series of parameters $\nu_1, \nu_2, \nu_3, \nu_4 > 0$ such that for all $t \in \{0, 1, \ldots, T - 1\}$, the following holds:*

$$\operatorname{dist}(\mathbf{0}, \partial \Phi(x^{t+1}, y^{t+1}, z^{t+1}, v^{t+1}))$$
$$\leq \nu_1 \|x^t - x^{t+1}\| + \nu_2 \|y^t - y_+^t(z^t, v^t)\| + \nu_3 \|v^t - v_+^t(z^{t+1})\| + \nu_4 \|z^t - z^{t+1}\|.$$

**Proof** We observe from the update of $x^{t+1} = \operatorname{argmin}_{x \in \mathcal{X}} \frac{1}{2}\|x - (x^t - c\nabla_x F(x^t, y^t, z^t, v^t))\|^2$ and $y^{t+1} = \operatorname{argmin}_{y \in \mathcal{Y}} \frac{1}{2}\|y - (y^t + \alpha\nabla_y F(x^{t+1}, y^t, z^t, v^t))\|^2$ that

$$\begin{aligned} \mathbf{0} &\in x^{t+1} - x^t + c\nabla_x F(x^t, y^t, z^t, v^t) + \partial \mathbf{1}_{\mathcal{X}}(x^{t+1}), \\ \mathbf{0} &\in y^{t+1} - y^t - \alpha\nabla_y F(x^{t+1}, y^t, z^t, v^t) + \partial \mathbf{1}_{\mathcal{Y}}(y^{t+1}). \end{aligned} \tag{25}$$

Moreover, we have

$$\begin{aligned} &\partial \Phi(x^{t+1}, y^{t+1}, z^{t+1}, v^{t+1}) \\ =\; &\{\nabla_x F(x^{t+1}, y^{t+1}, z^{t+1}, v^{t+1}) + \partial \mathbf{1}_{\mathcal{X}}(x^{t+1})\} \times \\ &\{\nabla_y F(x^{t+1}, y^{t+1}, z^{t+1}, v^{t+1}) - 2\nabla_y d(y^{t+1}, z^{t+1}, v^{t+1}) + \partial \mathbf{1}_{\mathcal{Y}}(y^{t+1})\} \times \\ &\{\nabla_z F(x^{t+1}, y^{t+1}, z^{t+1}, v^{t+1}) - 2\nabla_z d(y^{t+1}, z^{t+1}, v^{t+1}) + 2\nabla_z q(z^{t+1})\} \times \\ &\{\nabla_v F(x^{t+1}, y^{t+1}, z^{t+1}, v^{t+1}) - 2\nabla_v d(y^{t+1}, z^{t+1}, v^{t+1})\}. \end{aligned}$$

This, together with (25), gives

$$\text{dist}(\mathbf{0}, \partial\Phi(x^{t+1}, y^{t+1}, z^{t+1}, v^{t+1}))$$
$$\leq \left\| \frac{x^t - x^{t+1}}{c} - \nabla_x F(x^t, y^t, z^t, v^t) + \nabla_x F(x^{t+1}, y^{t+1}, z^{t+1}, v^{t+1}) \right\| +$$
$$\left\| \frac{y^t - y^{t+1}}{\alpha} + \nabla_y F(x^{t+1}, y^t, z^t, v^t) + \nabla_y F(x^{t+1}, y^{t+1}, z^{t+1}, v^{t+1}) - 2\nabla_y d(y^{t+1}, z^{t+1}, v^{t+1}) \right\| +$$
$$\| r_1(z^{t+1} - x^{t+1}) - 2r_1(z^{t+1} - x(y^{t+1}, z^{t+1}, v^{t+1})) + 2r_1(z^{t+1} - x(z^{t+1}, v(z^{t+1}))) \| +$$
$$\| r_2(y^{t+1} - v^{t+1}) \|.$$

(26)

Under Assumption 1 and utilizing the error bounds in Lemmas 2 and 4, we can simplify (26). When $z^t$ reaches the local region that satisfies $\beta \| x(z^t, v^{t+1}) - z^t \| \leq 1$, we can show that the right-hand side is actually upper bounded by the linear combination of the desired four terms:

$$\text{dist}(\mathbf{0}, \partial\Phi(x^{t+1}, y^{t+1}, z^{t+1}, v^{t+1}))$$
$$\leq \left( \frac{1}{c} + L_x + r_1 \right) \| x^t - x^{t+1} \| + L_x \| y^t - y^{t+1} \| + r_1 \| z^t - z^{t+1} \| + \left( \frac{1}{\alpha} + L_y + r_2 \right) \| y^t - y^{t+1} \| +$$
$$2L_x \| x^{t+1} - x(y^{t+1}, z^{t+1}, v^{t+1}) \| + r_2 \| v^t - v^{t+1} \| + r_1 \| z^{t+1} - x^{t+1} \| +$$
$$2r_1 \| x(y^{t+1}, z^{t+1}, v^{t+1}) - x(z^{t+1}, v(z^{t+1})) \| + r_2 \| y^{t+1} - v^{t+1} \|$$
$$\leq \nu_1 \| x^t - x^{t+1} \| + \nu_2 \| y^t - y_+^t(z^t, v^t) \| + \nu_3 \| z^t - z_+^t(v^{t+1}) \| + \nu_4 \| v^t - v^{t+1} \|.$$

Here, $\nu_1 = \frac{1}{c} + L_x + r_1 + 2L_x\sigma_6 + 2r_1\sigma_1 L_y\alpha\sigma_6 + (L_x + \frac{1}{\alpha} + L_y + 2r_2 + 2L_x\sigma_1)L_y\alpha\sigma_6$, $\nu_2 = L_x + \frac{1}{\alpha} + L_y + r_2 + 2L_x\sigma_1 + r_1(2\sigma_1 + 1)(\sigma_8 + 1)$, $\nu_3 = \frac{r_1}{\beta} + 2L_x\sigma_2 + 2r_1\sigma_1\sigma_3 + r_2\sigma_3$, and $\nu_4 = \frac{r_2}{\mu} + 2r_1\sigma_1\sigma_5 + 2r_1\sqrt{\omega_1}$. Moreover, since $f(\cdot, y)$ and $f(x, \cdot)$ is semi-algebraic [10, Theorem 3], using results in [10, Example 2], we find that $\Phi(x, y, z, v) = F(x, y, z, v) - 2d(y, z, v) + 2q(z)$ is also semi-algebraic in $x, y, z, v$. Based on Assumption 1, $\Phi(x, y, z, v)$ is continuous in $x, y, z, v$. Building on the unified convergence analysis framework in [4], we can conclude that $\{(x^t, y^t, z^t, v^t)\}$ converges to one stationary point $(x^*, y^*, z^*, v^*)$. On top of Lemma 8 and $\lim_{t\to\infty} \| x^t - x^{t+1} \| = 0$, $\lim_{t\to\infty} \| y^t - y^{t+1} \| = 0$, we have $\{(x^t, y^t)\}$ actually converges to a GS. For $z^t$, we have

$$\| z^t - x^*(z^t) \|$$
$$\leq \| z^t - x^{t+1} \| + \| x^{t+1} - x(y^t, z^t, v^t) \| + \| x(y^t, z^t, v^t) - x(z^t, v^t) \| + \| x(z^t, v^t) - x^*(z^t) \|$$
$$\leq \frac{\| z^t - z^{t+1} \|}{\beta} + \sigma_6 \| x^t - x^{t+1} \| + \sigma_1\sigma_8 \| y^t - y_+^t(z^t, v^t) \| + \omega_2 \| v^t - y(z^t, v^t) \|$$
$$\leq \frac{\| z^t - z^{t+1} \|}{\beta} + \sigma_6 \| x^t - x^{t+1} \| + \sigma_1\sigma_8 \| y^t - y_+^t(z^t, v^t) \| + \omega_2 \left( \frac{\| v^t - v_+^t(z^{t+1}) \|}{\mu} + \right.$$
$$\left. \sigma_3 \| z^t - z^{t+1} \| + 2L_y\alpha\sigma_6 \| x^{t+1} - x^t \| + 2(\sigma_8 + 1)\| y_+^t(z^t, v^t) - y^t \| \right).$$

Since $\lim_{t\to\infty} \| z^t - z^{t+1} \| = 0$, $\lim_{t\to\infty} \| x^t - x^{t+1} \| = 0$, $\lim_{t\to\infty} \| y^t - y_+^t(z^t, v^t) \| = 0$, $\lim_{t\to\infty} \| v^t - v_+^t(z^{t+1}) \| = 0$, we conclude $\{z^t\}$ convergs to an OS. $\square$

## J  Proof of Theorem 3

Here, without the generality, we assume $L_x = L_y = L$ and $r_1 = r_2 = r$. Then from the Proposition 1 and Proposition 5, we can easily derive the bounds for all the parameters as follows:

$$r \geq 2L, 0 < c = \alpha = \min\left\{ \frac{4}{3(L+r)}, \frac{1}{6L_d}, \frac{1}{5\sqrt{6}L} \right\} < 1, \alpha \geq \frac{3L(2\alpha r + 1)^2}{2(r - L)^2}, \tag{27a}$$

$$0 < \beta = \mu \leq \min\left\{ \frac{24r}{360r + 5r^2 + (2L + 5r)^2}, \frac{cL^2}{9216r}, \frac{12}{12 + L^2} \right\} \leq \mathcal{O}(T^{-\frac{1}{2}}). \tag{27b}$$

There are numerous feasible choices for $\beta$ and $\mu$ that satisfy equation (27b). Next, we will demonstrate the existence of a viable solution for $c$ and $\alpha$ by establishing the intersection of the last two constraints in equation (27a). Solving these constraints under the condition $r \geq 2L$ yields

$$\frac{-(r-L)\sqrt{L^2 - 14Lr + r^2} + L^2 - 8Lr + r^2}{12Lr^2} \leq c = \alpha \leq \frac{(r-L)\sqrt{L^2 - 14Lr + r^2} + L^2 - 8Lr + r^2}{12Lr^2}$$

where $r$ should further satisfy $r \geq 14L$. Combing these with the penultimate constraint in equation (27a), we can provide a set of feasible parameters as follows:

$$r \geq 20L, \quad \frac{-(r-L)\sqrt{L^2 - 14Lr + r^2} + L^2 - 8Lr + r^2}{12Lr^2} \leq c = \alpha \leq \frac{1}{6L_d}. \tag{28}$$

Armed with equation (27b) and equation (28), we finish the proof of Theorem 3.

# K   Relationship Between Different Notions of Stationary Points

In this section, we illustrate the quantitative relationship among several notions of stationary measure.

**Definition 2** *The point $(x, y) \in \mathcal{X} \times \mathcal{Y}$ is said to be an*

- *$\epsilon$-optimization game stationary point* (OS) *if*

$$\| \operatorname{prox}_{\frac{\max_{y \in \mathcal{Y}} f(\cdot, y)}{r_1} + \mathbf{1}_{\mathcal{X}}}(x) - x \| \leq \epsilon;$$

- *$\epsilon$-game stationary point* (GS) *if*

$$\operatorname{dist}(\mathbf{0}, \nabla_x f(x, y) + \partial \mathbf{1}_{\mathcal{X}}(x)) \leq \epsilon, \operatorname{dist}(\mathbf{0}, -\nabla_y f(x, y) + \partial \mathbf{1}_{\mathcal{Y}}(y)) \leq \epsilon.$$

Recalling that $x^*(z) = \operatorname{argmin}_{x \in \mathcal{X}} \max_{y \in \mathcal{Y}} f(x, y) + \frac{r_1}{2}\|x - z\|^2$, we have $\| \operatorname{prox}_{\frac{\max_{y \in \mathcal{Y}} f(\cdot, y)}{r_1} + \mathbf{1}_{\mathcal{X}}}(x) - x \| = \|x^*(x) - x\|$. With Proposition 3, we are ready to prove the relationship between OS and GS.

**Proposition 8** *Suppose that Assumption 2 holds. If $(x, y) \in \mathcal{X} \times Y$ is a $\epsilon$-GS, then it is a $\mathcal{O}(\epsilon^{\min\{1, \frac{1}{2\theta}\}})$-OS.*

Building on the proposition 3, we have

$$\begin{aligned}
\|x^*(x) - x\| &\leq \|x^*(x) - x(x, y)\| + \|x(x, y) - x\| \\
&\leq \omega_2 \|y - y(x, y)\|^{\frac{1}{2\theta}} + \|x(x, y) - x\| \\
&\leq \omega_2 \|y - y(x, y)\|^{\frac{1}{2\theta}} + \|x(y, x, v) - x\| + \sigma_1 \|y - y(x, y)\|.
\end{aligned} \tag{29}$$

By the nonexpansiveness of projection operator and error bounds in Lemmas 3 and 4, we can further bound $\|y - y(z, v)\|$ as follows:

$$\begin{aligned}
&\|y - y(z, v)\| \leq \sigma_8 \|y - y_+(z, v)\| \\
&= \sigma_8 \|y - \operatorname{proj}_{\mathcal{Y}}(y + \alpha \nabla_y F(x(y, z, v), y, z, v))\| \\
&\leq \sigma_8 \|y - \operatorname{proj}_{\mathcal{Y}}(y + \alpha \nabla_y F(x, y, z, v))\| + \\
&\quad \sigma_8 \| \operatorname{proj}_{\mathcal{Y}}(y + \alpha \nabla_y F(x, y, z, v)) - \operatorname{proj}_{\mathcal{Y}}(y + \alpha \nabla_y F(x(y, z, v), y, z, v))\| \\
&\leq \sigma_8 \|y - \operatorname{proj}_{\mathcal{Y}}(y + \alpha \nabla_y F(x, y, z, v))\| + L_y \alpha \sigma_8 \|x - x(y, z, v)\| \\
&\leq \sigma_8 \|y - y(x, z, v)\| + \sigma_8 \|y(x, z, v) - \operatorname{proj}_{\mathcal{Y}}(y + \alpha \nabla_y F(x, y, z, v))\| + L_y \alpha \sigma_8 \|x - x(y, z, v)\| \\
&\leq \sigma_8 \|y - y(x, z, v)\| + \sigma_8 \left(1 + \alpha L_y + \alpha r_2\right) \|y(x, z, v) - y\| + L_y \alpha \sigma_8 \|x - x(y, z, v)\| \\
&\leq L_y \alpha \sigma_8 \|x - x(y, z, v)\| + \left(2 + \alpha L_y + \alpha r_2\right) \sigma_8 \|y - y(x, z, v)\|.
\end{aligned}$$

Plugging this bound into (29), we get

$$\begin{aligned}
\|x^*(x) - x\| &\leq \sigma_8 \sigma_1 \left(2 + \alpha L_y + \alpha r_2\right) \|y - y(x, x, y)\| + \left(L_y \alpha \sigma_8 \sigma_1 + 1\right) \|x - x(y, x, y)\| + \\
&\quad \omega_2 \left(L_y \alpha \sigma_8 \|x - x(y, x, y)\| + \left(2 + \alpha L_y + \alpha r_2\right) \sigma_8 \|y - y(x, x, y)\|\right)^{\frac{1}{2\theta}}.
\end{aligned}$$

Next, we will explore the relationship between $\|x - x(y, x, v)\|$ and $\text{dist}(\mathbf{0}, \nabla_x f(x, y) + \partial \mathbf{1}_{\mathcal{X}}(x))$. Let $x_+(y, x, v) := \text{proj}_{\mathcal{X}}(x - c\nabla_x F(x, y, x, v))$. Then, from the primal error bound (see [49]), we know that

$$\|x - x(y, x, v)\| \leq \frac{cL_x + cr_1 + 1}{cr_1 - cL_x}\|x - x_+(y, x, v)\|.$$

Moreover, since $\nabla_x F(x, y, x, v) = \nabla_x f(x, y)$, it follows from [39, Lemma 4.1] that

$$\begin{aligned}
\|x - x(y, x, v)\| &\leq \frac{cL_x + cr_1 + 1}{cr_1 - cL_x}\|x - x_+(y, x, v)\| \\
&= \frac{cL_x + cr_1 + 1}{cr_1 - cL_x}\|x - \text{proj}_{\mathcal{X}}(x - c\nabla_x f(x, y))\| \\
&\leq \frac{cL_x + cr_1 + 1}{r_1 - L_x}\,\text{dist}(\mathbf{0}, \nabla_x f(x, y) + \partial \mathbf{1}_{\mathcal{X}}(x)).
\end{aligned}$$

A similar analysis can be applied to derive the bounds for $\|y - y(x, z, y)\|$. Thus, if $(x, y)$ is a $\epsilon$-GS, then it is an $\mathcal{O}(\epsilon^{\min\{1, \frac{1}{2\theta}\}})$-OS.

# L  Details about Examples in Section 4.3 and Behaviors of Wrong Selection of Smoothing Sides in S-GDA

This section will check all the regularity conditions for the examples mentioned in Section 4.3. The violation of the global KŁ condition can be easily vertified by plotting the figures for these examples. Therefore, we mainly discuss whether they satisfy "weak MVI" and "$\alpha$-interaction dominant" conditions, which are two representative classes of conditions in the nonconvex-nonconcave setting. Moreover, we will also give an example showing the slow convergence caused by the wrong selection of S-GDA.

## L.1  Proof of Proposition for "Forsaken" Example

This subsection considers the "Forsaken" example in [32, Example 5.2] on the constraint sets $\mathcal{X} = \mathcal{Y} = \{z : -1.5 \leq z \leq 1.5\}$. In [50], it is vertified that the "Forsaken" example violates "weak MVI" condition with $\rho < -\frac{1}{2L}$. Therefore, we only consider the $\alpha$-interaction dominant condition here. By a simple calculation, we get $\nabla_{xx}^2 f(x, y) = \frac{1}{2} - 6x^2 + 5x^4$, $\nabla_{xy}^2 f(x, y) = \nabla_{yx}^2 f(x, y) = 1$, and $\nabla_{yy}^2 f(x, y) = -\frac{1}{2} + 6y^2 - 5y^4$. Armed with these, $\alpha$ can be found globally by minimizing the following:

$$\nabla_{xx}^2 f(x, y) + \nabla_{xy}^2 f(x, y)(\eta\mathbf{1} - \nabla_{yy}^2 f(x, y))^{-1}\nabla_{yx}^2 f(x, y)$$
$$= \frac{1}{2} - 6x^2 + 5x^4 + \left(\eta + \frac{1}{2} - 6y^2 + 5y^4\right)^{-1}.$$

It is less than zero when $[x; y] = [1; 0]$. That is, $\alpha < 0$ in the constraint set $\mathcal{X}$, which means the $\alpha$-interaction dominant condition is violated for the primal variable $x$. Similar proof could be adapted for the dual variable. This rules out the convergence guarantees of damped PPM, which is validated in Figure 4.

## L.2  Proof of Proposition for "Bilinearly-coupled Minimax" Example

This "Bilinearly-coupled Minimax" example is mentioned as a representative example where $\alpha$ is in the interaction moderate regime [29]. Experiments also validate that the solution path will be globally trapped into a limit cycle (see Figure 4). For this reason, we only check the "weak MVI" condition. In this example, $\mathcal{X} = \mathcal{Y} = \{z : -4 \leq z \leq 4\}$, $G(u) = [\nabla_x f(x, y); -\nabla_y f(x, y)] = [4x^3 - 20x + 10y; 4y^3 - 20y - 10x]$, and $u^\star = [0; 0]$. Then, $\rho$ can be found by globally minimizing $\rho(u) := \frac{\langle G(u), u - u^\star\rangle}{\|G(u)\|^2}$ for all $u \in \mathcal{X} \times \mathcal{Y}$. Notice that

$$\frac{\langle G(u), u - u^\star\rangle}{\|G(u)\|^2} = \frac{x^4 + y^4 - 5x^2 - 5y^2}{(2x^3 - 10x + 5y)^2 + (2y^3 - 10y - 5x)^2}.$$

We have $\rho(u) = \frac{\langle G(u), u - u^\star\rangle}{\|G(u)\|^2} = -\frac{4}{89}$ when $u = [x; y] = [0; 1]$, which implies that $\rho < -\frac{4}{89}$. Moreover, we find $L = 172$, so $\rho < -\frac{4}{89} < -\frac{1}{344} = -\frac{1}{2L}$. We conclude that this example does not satisfy the "weak MVI" condition and the limit cycle phenomenon is actually observed in Figure 4.

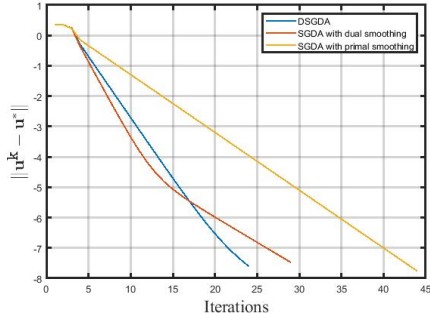

Figure 5: Converegnce behavior of DS-GDA and different smoothing sides for S-GDA.

### L.3 Proof of Proposition for "Sixth-order polynomial" Example

We demonstrate that this example violates both the 'weak MVI' and '$\alpha$-interaction' conditions. To provide evidence of this violation, we follow the same approach used to prove violations in Sections L.1 and L.2 and present counter-examples. On the one hand, we found that $\rho \leq \rho(\hat{u}) = -0.0795 < -0.0368 = -\frac{1}{2L}$ with $\hat{u} = [x; y] = [-1; 0.5]$, which implies the violation of "weak MVI" condition. On the other hand, we have

$$\nabla_{xx}^2 f(x,y) + \nabla_{xy}^2 f(x,y)(\eta \mathbf{1} - \nabla_{yy}^2 f(x,y))^{-1} \nabla_{yx}^2 f(x,y) \Big|_{[x;y]=[0;1]}$$

$$= (21609 \exp(-1/50))/(625(\eta + (77061 * \exp(-1/100))/25000)) - (4989 \exp(-1/100))/500.$$

It is less than zero when $\eta \geq L$, which indicates the violation of the $\alpha$-interaction condition.

### L.4 Proof of Proposition for "PolarGame" Example

In this subsection, we check the two conditions for the "PolarGame" example. Firstly, we have $\rho \leq \rho(\hat{u}) = -0.3722 < -0.0039 = -\frac{1}{2L}$ with $\hat{u} = [0.8; 0]$. Thus, it does not satisfy the "weak MVI" condition. Next, consider the following at $[x; y] = [0.8; 0]$:

$$\nabla_{xx}^2 f(x,y) + \nabla_{xy}^2 f(x,y)(\eta \mathbf{1} - \nabla_{yy}^2 f(x,y))^{-1} \nabla_{yx}^2 f(x,y) \Big|_{[x;y]=[0.8;0]}$$

$$= 1/(\eta - 279/625) - 779/125.$$

It is less than zero when $\eta \geq L$. Thus, the "PolarGame" violates the $\alpha$-interaction condition.

### L.5 Wrong Smoothing Side of S-GDA

For S-GDA, we show that if we choose the wrong side, it will result in a slow convergence. To validate this, we conduct a new experiment for the KL-NC problem

$$\min_{x \in \mathcal{X}} \max_{y \in \mathcal{Y}} f(x,y) = 2x^2 - y^2 + 4xy^6 + \frac{4y^3}{3} - \frac{y^4}{4},$$

where $\mathcal{X} = \mathcal{Y} = \{z : -1 \leq z \leq 1\}$. With a wrong smoothing side, S-GDA (with primal smoothing) leads to a slower convergence compared with dual smoothing (see Figure 5).

