# OpenReview forum: "Universal Gradient Descent Ascent Method for Nonconvex-Nonconcave Minimax Optimization"
_NeurIPS.cc/2023/Conference — NeurIPS 2023 poster_

### Official Review · Reviewer_KYPK · 2023-06-25

**Soundness:** 3 good
**Presentation:** 2 fair
**Contribution:** 3 good
**Rating:** 6
**Confidence:** 3

**Summary:**

The paper proposes the doubly smoothed gradient descent ascent method and studies its convergence under nonconvex-KL and nonconvex-concave minimax problems.

**Strengths:**

This paper proposes a novel model to address the limiting cycle phenomena in nonconvex-nonconcave minimax optimization. The theoretal analysis match the best-known results for other methods.

**Weaknesses:**

The algorithm proposed in this paper comes naturally. From my experience, I can expect that DSGDA performs well in practice. My main concern is the theoretical guarantee of DSGDA. However, it seems that the results given by this paper do not show any superiority of DSGDA compared with other methods.


**Questions:**

Are there any differences between the convergence of Doubly Smoothed GDA and Smoothed GDA (Zhang et.al 2020) / Smoothed PLDA (Li et. al 2022) in theory? From the theoretical viewpoint, why is  Doubly Smoothed GDA better than Smoothed GDA? Can we show that the Doubly Smoothed GDA provably gets rid of limiting cycles?

The authors claimed that "our work demonstrates, for the first time, the possibility of having a
simple and unified single-loop algorithm for solving nonconvex-nonconcave, nonconvex-concave,
and convex-nonconcave minimax problems."  Can the authors explain that why the Smoothed GDA cannot have such a "unified analysis"?
By the way, I think it is better to use "nonconvex-KL" instead of "nonconvex-nonconcave" since the difficulties of these two problems seem to be very different from my point of view.

---

> ### Author Rebuttal · Authors · 2023-08-10
>
> Thanks for your comments. We hope the clarification can clear your concerns and may reevaluate our contribution.
>
> **Q1: Superiority of DSGDA compared with other methods.**
>
> DSGDA can directly achieve both $\epsilon$-GS and $\epsilon$-OS points at the same rate.  Furthermore, its convergence rate stands out as the fastest among the current body of literature, aligning with the results of SGDA. For the superiority of DSGDA compared with SGDA, please refer to the global response to Q2.
>
> **Q2: Differences between the convergence of DSGDA and SGDA (Zhang et al. 2020)/Smoothed PLDA (Li et al. 2022) in theory. Why is DSGDA better than SGDA? Can we show that the DSGDA provably gets rid of limiting cycles?**
>
> - The introduction of double extrapolation in DSGDA fundamentally differentiates it from SGDA. Several key aspects highlight these differences. 1. The previously used Lyapunov function in SGDA is no longer applicable and a novel Lyapunov function tailored for DSGDA is constructed (see line 198). 2. To establish the sufficient decrease property of this newly proposed Lyapunov function, the derivation of a new proximal error bound is required (refer to Proposition 1). 3. Conceptually, due to the two extrapolation steps, extending the proof technique from SGDA to DSGDA presents significant challenges in primal-dual balancing. To be specific, our theoretical analysis, as shown in Theorem 1, two additional parameters, $r_2$ and $\mu$, make it difficult to provide explicit bounds for all hyperparameters. In SGDA, since there is only one-sided extrapolation, the relationship between different parameters is rather simple. The step-sizes can be sufficiently small.  While in DSGDA, due to the double extrapolation, their relationship is complicated and the step-sizes will have non-trivial lower bounds. Although the algorithmic extension of DSGDA may seem intuitive and straightforward, we firmly believe that substantial effort and theoretical development are required to establish the global convergence of our proposed DSGDA method. The complexities introduced by double extrapolation demand rigorous investigation and are the focus of our research efforts.
>
> - From the theoretical viewpoint, DSGDA is the first simple and universal single-loop algorithm for solving NC-NC, NC-C, and C-NC minimax problems. SGDA does not have such "unified analysis". Please refer to the global response to Q1.
>
> - Yes, we believe DSGDA can be proven for getting rid of the limit cycles. Regarding the gap between the practical claim and the theoretical claim, please refer to the global response to Q3.
>
> **Q3: Can the authors explain why the SGDA cannot have such a "unified analysis"? It is better to use "NC-KL" instead of "NC-NC" since the difficulties of these two problems seem to be very different.**
>
> - For the SGDA proposed in [1] and [2], it can only solve concave (or NC-KL) problems. Further changes in algorithms are needed for solving KL-NC problems. Please refer to the global response to Q1 for details.
>
> - Intuitively, DSGDA is designed to address general NC-NC problems. Although our theoretical analysis currently focuses on cases that satisfy one-sided KL (or convex/concave) problems, we observe that DSGDA effectively escapes the limit cycle in numerous challenging NC-NC examples without any regularity condition (see Section 2). Moreover, our algorithm can also be applied to NC-C problems, which may not satisfy the KL property (see counterexamples in Corollary 4 of [3]). It may be better to use "a class of Constrained Nonconvex-Nonconcave Minimax Optimization" instead.
>
> [1]. Zhang J, Xiao P, Sun R, et al. A single-loop smoothed gradient descent-ascent algorithm for nonconvex-concave min-max problems.
>
> [2]. Li J, Zhu L, So A M C. Nonsmooth Composite Nonconvex-Concave Minimax Optimization.
>
> [3]. Bolte J, Pauwels E. Curiosities and counterexamples in smooth convex optimization.

---

> > ### Comment · Reviewer_KYPK · 2023-08-12
> > **Thanks a lot for helping me understanding this paper**
> >
> > The authors' rebuttal helps me further understand the contribution and technical challenges in this paper.
> >
> > I now think that this article has made a sufficient theoretical contribution and shed light on the nonconvex-nonconcave minimax problems. I think many useful insights provided in this paper would be of great help for future research on nonconvex-nonconcave minimax problems.
> > I now like this paper very much. I will be glad if I see this paper published in NeurIPS 2023.
> >
> > I decide to raise my score to 6.
> >
> > By the way, I agree with reviewer gc6W that it would be better to use  “S-GDA” / “DS-GDA” instead of "SGDA" / "DSGDA to avoid possible confusion.

---

> > > ### Author Response · Authors · 2023-08-12
> > > **Thanks for your response**
> > >
> > > Thanks for your response. We are delighted to hear you are willing to re-evaluate our work and happy to see you appreciate our work.  For the abbreviation of S-GDA/DS-GDA, we will revise it in the updated version.

---

### Official Review · Reviewer_71Wf · 2023-07-06

**Soundness:** 3 good
**Presentation:** 3 good
**Contribution:** 2 fair
**Rating:** 6
**Confidence:** 3

**Summary:**

This paper addresses the nontrivial general minimax problem by proposing a novel algorithm called Doubly Smoothed Gradient Descent Ascent (DSGDA). The authors provide both experimental and theoretical evidence to support their claims.

**Strengths:**

1. Overview I enjoyed reading this work. It's well written and organized, although some refernece and comparison is missing (see below).

2. The proposed algorithm is not complicated and the intuition is sound.

3.  Convergence results are sound.


**Weaknesses:**


While I acknowledge the soundness of the algorithm and the obtained results, I do have several concerns about this paper.

### W1:
The theoretical convergence results demonstrating that the proposed method converges to a stationary point do not necessarily showcase its effectiveness in addressing the issue of **general** minimax problems. Although it is generally difficult for first-order methods to derive other convergence properties, such as local minimax points, for nonconvex-nonconcave problems, could the authors show the superiority of the algorithm under other settings?

### W2:
The authors claim that their work demonstrates, for the first time, the possibility of having a simple and unified single-loop algorithm for solving nonconvex-nonconcave, nonconvex-concave, and convex-nonconcave minimax problems. However, to my knowledge, GDA-AM (ICLR 2022) also demonstrated such theoretical results (Thm C.4 and C.9) and showed convergent experiments. Additionally, the fast extragradient (NeurIPS 2021) made similar claims, as mentioned in Remark 1. Could the authors discuss the differences in contributions between their work and these existing papers?

### W3:
Experiments
The authors only compared their algorithm with SGDA. However, what about other methods? For example, how does it compare to fast extragradient or GDA-AM on these problems?

1. He et.al. GDA-AM: On the effectiveness of solving minimax optimization via Anderson Acceleration, ICLR 2022
2. Lee et.al. Fast Extra Gradient Methods for Smooth Structured Nonconvex-Nonconcave Minimax Problems, Neurips 2021


Except several concerns unaddressed, I tend to accept. I look forward to the authors' response.

**Questions:**

### Explainantion on Theorem 1
Could authors make additional comments are conditions of theorem 1.



**Limitations:**

Authors discussed limitations

---

> ### Author Rebuttal · Authors · 2023-08-10
>
> Thanks for your comments. We are happy that the reviewer enjoys our paper and would like to thank you for your insightful comments. Here are some illustrations based on your questions/suggestions.
>
> **Q1: Superiority of the algorithm under other settings.**
>
> We kindly request the reviewer to confirm whether we misunderstand your question. We answer it in the following two possible directions. We'd be happy to clarify and answer your follow-up questions during the discussion period.
> * For general C-NC and NC-C problems, we have shown our algorithm can achieve both $\epsilon$-game stationary (GS) and $\epsilon$-optimization stationary (OS) points. They are the best points we can get in these settings and similar results can be obtained by SGDA. For C-C problems, we believe DSGDA can achieve Nash equilibrium point and we intend to explore it in future research.
> * For the superiority of DSGDA on C-NC and NC-C problems, our convergence rate matches the results of SGDA, which is the SOTA. For the superiority compared with SGDA, please refer to the global response to Q1.
>
> **Q2: Differences with GDA-AM (ICLR 2022) and fast extragradient (NeurIPS 2021).**
>
> Thanks for pointing out two useful references. We believe our proposed DSGDA is fundamentally different from GDA-AM and fast extra-gradient. Here are detailed comparisons:
>
> **For GDA-AM:** GDA-AM only enjoys the global convergence guarantee for **unconstrained** C-C and C-NC problems. We have gone through the proof details, to the best of our knowledge, GDA-AM cannot handle the constrained case and its extension to the constrained scenario is highly non-trivial. Even under the unconstrained setting, the convergence of NC-NC problems is unsolved. As far as we know, no first-order method can converge for unconstrained NC-NC problems. We conduct a new experiment for the ``sixth-order polynomial'' example with initialization $(x,y)=(15,15)$, none of the first-order methods (SimGDA, AltGDA, OMD, EG, DSGDA, AltGDA-RAM, SimGDA-RAM) converge (refer to Figure (h) in **PDF**). For constrained NC-NC problems that we studied in the paper, the applicability of GDA-AM remains uncertain. In contrast, we have demonstrated that our algorithm effectively eliminates limit cycles in challenging constrained NC-NC scenarios, accompanied by theoretical guarantees for KL-NC and NC-KL settings.
>
> **For fast extra-gradient:** Firstly, it is only shown to converge under the negative comonotone condition, which is a stronger assumption than weak MVI. Such an assumption is restrictive and general C-NC/NC-C problems are easy to violate it. For example, the violations of this condition for C-NC problem $\min_{x\in \mathcal{X}}\max_{y\in \mathcal{Y}}f(x,y)=2x-y^2+4xy^6$ with $\mathcal{X}=\mathcal{Y}=\\{z:-1\leq z\leq 1\\}$ can be checked. Secondly, the convergence for C-NC/NC-C problem is unaddressed in this paper (i.e., NC-C/C-NC problems are not necessary to satisfy the negative comonotone condition), while DSGDA enjoys the global convergence with an iteration complexity of $\mathcal{O}(\epsilon^{-4})$. Finally, CurvatureEG+, an algorithm that converges under the weak MVI condition, is compared with DSGDA in Section 2. We can observe that CurvatureEG+ diverges or falls into a limit cycle for many examples. Given that weak MVI represents the weakest VI condition, we can reasonably anticipate similar outcomes for the fast extra-gradient method.
>
> **Q3: Experiments for GDA-AM and fast extragradient.**
>
> GDA-AM cannot directly apply to constrained problems and the fast extra gradient will have similar performance with CurvatureEG+. Please refer to the response to Q2 for details.
>
> **Q4: Explanation on Theorem 1: Could authors make additional comments are conditions of theorem 1.**
>
> Please refer to the global response to Q2.
>
> **Missing reference**
>
> Thanks for your advice. We would add the missing reference in the updated version.

---

> > ### Comment · Reviewer_71Wf · 2023-08-13
> > **Thanks for the clarification**
> >
> > Thank the authors for their comprehensive response. It helped me to understand the contributions of DSGDA. I will read two references again and more closely. Will come back later.

---

> > > ### Comment · Reviewer_71Wf · 2023-08-14
> > >
> > > I appreciate the authors' additional discussion on differences from other existing works. The proposed method can solve more complicated problems in the literature.  I decide to raise my score to 6.

---

### Official Review · Reviewer_gc6W · 2023-07-06

**Soundness:** 3 good
**Presentation:** 3 good
**Contribution:** 3 good
**Rating:** 6
**Confidence:** 4

**Summary:**

This paper introduces a novel single-loop algorithm called the doubly smoothed gradient descent ascent method (DSGDA). DSGDA effectively balances the primal and dual updates, eliminating limit cycles in various challenging nonconvex-nonconcave scenarios in the literature. The paper establishes that under a one-sided Kurdyka-Łojasiewicz condition, with an exponent $\theta\in (0,1)$ (or for convex primal/concave dual functions), DSGDA can discover a game-stationary point with an iteration complexity of $O(\epsilon^{-2\max\{(2\theta, 1)\}})$ (or $O(\epsilon^{-4})$), respectively. These complexity results match the best outcomes achieved by single-loop algorithms for solving nonconvex-concave or convex-nonconcave minimax problems, as well as problems satisfying the restrictive one-sided Polyak-Łojasiewicz condition.


**Strengths:**

**Originality**: The paper provides a comprehensive discussion of existing algorithms for mini-max optimization problems and introduces a novel algorithm aimed at avoiding limit cycles. The proposed algorithm is supported by both theoretical results and empirical validation, strengthening its efficacy.

**Quality and Clarity**: The paper exhibits a high level of writing quality, featuring a well-structured presentation that is easy to follow. It effectively employs examples and explanations to enhance comprehension of both the algorithm and the underlying theory.


**Weaknesses:**

**Significance**: While I acknowledge the valuable contributions made by this paper in terms of algorithmic and theoretical aspects, it is important to note that the current theory relies on one-sided KL or one-sided convex/concave conditions. Considering the paper's objective of addressing limit cycles, it would be beneficial to establish a clearer connection between the existing theory and the role of DSGDA in avoiding such cycles. Elaborating on this relationship would significantly enhance the paper's overall strength and clarity.


**Questions:**

**Main Questions**

1. Relation between OS and GS: I would suggest including further discussions in the paper regarding the relationship between OS and GS notations. It appears that there exists a translation method between these two notations, as mentioned in [1]. Elaborating on this relationship can greatly enhance the paper's quality and comprehensibility.

2. Odd constants in Theorem 1:The extrapolation step in the algorithm introduces exceedingly large constants, which might seem counter-intuitive. Although the calculations on Page 23 provide some insight into their derivation, further explanation is required to clarify the reasoning behind these values and make them more understandable to readers.

3. Connection between theory and practice: Considering the paper's objective of addressing limit cycles, it would be beneficial to establish a clearer connection between the existing theory and the role of DSGDA in avoiding such cycles. Elaborating on this relationship would significantly enhance the paper's overall strength and clarity.

4. Unconstrained setting: While the paper focuses on the two-sided constrained setting, it is worth noting that some existing papers tackle the scenario where only the domain of the max variable is bounded, as demonstrated in [1]. To enhance the paper, it could be valuable to discuss the disparities and challenges encountered in the unconstrained setting, providing a comparative analysis between the two scenarios.

5. Stochastic setting: It would be interesting to see if similar theoretical results can be obtained in the stochastic setting as provided in [2]

**Minor issues**

6. I would recommend not directly using “nonconvex-nonconcave” in the title as the paper does not consider the general nonconvex-nonconcave setting.

7. I recommend not using “SGDA” to represent the smoothed GDA as SGDA is well-known for stochastic gradient descent ascent algorithms. Using “S-GDA” and “DS-GDA” instead would avoid confusion.

8. Line 77: The size of the right parenthesis

**Reference**

[1]  Lin, Tianyi, Chi Jin, and Michael Jordan. "On gradient descent ascent for nonconvex-concave minimax problems." International Conference on Machine Learning. PMLR, 2020.

[2] Yang, Junchi, et al. "Faster single-loop algorithms for minimax optimization without strong concavity." International Conference on Artificial Intelligence and Statistics. PMLR, 2022.


**Limitations:**

The paper is theoretical and does not have any potential negative societal impact.

---

> ### Author Rebuttal · Authors · 2023-08-09
>
> We are happy that the reviewer enjoys our paper and would like to thank you for your insightful comments. Below we provide a point-by-point response to your comments and questions.
>
> **Q1: Connection between theory and practice: It would be beneficial to establish a clearer connection between the existing theory and the role of DSGDA in avoiding such cycles.**
>
> Thanks for your advice. We will add additional discussions in our updated version.
> * All existing work for NC-NC problems relies on restricting the function classes to achieve convergence. There are mainly three types of regularity conditions in the literature, PL condition, VI-related condition, and $\alpha$-interaction dominance (see Appendix B). With PL condition imposed on the dual function, the inner max function $\phi(\cdot)=\max_{y\in \mathcal{Y}} f(\cdot,y)$ is smooth and we can regard minimax problem as a pure smooth minimization problem. VI conditions regard the primal and dual variables as integral and update them together, which does not follow the nature of sequential games. The one-sided $\alpha$-dominance condition imposes the dominant player before the game. Instead, we address the NC-NC minimax problem by achieving primal-dual balance through algorithmic developments. We believe the double smoothing technique is an efficient manner to get rid of the limit cycle.
>  * Extensive experiment results also validate the power of DSGDA (see Section 2). To the best of our knowledge, DSGDA is the first algorithm, which can converge on all four difficult examples in literature. We admit the theoretical result of DSGDA is restricted to one-sided KL (convex/concave) cases.  Although there is a gap between practice and theory, we hope it opens a new path for studying and developing new theoretical frameworks.
>
> **Q2: Relation between OS and GS**
>
> Thanks for the good question. We also have such GS and OS translation results: **If $(x,y)\in \mathcal{X}\times \mathcal{Y}$ is an $\epsilon$-GS, then it is an $\mathcal{O}(\epsilon^{\min\\{1,1/2\theta\\}})$-OS.** One should be noted that our algorithm-dependent result (Theorem 2 for the proposed DSGDA) will achieve the GS and OS at the same rate, which is stronger than directly applying the translation result when $\theta\in(\frac{1}{2},1)$. We will add the proof in our updated version and here is a proof sketch.
>
> **Proof Sketch** Building on the Proposition 2 in Appendix, we have the following bound of the measurement of OS:
>          $$\\|x^*(x)-x\\|\leq \\|x^*(x)-x(x,y)\\|+\\|x(x,y)-x\\|\leq \omega_2\\|y-y(x,y)\\|^{\frac{1}{2\theta}}+\\|x(y,x,v)-x\\|+\sigma_1\\|y-y(x,y)\\|.$$
>
> By the nonexpansiveness of the projection operator and error bounds in Lemma 2 and 3, we can further bound $\\|y-y(z,v)\\|$ as follows:
> $$\\|y-y(z,v)\\|\leq \sigma_8 \\|y-y_{+}(z,v)\\|\leq  L_y\alpha\sigma_8\\|x-x(y,z,v)\\|+\left(2+\alpha L_y+\alpha r_2\right)\sigma_8\\|y-y(x,z,v)\\|.$$
>
> Next, we will explore the relationship between $\\|x-x(y,x,v)\\|$ and $dist(z, \nabla_{x}f(x,y)+\partial1_{\mathcal{X}}(x))$. Let $x_{+}(y,x,v):=proj_{\mathcal{X}}(x-c\nabla_{x}F(x,y,x,v))$, then from the primal error bound (see [3]), we know that
> $$\\|x-x(y,x,v)\\| \leq \frac{cL_x+cr_1+1}{cr_1-cL_x}\\|x-x_{+}(y,x,v)\\|.$$
>
> Moreover, since $\nabla_{x}F(x,y,x,v)=\nabla_{x}f(x,y)$, then following from Lemma 4.1 of [4], we get
> 			$$\\|x-x(y,x,v)\\| \leq  \frac{cL_x+cr_1+1}{cr_1-cL_x}\\|x-proj_{\mathcal{X}}(x-c\nabla_{x} f(x,y))\\|\leq  \frac{cL_x+cr_1+1}{r_1-L_x} dist(z, \nabla_{x}f(x,y)+\partial1_{\mathcal{X}}(x)).$$
> 		The similar analysis can be applied to derive the bounds for $\\|y-y(x,z,y)\\|$. Thus, if $(x,y)$ is an $\epsilon$-GS point, then it is also an $\mathcal{O}(\epsilon^{\min\\{1,1/2\theta\\}})$-OS.
>
> **Q3: Odd constants in Theorem 1**
>
> We kindly request the reviewer to confirm whether we misunderstand your question. From our side, the extrapolation step $\beta$ should decrease to zero w.r.t. iteration $T$ when $\theta\in (\frac{1}{2},1)$ (see Theorem 2). This implies that the sequence ${z_t}$ will undergo a decreasing rate of change, gradually approaching convergence. Conceptually, you can regard $z$ as an approximated proximal mapping of the max function $\max_{y\in\mathcal{Y}} f(\cdot,y)$. For further insights into parameter selection, please refer to the global response to Q2.
>
> **Q4: Unconstrained setting**
>
> For the NC-C problem, the primal boundedness requirement can be removed and only the lower boundedness of the max function is enough, which allows the unbounded case and match those results in [1]. Otherwise, for the NC-NC case, the boundedness requirement is needed to ensure the lower boundedness of the Lyapunov function.
>
> **Q5: Stochastic setting**
>
> Yes, it can be done. There are no intrinsic difficulties. However, since the paper is already 32 pages, it may disperse the current focus of the paper if the proof of stochastic setting is included.
>
> **Q6: Minor issues**
>
> Thanks for your advice. We will modify our title to "Doubly Smoothed GDA for a class of Constrained Nonconvex-Nonconcave Minimax Optimization". For the other suggestions on abbreviation and right parenthesis, we will revise it afterward.
>
> [3]. Pang J S. A posteriori error bounds for the linearly-constrained variational inequality problem.
>
> [4]. Li G, Pong T K. Calculus of the exponent of Kurdyka–Łojasiewicz inequality and its applications to linear convergence of first-order methods.

---

> > ### Comment · Reviewer_gc6W · 2023-08-13
> >
> > Thanks for the response! I am looking forward to seeing the upcoming revision, particularly regarding the relation between OS and GS. I would be much happier if authors can provide more theoretical and fundamental analyses on *"We address the NC-NC minimax problem by achieving primal-dual balance through algorithmic developments. We believe the double smoothing technique is an efficient manner to get rid of the limit cycle."*

---

### Official Review · Reviewer_pZz2 · 2023-07-07

**Soundness:** 2 fair
**Presentation:** 3 good
**Contribution:** 2 fair
**Rating:** 4
**Confidence:** 4

**Summary:**

Constructing a method that finds an optimal point of nonconvex-nonconcave problems is of interest. Built upon the "one-sided" smoothed GDA (SGDA) that exploits the Moreau-Yosida smoothing technique, this paper studies the "doubly" smoothed GDA (DSGDA), applying the smoothing to both the primal and dual variables. As an illustration, this paper considers four challenging nonconvex-nonconcave minimax problems that do not satisfy any of the regularity conditions considered in the literatures and that none of existing methods work. It turns out that the DSGDA is the first known method that works for all four difficult problems. To complement this empirical success, the authors theoretically show that the DSGDA converges to a stationary point under an one-sided KL condition. (The aforementioned four problems do not satisfy the one-sided KL condition.) The authors also present that the DSGDA works for a wider range of hyperparameters than the SGDA for nonconvex-nonconcave problems.

**Strengths:**

- The proposed DSGDA methods empirically converges to stationary points of four notoriously difficult minimax problems, while other existing methods do not, which is impressive.
- The authors shows that the DSGDA converges to a stationary point of nonconvex-nonconcave problems with one-sided KL property. The method achieves the best known rate for a more restrictive setting that the dual function satisfies either the concavity or the PL condition.



**Weaknesses:**

* The DSGDA does not outperform SGDA under the one-sided KL setting, where the proposed DSGDA is theoretically shown to work well. Would it be possible to show that the SGDA also converges under the one-sided KL setting? Or instead, would it be possible to show that it does not converge? Are there other methods that work under the one-sided KL setting?
* As mentioned by the authors, there remains a gap between the theoretical claim and practical claim.
* The robustness claim is only based on one nonconvex-nonconcave experiment. In addition, the region of convergence of DSGDA for the nonconvex-nonconcave case is relatively small, so it does not seem to be sufficient to claim that DSGDA is robust (even though it is relatively robust than the SGDA).
* The choice of hyperparameters that guarantees convergence in Theorem 1 is complicated, so it is not easy to find an optimal choice in terms of the rate, making it difficult for the users to choose.

**Questions:**

* Line 33: Why does no player inherently dominate the other in nonconvex-nonconcave problems? Later in line 50, the authors claim that the primal player becomes dominant for the nonconvex-concave problem. Then, consider a nonconvex-nonconcave problem that is locally nonconvex-concave at some stationary points. Then will the primal player suddenly become dominant locally?

* Line 34: How do you achieve a good balance between primal and dual updates in this paper? Of course, applying smoothing to both variables improve the balance, as claimed by the authors. However, the choice of regularization parameters $r_1$ and $r_2$ and step sizes $\alpha$, $c$, $\beta$, $\mu$ are not the same for both variables. At this point, I am not following what it really means by the good balance.

* line 83: In what sense, is this work the best known convergence analysis?

* Are there practical examples that satisfy the one-sided KL property but not the one-sided PL property?

**Limitations:**

This work has potential but the gap between the practical claim and the theoretical claim seems not negligible, and I think that the paper could have been better if such gap was further reduced. For example, an experiment with one-sided KL where DSGDA outperforms SGDA could have better supported the paper's claim.

---

> ### Author Rebuttal · Authors · 2023-08-09
>
> Thanks for your constructive comments. We hope the clarification that we emphasize may help for a better understanding of our contributions.
>
> **Q1: Does SGDA converge under the one-sided KL setting?  As there other methods that work under the one-sided KL setting?**
>
> Recent work [1] has demonstrated the convergence of the SGDA under the one-sided KL condition, and it is currently the sole method to have a global convergence guarantee under this setting. Despite this, DSGDA still outperforms SGDA due to its universality. Please refer to the global response to Q1.
>
> **Q2: Gap between theory and practice.**
>
> Please refer to the global response to Q3.
>
> **Q3: The region of convergence of DSGDA for the nonconvex-nonconcave case is small, so it is not sufficient to claim that DSGDA is robust.**
>
> The NC-NC example ($a=11$) we evaluate in the paper (Figure 3(c)) is already an extremely challenging case as SGDA fails to converge at $a=10$. The challenge stems from the difficulty in identifying the dominant side. Thus, the problem becomes easier and the range of parameters becomes wider when $a$ becomes larger (see Figure (e)-(f) in the **PDF**). To further corroborate the robustness of DSGDA, we provide the parameter ranges for several less challenging problems when $a=11.6$ and $a=13$. It's evident that both robustness (see Figures (c)-(d) in the **PDF**) and fast convergence (see Figures (a)-(b) in the **PDF**) of DSGDA persist across these NC-NC problems.
>
> **Q4: It is not easy to find an optimal choice in terms of the rate.**
>
> The choice of parameters will not affect the convergence rate. Actually, all parameters only relate to the Lipschitz constants of the gradients $L_x, L_y$. Please refer to the global response to Q2.
>
> **Q5: Will the primal player suddenly become dominant locally for local NC-C problem?**
>
> Thanks for pointing out it. We would like to clarify that dominance is a global concept. One player is said to dominate the other if its decisions determine the convergence no matter how good action the other player has done.  For NC-NC problems, there is no player inherently dominates the other and the local structure plays no effect on global convergence.
>
> **Q6: How do you achieve a good balance between primal and dual updates in this paper? The choice of parameters is not same**
>
> The unbalance between primal and dual updates in minimax raises from the different optimal directions and these related changing quantities of two players. That is, one aims to minimize the function value, while the other aims to maximize it, making it hard to guarantee the sufficient decrease property of a function. Thus, achieving a good balance means that we can construct a novel Lyapunov function that possesses the "sufficient decrease" property (see Theorem 1). Here, the parameters are not the same in general. They are precisely controlled to ensure the 'sufficient decrease' property, achieving a good balance between primal-dual updates.
>
> **Q7: In what sense, is this work the best-known convergence analysis?**
>
> Thanks for your questions. We will clarify this point in our updated version.
>
> * **NC-C:** DSGDA attains both the $\epsilon$-GS (game-stationary point) and $\epsilon$-OS (optimization-stationary point) with a complexity of $\mathcal{O}(\epsilon^{-4})$, matching the sharpest rate among single-loop algorithms for NC-C minimax problems.
>
> * **NC-PL/NC-SC:** DSGDA attains both the $\epsilon$-GS and $\epsilon$-OS with a complexity of $\mathcal{O}(\epsilon^{-2})$, which is already optimal.
>
> **Q8: Are there practical examples that satisfy the one-sided KL property but not the one-sided PL property?**
>
> PL property is only defined for unconstrained problems, while KL property is more general and can be applied to constrained (nonsmooth) problems. Additionally, even in unconstrained cases, PL condition is a special case of KL with exponent $\theta=\frac{1}{2}$. Hence, KL functions are considerably broader than PL functions. Many constrained minimax problem examples will be that case. More specifically,  we may consider the widely considered max-structured problem $\min_{x\in\mathcal{X}} \max_{y\in\Delta} y^\top G(x)$,
> where $\Delta =\\{y \in \mathbb{R}^d:\sum_{i=1}^d y_i = 1, \ y \ge 0\\}$ is the standard simplex. Such a problem arises frequently in machine learning applications, including distributionally robust optimization, adversarial training and fairness training. It can be shown that this problem possesses the KL property with exponent $\theta=0$ for the dual problem under the mild condition that there exists $\delta>0$ such that $\max_{i\in [d]} G_i(x^*)\ge G_j(x^*)+\delta$ for $j\in[d]$ satisfying $y_j^*=0$, see [1] for details. Moreover, the constrained NC-SC and NC-PL problems all satisfy the one-sided KL property. For example, the nonconvex-regularized variant of DRO problem and multi-class classification problems mentioned in [2] possesses KL property with $\theta=\frac{1}{2}$ for dual problem.
>
> **Q9: An experiment with one-sided KL where DSGDA outperforms SGDA could have better supported the paper's claim.**
>
> Thank you for your suggestions. To further demonstrate the superiority of DSGDA over SGDA for the NC-KL case, we have included an additional experiment, see Figure (g) in the **PDF** (the example we test in the global response to Q1). As we have mentioned, the universal applicability of DSGDA has been clarified in the global response to Q1. We further support it with the new experiment illustrated in Figure (g). Notably, DSGDA, even without prior knowledge, demonstrates faster convergence upon entering the local region of the stationary point. This efficiency contrasts with SGDA, even when equipped with the correct smoothing side.
>
> [1] Li J, Zhu L, So A M C. Nonsmooth nonconvex-nonconcave minimax optimization: primal-dual balancing and iteration complexity analysis.
>
> [2] Zhang X, Aybat N S, Gurbuzbalaban M. Sapd+: An accelerated stochastic method for nonconvex-concave minimax problems.

---

> > ### Comment · Reviewer_pZz2 · 2023-08-14
> >
> > Thank you for the detailed response. I am still reading other reviews and rebuttals, so I will finalize my decision shortly.
> >
> > In the meantime (as the discussion period is nearing the end), I have a question about the universality of the proposed method. The authors claim that the DSGDA is universal due to its symmetry of extrapolation, unlike the SGDA that extrapolates only one side. I agree with that. However, still doesn't the DSGDA choose the hyperparameters asymmetrically? In the paper, I was not able to find any formal mathematical statement that DSGDA with the chosen asymmetric hyperparameters can converge across NC-NC, NC-C, and C-NC problems. If you need to adjust the hyperparameters manually, then I think the universality claim should be weakened. Let me know what I am misunderstanding here.

---

> > > ### Author Response · Authors · 2023-08-16
> > >
> > > Thank you for your response and follow-up questions. We would like to emphasize and clarify that the symmetric ability to extrapolate offered by DSGDA does not imply the symmetric selection of primal-dual parameters; DSGDA is indeed theoretically ensured to converge across various scenarios (NC-C/C-NC/NC-KL/KL-NC) without the need for manual parameter adjustments.
> > >
> > > In standard optimization problems, an algorithm is typically considered **universally effective** in ensuring a performance guarantee when there exists a well-defined parameter prescription, based on local problem characteristics and global Lipschitz constant, that attains the desired performance guarantee across a broad class of problems.  In our (more complex setting involving min-max optimization) this is what DSGDA achieves. Note that every parameter's choice depends only on Lipschitz constants and (importantly) not on the knowledge of the one-sided KL exponent/convexity/concavity. Maybe, the referee can provide some insight into why symmetry of parametric selection is something that is reasonable to expect or even desirable in this setting, especially when the nature of the problem and algorithm here is very non-symmetrical in that the roles of the variables are non-exchangeable.
> > >
> > > Then, we offer a comprehensive elucidation of why DSGDA  does not enforce manual tuning of hyperparameters. All hyperparameters are just based on the Lipschitz constants. To start with, we want to emphasize only two factors that contribute to the non-symmetric parameter selections within DSGDA. We will discuss two factors separately.
> > >
> > >
> > > 1. [Theorem 1] The asymmetric parameters selection in DSGDA only stems from which variables we choose to update first (i.e., whether to update $x$ followed by $y$, or vice versa), **which is not related to the inherent asymmetry of NC-C/C-NC/NC-KL/KL-NC at all.**
> > > As you can observe from Theorem 1 (basic descent estimate), all hyperparameters are solely based on the Lipschitz constant $L_x, L_y$. We did not rely on any convexity/concavity or one-side KL exponent. We can conclude that the symmetric ability offered by DSGDA does not imply the symmetric selection of primal-dual parameters.
> > >
> > > 2. [Theorem 2] Moreover, upon Theorem 1, we use Proposition 1 (error bound condition) to quantitatively control the negative term in basic estimate descent (line 216) in Theorem 1 and establish the main theorem (see Theorem 2). The remaining asymmetry only arises from the selection of parameters $\beta$ and $\mu$, which is attributed to which side we want to apply Proposition 1 (error bound condition). Thus, we can always choose the smallest extrapolation steps between these two sides with KL $\theta =1$, e.g., $\mu$ and $\beta$ as the same order $\mathcal{O}(T^{-\frac{1}{2}})$ to guarantee the convergence of DSGDA. However, it is worth noting that this universality will result in a suboptimal rate as $\mathcal{O}(\epsilon^{-4})$.
> > > In cases where additional information is available regarding the KL exponent $\theta$ (**what we discussed in Theorem 2**), a better choice of $\beta$ or $\mu$ can be selected to achieve a sharper convergence rate of $\mathcal{O}(\epsilon^{-2\max\{2\theta,1\}})$.
> > >
> > > In summary, we survey here why our parameter selection obeys only the global Lipschitz constant and the impact of additional information on sharper convergence rate supplied by the KL exponent.
> > >
> > > We will add this detailed discussion in our updated version. We believe our response will address your concerns about the universality issue. We'd be happy to take more follow-up questions.

---

> > > > ### Comment · Reviewer_pZz2 · 2023-08-16
> > > >
> > > > Thanks for the detailed response. I appreciate it.
> > > >
> > > > I understand that the parameters do not depend on the inherent asymmetry of NC-C/C-NC/NC-KL/KL-NC. What I was interested here is that, for the chosen parameters in Theorem 1 that is used to prove Theorem 2 for NC-C/NC-KL problems, can you also show that it converges for C-NC/KL-NC? I am curious about this because in Remark 2 the authors suggest similar but different Lyapunov function for C-NC/KL-NC problems.
> > > >
> > > > In short what I am interested in is the details about the following statement in Remark 2: "For KŁ-nonconcave and convex-nonconcave minimax problems, similar results as Theorem2 could be derived".

---

> > > > > ### Author Response · Authors · 2023-08-16
> > > > >
> > > > >
> > > > > Thank you very much for your prompt response and the additional clarification. Your dedication to reviewing our paper is greatly appreciated.
> > > > > Now, we have a much clearer understanding of your concern. The Lyapunov function utilized in Theorem 2 can also effectively be utilized to establish the convergence of DSGDA on C-NC/KL-NC but will enjoy the suboptimal rate of $\mathcal{O}(\epsilon^{-4})$ as we discussed in the previous response \#point 2. The adoption of the alternative Lyapunov function in Remark 2 aims at obtaining a sharper convergence rate whenever additional information, such as the one-side KL exponent, is available.

---

> > > > > > ### Comment · Reviewer_pZz2 · 2023-08-16
> > > > > >
> > > > > > I appreciate the authors' prompt response. I think the result for the C-NC/KL-NC was only vaguely stated, so the universality of the DSGDA was not really obvious in the submitted paper.
> > > > > >
> > > > > > Are you now saying that the DSGDA with the parameters in Theorem 1 (that has a specific order) also works for the C-NC/KL-NC?  If yes, I think this is a very strong result and that is why I am really curious about here. Looking at how you deal with the positive and negative terms of the Lyapunov function in the proof of Theorem 2 (exploiting the NC-C/NC-KL structure), I cannot easily see how everything will work the same for the opposite C-NC/KL-NC case without seeing the detailed proof. So, I would really appreciate to see a formal statement and the proof for C-NC/KL-NC, which I think should be included in the paper to strongly support the authors' universality claim. Then, I will be happy to raise the score for acceptance.

---

> > > > > > > ### Author Response · Authors · 2023-08-18
> > > > > > >
> > > > > > > Thanks so much for your prompt reply and instructive feedback. We fully agree with your suggestions. We sincerely apologize for any confusion that may have arisen.
> > > > > > >
> > > > > > > The parameters presented in Theorem 1 are specifically tailored to address the NC-C and NC-KL problems. We cannot use these parameters in a specific order to get the convergence proof of C-NC and KL-NC directly. However, the universality claim still holds. We will provide detailed formal statements and explanations here.
> > > > > > >
> > > > > > > As we mentioned in **global response to Q2**, there are numerous alternative selections. In Theorem 1, we just pick a single set of them which is convenient for us to get a sharper rate in Theorem 2. Furthermore, the non-symmetric nature of the parameter ranges (as we have expounded on the rationale behind this asymmetry) does not imply that selecting a symmetric parameter pair (such as $r_1=r_2, c=\alpha, \beta=\mu$) to derive an analogous version of the basic sufficient descent (Theorem 1) is impossible. As we mentioned in the previous response, these universality/symmetric requirements will shrink the parametric range and we have to further pick up both $\beta$ and $\mu$ are sufficiently small and decrease to zero at the order $\mathcal{O}(T^{-1/2})$, thus leading to the suboptimal rate. The underlying idea here is to shift the cost towards extrapolation parameters in order to achieve a sense of equilibrium.
> > > > > > >
> > > > > > > Below, we present a formal corollary statement:
> > > > > > >
> > > > > > > **Corollary** Without loss of generality, we assume $\lambda=1$ (i.e., $L_x=L_y=L$). Suppose that Assumption 1 holds and $r_1=r_2=70L$, $c=\alpha=1/(1400L)$, $0<\beta=\mu\leq \min\\{\mathcal{O}(T^{-1/2}),1/10000\\}$. Then, for any $t\geq 0$, we have  following sufficient decrease property (A) holds
> > > > > > > $$
> > > > > > > \Phi^t-\Phi^{t+1}\geq \frac{r}{1000}\\|x^{t+1}-x^t\\|^2+\frac{r}{1000}\\|y^t-y^t_{+}(z^t,v^t)\\|^2+\frac{r}{1000\beta}\\|z^t-z^{t+1}\\|^2+\frac{r}{1000\mu}\\|v_{+}^t(z^{t+1})-v^t\\|^2\ - 4r\beta\\|x(z^{t+1},v(z^{t+1}))-x(z^{t+1},v_+^t(z^{t+1}))\\|^2.
> > > > > > >  $$
> > > > > > > **Proof** The proof details remain identical to those of Theorem 1 up to line 652 (in Appendix). Different from Lines 653-670 (manually control all hyperparameters), we simply let $r_1=r_2=t_2 \cdot L$, $c=\alpha = \tfrac{1}{t_1\cdot t_2 \cdot L}$ and $\beta = \mu = \tfrac{1}{10000}$.
> > > > > > > Consequently, we insert these relationships into Line 652, and a straightforward verification reveals that choosing $t_1 = 20$ and $t_2 = 70$ renders the basic descent estimate (A) satisfiable.
> > > > > > >
> > > > > > > It's worth noting that beyond the specific choice of $(20, 70)$, a wide spectrum of feasible pairs exists within which the basic descent estimate (A) remains achievable. To visualize this, we can utilize MATLAB to visualize the feasible region for $0 < t_1, t_2 \leq 100$ that ensures convergence.
> > > > > > >
> > > > > > > **MATLAB Code**
> > > > > > > ```
> > > > > > > % Define the parameters
> > > > > > > mu = 1 / 10000;
> > > > > > > beta = 1 / 10000;
> > > > > > > kappa = 2 * beta;
> > > > > > > % Create a grid of t1 and t2 values
> > > > > > > t1_values = linspace(0, 100, 500);
> > > > > > > t2_values = linspace(0, 100, 500);
> > > > > > > [T1, T2] = meshgrid(t1_values, t2_values);
> > > > > > > % Compute the expressions
> > > > > > > c = 1 ./ (T1 .* T2);
> > > > > > > sigma_1 = T2 ./ (T2 - 1);
> > > > > > > sigma_2 = T2 ./ (T2 - 1);
> > > > > > > sigma_5 = T2 ./ (T2 - 1);
> > > > > > > sigma_3 = sigma_2 .* (T2 - sigma_1) ./ (T2 - sigma_1 - 1);
> > > > > > > sigma_6 = T2 .* (2 + T1) ./ (T2 - 1);
> > > > > > > sigma_8 = (sigma_1 + 1 + T2 + T1 .* T2) ./ (T2 - 1);
> > > > > > > L_d = (sigma_1 + 1) + T2 ;
> > > > > > > s1 = 1 ./ c - (T2 + 1)/ 2 - 1; % Line 643
> > > > > > > s2 = 1 ./ c -  sigma_6.^2 - L_d + (T2 - 1) / 2; % Line 643
> > > > > > > s3 = T2 .* (1/beta - 1/2 - 2 * sigma_2 - 1/kappa); % Line 644
> > > > > > > coeff = mu * (2 - mu) .* T2  + 6 * T2 * kappa .* sigma_1.^2; % Line 655
> > > > > > > coeff_x = s1 - s2 .* c.^2 .* sigma_6.^2 - coeff .* 2 .* c.^2 .* sigma_6.^2 ;
> > > > > > > coeff_y = s2/2 - 2 * coeff .* (1 + sigma_8).^2;
> > > > > > > coeff_z = s3 - coeff .* sigma_3.^2;
> > > > > > > coeff_v = T2 .* (1/(2 * mu) - 1/4 - 6 * kappa * sigma_1.^2 .* sigma_5.^2);
> > > > > > > % Compute pos_x, pos_y, pos_z, and pos_v
> > > > > > > pos_x = coeff_x - T2 / 1000;
> > > > > > > pos_y = coeff_y - T2 / 1000;
> > > > > > > pos_z = coeff_z - T2 / (1000 * beta);
> > > > > > > pos_v = coeff_v - T2 / (1000 * mu);
> > > > > > > % Create a binary mask for feasible region
> > > > > > > feasible_mask = pos_x > 0 & pos_y > 0 & pos_z > 0 & pos_v > 0;
> > > > > > > % Plot the feasible region
> > > > > > > figure;
> > > > > > > contourf(T1, T2, double(feasible_mask), [0, 1]);
> > > > > > > xlabel('t1');
> > > > > > > ylabel('t2');
> > > > > > > title('Feasible Region');
> > > > > > > grid on;
> > > > > > > xt = [20 60];
> > > > > > > yt = [70 50];
> > > > > > > text(xt,yt,{'Feasible', 'Infeasible'},'FontSize',15);
> > > > > > > ```
> > > > > > >
> > > > > > > We suggest running this simple Matlab code, and upon doing so, you will observe a broad spectrum of feasible pairs that render DSGDA universally applicable to NC-C/C-NC/NC-KL/KL-NC problems.
> > > > > > >
> > > > > > > As evident, the parameters in the Corollary are smaller compared to the asymmetric version. For a specific problem with prior knowledge of which sides are dominant, the application of asymmetric parameters as detailed in Theorem 1 remains advantageous in achieving a sharper convergence rate.
> > > > > > >
> > > > > > > We sincerely appreciate your time and thorough review of our work. We will add these details to our final version. We would be delighted to address any concerns you may have.

---

> > > > > > > > ### Comment · Reviewer_pZz2 · 2023-08-18
> > > > > > > >
> > > > > > > > I appreciate the authors' detailed response.
> > > > > > > >
> > > > > > > > In summary, the authors' provided a version of Theorem 1 with smaller and symmetric step sizes by using the Lyapunov function in line 198. The authors then imply that this will achieve a suboptimal rate for both NC-C/NC-KL and C-NC/KL-NC. Correct me if am wrong. I am impressed that the method still has the universality (but without the optimal rate). However, I was further expecting a formal statement analogous to Theorem 2 (presenting iteration complexity). For example, which suboptimal rate does DSGDA (with such small and symmetric step sizes) has under the NC-C/NC-KL? A formal statement for the C-NC/NC-KL (for both asymmetric and symmetric step sizes) is still missing. I requested to see a formal statement and proof for the C-NC/KL-NC setting but this is not provided.
> > > > > > > >
> > > > > > > > I appreciate the authors' effort for responding my requests. However, although I commented that I might increase my score, I must say that I am not much convinced to raise a score. I now think that this paper needs a major revision, and is not ready for publication in this round.

---

> > > > > > > > > ### Author Response · Authors · 2023-08-18
> > > > > > > > > **(1/2) The formal statements for all symmetric counterparts have been provided.**
> > > > > > > > >
> > > > > > > > > Thanks for your quick response and detailed comments. The universal version of DSGDA  will achieve a suboptimal rate of $\mathcal{O}(\epsilon^{-4})$ across NC-C/NC-KL/C-NC/KL-NC minimax problems (**we mentioned in our previous response multiple times**).
> > > > > > > > >
> > > > > > > > > We wish to highlight that all the supplementary formal statements you have inquired about are essentially symmetrical counterparts of what we have already presented in the paper or the provided corollary in the last response. That is, **the adjustment only entails interchanging the positions of $(x,z,r_1,c,\beta,L_x)$ and $(y, v, r_2,\alpha,\mu,L_y)$, and the positions of $(d,q)$ and $(h,g)$. This modification does not entail any alterations in terms of scientific content.**
> > > > > > > > > Considering the constraints of the 9-page format, we have made the choice to allocate our space to convey what we consider to be the most essential content.
> > > > > > > > > It is worth noting that, upon acceptance, we will have an additional page to add back those statements without additional effort.
> > > > > > > > > It would be disappointing if the rejection of our paper were solely attributable to this matter of communication.
> > > > > > > > >
> > > > > > > > >
> > > > > > > > > We will now proceed to provide all symmetrical counterparts you requested. This encompasses the primal error bound (**Proposition 2**), the selection of asymmetric step sizes for C-NC/KL-NC (**Theorem 4**), and the iteration complexity of DSGDA  for C-NC/KL-NC with asymmetric step sizes (**Theorem 5**). Lastly, we provide the formal statement for the iteration complexity of universal DSGDA across NC-C/C-NC/NC-KL/KL-NC (**Theorem 6**). Please note that all these theoretical statements can be readily derived from the content presented in the current manuscript.
> > > > > > > > >
> > > > > > > > > Please see the next **official comment** for further details.
> > > > > > > > >
> > > > > > > > > As of now, we haven't introduced any new proof in our rebuttal.  We're struggling to discern the basis for considering a major revision of our paper from your perspective. From our end, there is no further need for an additional round of review for any new statement/proof details.  We sincerely hope you might reconsider your stance. Thanks so much for your effort in reviewing our paper.

---

> > > > > > > > > > ### Author Response · Authors · 2023-08-18
> > > > > > > > > > **(2/2) The formal statements for all symmetric counterparts have been provided.**
> > > > > > > > > >
> > > > > > > > > > **Proposition 2** (**[symmetric to Proposition 1]** Proximal error bound) With convexity or KL property for primal function, for any $z \in \mathbb{R}^n, v \in \mathbb{R}^d$ one has
> > > > > > > > > > - KL exponent $\theta \in (0,1)$:
> > > > > > > > > > $$\\|y(z_+^t(v^{t+1}),v^{t+1})-y(z(v^{t+1}),v^{t+1})\\|^2\leq \omega_2\\|z_+^t(v^{t+1})-z^t\\|^{\frac{1}{\theta}};$$
> > > > > > > > > > - Convex:
> > > > > > > > > > $$    \\|y(z_+^t(v^{t+1}),v^{t+1})-y(z(v^{t+1}),v^{t+1})\\|^2 \leq \omega_3\\|z_+^t(v^{t+1})-z^t\\|,$$
> > > > > > > > > > where
> > > > > > > > > >   $\omega_2 :=\frac{2}{(r_2-L_y)\tau}\left(\frac{r_1(1-\beta)}{\beta}+\frac{r_1^2}{r_1-L_x}\right)^{\frac{1}{\theta}}$ and $\omega_3 := \frac{4r_1 diam(\mathcal{X})}{r_2-L_y}\left(\frac{1-\beta}{\beta}+\frac{r_1}{r_1-L_x}\right)$.
> > > > > > > > > >
> > > > > > > > > > **Proof: The proof remains unchanged from that of Proposition 1, with the sole exchange between  $(x,z,r_1,c,\beta,L_x)$ and  $(y,v,r_2,\alpha,\mu,L_y)$.**
> > > > > > > > > >
> > > > > > > > > >
> > > > > > > > > >
> > > > > > > > > >
> > > > > > > > > >
> > > > > > > > > >
> > > > > > > > > > **Theorem 4** (**[symmetric to Theorem 1]** C-NC/KL-NC with asymmetric step sizes) Suppose that Assumption 1 holds and $3(1+\lambda)L\leq r_2\leq 4(1+\lambda)L$, $(\frac{\lambda^2}{2+3\lambda}+4\lambda+4)L\leq r_1\leq 2(3\lambda+2)L$ with the parameters $\alpha \in\left[\frac{1}{2(2+3\lambda)L},\frac{1}{\max\\{\frac{3}{4}(L+r_1),6\lambda L\\}}\right]$, $c\in\left[\frac{1}{\max\\{\frac{3\lambda L(2cr_1+1)^2}{2(cr_1-cL)^2},  8(r_2+2\lambda L+\frac{\lambda^2 L^2}{r_1-L})  \\} + L },   \frac{1}{\max\\{\frac{3\lambda L(2cr_1+1)^2}{2(cr_1-cL)^2},  8(r_2+2\lambda L+\frac{\lambda^2 L^2}{r_1-L})  \\} }  \right]$
> > > > > > > > > > ; $\mu\in(0, \frac{1}{20000(5\lambda+4)}]$,    $\beta\in(0, \frac{1}{2500(2\lambda+1)}]$. Then, for any $t\geq 0$,
> > > > > > > > > >
> > > > > > > > > > $$\Psi^t-\Psi^{t+1}\geq \frac{r_2}{20}\\|y^{t+1}-y^t\\|^2+\frac{r_1}{10}\\|x^t-x^t_{+}(z^t,v^t)\\|^2+\frac{r_2}{2\mu}\\|v^t-v^{t+1}\\|^2+\frac{12r_1}{25\beta}\\|z_{+}^t(v^{t+1})-z^t\\|^2\ -2r_2(4+5\lambda)\mu\\|y(z(v^{t+1}),v^{t+1})-y(z_+^t(v^{t+1}),v^{t+1})\\|^2.
> > > > > > > > > > $$
> > > > > > > > > > where
> > > > > > > > > > $x_{+}(z,v):=proj_{\mathcal{X}}(x-c\nabla_{x}F(x,y(x,z,v),z,v))$ and $z_{+}(v):=z+\beta(x(z,v)-z)$ with $z(v):=argmin_{z\in \mathbb{R}^n} p(z,v)$.
> > > > > > > > > >
> > > > > > > > > > **Proof: The proof remains unchanged from that of Theorem 1, with the sole exchange between  $(x,z,r_1,c,\beta,L_x)$ and  $(y,v,r_2,\alpha,\mu,L_y)$.**
> > > > > > > > > >
> > > > > > > > > >
> > > > > > > > > >
> > > > > > > > > > **Theorem 5** (**[symmetric to Theorem 2]** Iteration complexity of DSGDA C-NC/KL-NC with asymmetric step sizes)
> > > > > > > > > > Under the setting of Theorem 4 and Proposition 2, for any $T>0$, there exists a $t\in\{1,2,\cdots,T\}$ such that
> > > > > > > > > >
> > > > > > > > > > - KL exponent $\theta \in (\frac{1}{2},1)$: $(x^{t+1},y^{t+1})$ is an $\mathcal{O}(T^{-\frac{1}{4\theta}})$-GS and $v^{t+1}$ is an $\mathcal{O}(T^{-\frac{1}{4\theta}})$-OS if $\mu \leq \mathcal{O}(T^{-\frac{2\theta-1}{2\theta}})$;
> > > > > > > > > >
> > > > > > > > > > - KL exponent $\theta \in (0,\frac{1}{2}]$: $(x^{t+1},y^{t+1})$ is an $\mathcal{O}(T^{-\frac{1}{2}})$-GS and $v^{t+1}$ is an $\mathcal{O}(T^{-\frac{1}{2}})$-OS if $\mu \leq \frac{3r_1}{25r_2(5\lambda+4)\beta\omega_2\left(2diam(\mathcal{X})\right)^{\frac{1}{\theta}-2}}$;
> > > > > > > > > >
> > > > > > > > > > - Concave: $(x^{t+1},y^{t+1})$ is an $\mathcal{O}(T^{-\frac{1}{4}})$-GS and $v^{t+1}$ is an $\mathcal{O(}T^{-\frac{1}{4}})$-OS if $\mu \leq \mathcal{O}(T^{-\frac{1}{2}})$.
> > > > > > > > > >
> > > > > > > > > > **Proof: The proof remains unchanged from thoes of Theorem 2, with the sole exchange between  $(x,z,r_1,c,\beta,L_x)$ and  $(y,v,r_2,\alpha,\mu,L_y)$. Notably, instead of invoking Proposition 1, we employ the symmetric counterpart (i.e., Proposition 2) to handle the negative term.**
> > > > > > > > > >
> > > > > > > > > >
> > > > > > > > > > **Theorem 6 (Iteration complexity of the universal DSGDA cross NC-C/NC-KL/C-NC/KL-NC)**
> > > > > > > > > > Without loss of generality, we assume $\lambda=1$ (i.e., $L_x=L_y=L$). Suppose that Assumption 1 holds and $r_1=r_2=70L$, $c=\alpha=1/(1400L)$, $\beta=\mu\leq \min\\{\mathcal{O}(T^{-1/2}),1/10000\\}$. Then, for any $t\geq 0$, the basic descent estimate (A) holds for $\Phi$ (refer to Corollary in the last response) and the basic descent estimate also holds for the symmetric Lyapunov function defined in Remark 1:
> > > > > > > > > > $$\Psi^t-\Psi^{t+1}\geq \frac{r}{1000}\\|y^{t+1}-y^t\\|^2+\frac{r}{1000}\\|x^t-x^t_{+}(z^t,v^t)\\|^2+\frac{r}{1000\mu}\\|v^t-v^{t+1}\\|^2+\frac{r}{1000\beta}\\|z_{+}^t(v^{t+1})-z^t\\|^2 -4r\mu\\|y(z(v^{t+1}),v^{t+1})-y(z_+^t(v^{t+1}),v^{t+1})\\|^2.$$
> > > > > > > > > > Then, for any $T>0$, there exists a $t\in\{1,2,\cdots,T\}$ such that $(x^{t+1},y^{t+1})$ is an $\mathcal{O}(T^{-\frac{1}{4}})$-GS.
> > > > > > > > > >
> > > > > > > > > > **Proof: For the negative terms in $\Phi$ and $\Psi$, we can use Proposition 1 or 2 to  bound them.**

---

### Author Rebuttal · Authors · 2023-08-09

We appreciate and thank the reviewers for their instructive comments. Below we provide our response to some common comments and questions. We believe that our responses will effectively address your major concerns, leading to a re-evaluation of our paper's contribution and quality.

**Q1: Is there any superiority compared with SGDA?**

Yes. One of the key advantages, the universality of DSGDA across NC-NC, NC-C, and C-NC problems, might have been overlooked by the reviewers. In practice, justifying the convexity/KL property of the primal or the concavity/KL property of the dual is a considerably challenging task. Our proposed DSGDA can be applied without knowing this prior information, owing to its inherent symmetry. However, prior to implementing SGDA, we have to choose which side we would like to employ the extrapolation. If we choose the wrong side, it will result in a slow convergence or even diverge. To validate this, we conduct a new experiment for a KL-NC problem $\min_{x\in \mathcal{X}} \max_{y \in \mathcal{Y}} f(x,y)=2x^2 - y^2 + 4xy^6 + 4y^3/3-y^4/4$ with $\mathcal{X}=\mathcal{Y}=\\{z:-1\leq z\leq 1\\}$. With a wrong smoothing side, SGDA (with primal smoothing) leads to a slower convergence compared with dual smoothing, see Figure (g) in the **PDF**. We kindly request all reviewers to reevaluate the universal applicability of DSGDA. We will provide more details on this universality in our updated version.

On the theoretical side, the convergence rate matches the results of SGDA under the same condition. On the practical side, DSGDA has better empirical performance than SGDA, which stands out as the only algorithm capable of overcoming the limit cycle phenomenon in all four challenging NC-NC examples (see Section 2). Moreover, DSGDA is more robust in parameter selection (see Section 5).

**Q2: Explanation on Theorem 1: Could authors make additional comments on conditions of theorem 1?**

Thank you for bringing up this matter. We would like to emphasize that we only offer a single set of workable upper and lower bounds for all hyperparameters. However, there are numerous alternative selections, which is essentially not a big deal. The principle behind selecting those hyperparameters is to ensure the sufficient decrease property of the novel Lyapunov function, see Theorem 1. In practice, as we have shown, the range of viable parameter values can be quite large.

**Q3: There remains a gap between theory and practice.**

Regarding the gap between the practical claim and the theoretical claim, it's important to emphasize that achieving a game stationary point through first-order oracles for smooth constrained separable NC-NC optimization problems (where constraint sets of primal and dual variables are independent) within polynomial time remains an open question over an extended period. Recently, the pioneering work by [1] has demonstrated that finding a game stationary point in smooth constrained **non-separable** NC-NC optimization problems is already a PPAD-complete problem. We believe our paper marks the initial stride towards bridging this gap and aims to encourage further engagement from researchers in tackling this issue.

[1] Daskalakis C, Skoulakis S, Zampetakis M. The complexity of constrained min-max optimization.

---

### Decision · Program_Chairs · 2023-09-21

**Decision:**

Accept (poster)

**Comment:**

This paper proposes a doubly smoothed GDA for nonconvex-nonconcave minimax problem. It can escape the limit cycle of several well-known examples. Convergence to a stationary point and the iteration complexity under one-sided KL assumption are established. The results are important complement to the literature of minimax problem.